# Learning Treatment Representations for Downstream Instrumental Variable Regression

## Abstract

Traditional instrumental variable (IV) estimators face a fundamental constraint: they can only accommodate as many endogenous treatment variables as available instruments. This limitation becomes particularly challenging in settings where the treatment is presented in a high-dimensional and unstructured manner (e.g. descriptions of patient treatment pathways in a hospital). In such settings, researchers typically resort to applying unsupervised dimension reduction techniques to learn a low-dimensional treatment representation prior to implementing IV regression analysis. We show that such methods can suffer from substantial omitted treatment bias, violating exclusion restriction principle, due to implicit regularization in the representation learning step. We propose a novel approach to construct treatment representations by explicitly incorporating instrumental variables during the representation learning process. Our approach provides a framework for handling high-dimensional endogenous variables with limited instruments. We demonstrate both theoretically and empirically that fitting IV models on these instrument-guided representations ensures identification of directions that optimize outcome prediction. Our experiments show that our proposed methodology improves upon the conventional two-stage approaches that perform dimension reduction without incorporating instrument information.

## 1 Introduction

Instrumental-variable (IV) methods are among the most widely used tools for recovering causal effects in the presence of unmeasured confounding. Unfortunately, classical IV estimators scale poorly when the treatment variable $X$ is itself high-dimensional, unstructured, or both. In modern applications—where the treatment might be provided in the form of clinical treatment pathways encoded as free-text, purchase histories, or genome-wide expression profiles—the number of potentially endogenous coordinates of $X$ can dwarf the number of available instruments $Z$ (e.g. variables related to capacity constraints in a hospital setting, see, e.g., Dong et al. (2019); Dong et al.; Qin et al. (2023)). A common workaround is to compress $X$ to a low-dimensional summary $D$ with unsupervised techniques (e.g. PCA, auto-encoders) and then run a standard two-stage least squares (2SLS) on $D$. Because the dimension reduction step ignores $Z$, however, the resulting regression can suffer from severe omitted-variable bias: directions of $X$ that matter for the first-stage relationship between $Z$ and $X$ may be discarded, violating the exclusion restriction and invalidating the causal inference step.

We propose *Instrument-Guided Representation Learning* (IGRL), a methodology for learning low-dimensional treatment representations that preserve the validity of downstream IV analysis. IGRL folds the instruments directly into the representation learner so that the learned features $D$ capture the variation in $X$ that is driven by $Z$. The procedure can be viewed as a regularization of the unsupervised learner toward directions that satisfy the exclusion restriction, thereby eliminating the spurious back-door paths that plague two-step approaches. The resulting representation can then be used in an IV analysis, to learn directions of intervention in the representation space that will improve the target outcome and can be translated back to interventions in the original treatment space.

Prior work on that combines elements of representation learning with elements of instrumental variable analysis is limited and confined to linear methods. Rao and Sabatier et al. described a procedure of performing principal component analysis (PCA) of a response variable with respect to its instruments. Y Takane studied constrained principal component analysis, which takes external information into consideration during dimensional reduction Y Takane (2001). More recently, Kelly et al. and Wang incorporates instrumental variables in estimating factor models that improves rate of convergence and avoid overfitting for high-dimensional data Kelly et al. (2020),Wang (2024). The desiderata in all of these works are very different from identifying dimensions of variation that align with the instruments so that causal effects can be identified by downstream IV analysis.

Our work is also related to the literature on learning non-linear disentangled representations and causal representation learning Hyvärinen and Oja (2000); Hyvärinen (2013); Khemakhem et al. (2020); Hälvä and Hyvarinen (2020); Monti et al. (2020); Schölkopf et al. (2021a); Ahuja et al. (2022); Hyvärinen et al. (2023); Jin and Syrgkanis (2023); Hyvärinen et al. (2024); Hälvä et al. (2024). However, the focus of this line of work has primarily been on discovering causal structure in data Schölkopf et al. (2021b), rather than constructing representations for downstream causal tasks. Our work is closely related to the identifiable VAE (iVAE) Khemakhem et al. (2020). The instrument can be viewed as the auxiliary information that can guide non-linear latent factor analysis. However, a crucial difference of our work is that we view the instrument $Z$ as only privileged information that is available only when estimating the causal effects and not when performing interventions. Hence, crucially we want our encoder to only take as input the treatment $X$ and not the instrument $Z$. Moreover, our desiderata is not the discovery of the true latent factors, but solely the discovery of valid decompositions of the treatment for downstream IV analysis. This allows us to relax many of the assumptions that are prevalent in this line of work.

Our work also shares technical similarities with the *Rep4Ex* approach of Saengkyongam et al.. However, *Rep4Ex* focuses on intervening on $Z$ instead of the latent treatment space ($D$). Moreover, the structural equation model we consider is richer than the one in Saengkyongam et al. in that it addresses endogenous treatments and allows for noisy or orthogonal components of variation of the observed high-dimensional treatment. In contrast, the mixing function in Saengkyongam et al. is deterministic. Other dimensionality reduction studies for high-dimensional treatments (Nabi et al., 2022; Andreu et al., 2024) operated without unobserved confounders and used outcome-guided factor selection. Additional discussion appears in the appendix. Our work aligns closer to the recent contributions by Vafa et al. and Du et al., which also highlights the omitted variable bias problem in learned representations in the context where representation learning is used for a set of high-dimensional observed confounders of a treatment and designs representation learning techniques to alleviate it. In that setting, the learned representation can implicitly omit important parts of the observed confounders, causing bias in the final causal estimate due to implicit unobserved confounding. Our goal is inherently different as we want to learn a latent representation of a highly confounded, high-dimensional treatment, as opposed to learning a latent representation of a high-dimensional confounder.

## 2 PROBLEM STATEMENT: LEARNING INTERVENTIONS VIA REPRESENTATIONS

We consider a setting where we are given data that contain samples of variables $(Z, X, Y)$, where $X$ is a high-dimensional "treatment" variable, $Y$ is a scalar outcome of interest and $Z$ is a low-dimensional vector of instruments. The treatment $X$ is heavily confounded via unobserved confounding variables $U$ that have a causal influence on the value of $X$ and also on $Y$, as depicted in Figure 1a.

Our goal is to learn a latent representation of the highly confounded, high-dimensional treatment, so as to perform instrumental variable analysis on this learned representation and identify an outcome-improving direction of intervention in representation space and hence subsequently also in the original treatment space. Naive representation learning approaches for the treatment run the risk of an omitted variable problem that can invalidate the downstream causal analysis based on instrumental variables. Causal analysis using instrumental variables crucially assumes that the instrument $Z$, the treatment $X$, and the outcome $Y$ respect the causal graph depicted in Figure 1a. In particular, the instrument $Z$ is assumed to only affect outcome $Y$ through its effect on treatment $X$. When the high-dimensional treatment $X$ is replaced by a learned representation $D$, we run the risk that the part of $X$ that is not represented in $D$ contains elements that are correlated with both the instrument $Z$ and the outcome

$Y$. As a result, $D$ no longer absorbs the entire effect of the instrumental variable Z on the outcome $Y$. This creates causal pathways from the instrument $Z$ to the outcome $Y$ that do not flow through the representation $D$, as shown in Figure 1b. Therefore, we need to regularize the representation learning process to ensure that the causal influence through these omitted paths is minimal.

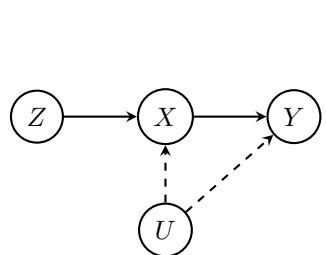 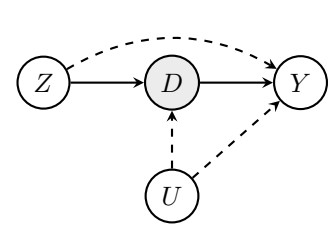 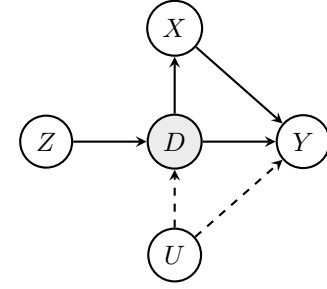

(a) Instrumental variable causal graph, with instruments $Z$, high-dimensional treatment $X$, outcome $Y$, unobserved confounders $U$.

(b) Causal graph when high-dimensional treatment $X$ is replaced by learned representation $D$.

(c) Causal graph that an ideal representation $D$ of the high-dimensional treatment $X$ would satisfy.

Figure 1: Omitted treatment bias in instrumental variable analysis with learned treatment representations.

An ideal latent representation $D$ should satisfy the causal graph depicted in Figure 1c. In particular, the instrument $Z$ should not have a causal effect on $X$ that is not absorbed by the latent representation $D$. If the representation encodes all outcome-relevant information, then a direct edge from $X$ to $Y$ should not exist. However, the existence of such an edge does not invalidate the downstream instrumental variable analysis, and hence, it is not essential to exclude it.

**Structural Equation Model.** To formalize our problem we will consider the following data generating process (structural causal model) [1] for our observed random variables:

$$
\begin{aligned}
D &= A \cdot Z + U, & U &\perp\!\!\!\perp Z \\
X &= f(D, V), & V &\perp\!\!\!\perp Z \\
Y &= h(D) + \eta(U, V, \epsilon), & \epsilon &\perp\!\!\!\perp Z
\end{aligned}
\tag{1}
$$

where the random variables $U, V, D, \epsilon$ are latent and $A$ is an $r \times k$ matrix that captures the effect of the instruments $Z \in \mathbb{R}^k$ on a vector of latent decisions $D \in \mathbb{R}^r$. For convenience of notation, we will assume that $\mathbb{E}[U] = \mathbb{E}[V] = \mathbb{E}[\eta(U, V, \epsilon)] = 0$.[2] $U$ represents the unobserved confounder that drives the elements of the treatment that are also driven by the instrument. $\epsilon$ represents an outcome noise variable and is allowed to be correlated with $U, V$. $D$ represents the aspects of the treatment $X$ that are affected by the instrument and $V$ represents the remaining aspects that describe the treatment $X$, but are independent of the instrument. In particular, we assume that the encoding/decoding between the latent representations and the observed treatment is invertible:

**Assumption 2.1** (Invertible Encoding). The function $f$ is invertible, and write the encoding function $e(X) = f^{-1}(X) = (D, V)$, i.e. there is a one-to-one correspondence between the high-dimensional treatment $X$ and the characteristics $(D, V)$ that describe the treatment.

From this perspective, $(D, V)$ can be thought as a non-linear decomposition of the treatment into the instrument-dependent and the instrument-independent components. We will further denote with $e_D(X) = D$ and $e_V(X) = V$ for the encodings of the treatment that return the corresponding components. Moreover, we assume that the transformation between the instrument to the latent representation $D$ is full rank.

**Assumption 2.2** (Full-Rank Latents). Assume that the matrix $A$ has full row-rank and $\mathbb{E}[ZZ^\top] \succ 0$.

---

[1]The inclusion of covariates is discussed in Appendix D.

[2]Appropriate intercept constants need to be added to the equations in the absence of this convention.

Note that the full rank assumption on $\mathbb{E}[ZZ^\top] \succ 0$ can always be satisfied by a preprocessing step that applies a PCA transformation to the instruments and removes co-linear or almost co-linear instruments.

**Learning Good Interventions via Representations.** Given data containing observations $(Z, X, Y)$ stemming from such a structural equation model, our goal is to learn a soft intervention mapping $t(X)$, such that the average intervened outcome is larger than the original outcome. We will denote with $Y^{(X \leftarrow x)}$ the random outcome from the intervention where we fix the value of $X$ to be $x$. Thus we are searching for a soft intervention $t(X)$ such that:

$$\mathbb{E}\left[ Y^{(X \leftarrow t(X))} \right] > \mathbb{E}[Y] \tag{2}$$

Note that due to the one-to-one correspondence of $X$ with its decomposition, any such interventional outcome can equivalently be thought as an intervention on the latent components of the treatment, i.e. $Y^{(D \leftarrow e_D(x), V \leftarrow e_V(x))}$. Given the structural Equation (1), the expected outcome under a soft intervention $t(X)$ can be written as:

$$\mathbb{E}\left[ Y^{(X \leftarrow t(X))} \right] = \mathbb{E}[h(e_D(t(X))) + \eta(U, e_V(t(X)), \epsilon)] \tag{3}$$

We will identify such an intervention via the means of intervention on a learned representation. In particular, given observations, we will learn an encoding $\tilde{e}_D(X) = \tilde{D}$ that respects the properties in Equation (1) (potentially together with a learned encoding $\tilde{e}_V(X) = \tilde{V}$) and a corresponding decoder $\tilde{f}(\tilde{D})$ (potentially also taking as input $\tilde{V}$) that maps the learned encoding back into a high-dimensional treatment. Subsequently, we will estimate an outcome improving direction $u$ in the learned representation space via instrumental variable analysis, viewing $\tilde{D}$ as the "treatment" and $Z$ as the instrument. We will apply the direction $u$ to the learned representations, i.e. $\tilde{D} + \alpha u$, for some scalar intervention amount $\alpha$. For ease of notation, we denote with $(\cdot)_{\alpha u}$ to be the corresponding random variable $(\cdot)$ after this intervention. Then decode back to the high-dimensional treatment space $X_{\alpha u} = \tilde{f}(\tilde{D} + \alpha u)$ (potentially $X_{\alpha u} = \tilde{f}(\tilde{D} + \alpha u, \tilde{V})$ if an encoding of $V$ was also learned). This process (depicted also visually in Figure 2 and described algorithmically in Algorithm 1) defines our soft-intervention mapping, formally defined as:

$$t(X) = \tilde{f}(\tilde{e}_D(X) + \alpha u, \tilde{e}_V(X)), \tag{4}$$

with the second input of $\tilde{f}$ omitted if an encoding $\tilde{e}_V$ is not learned.

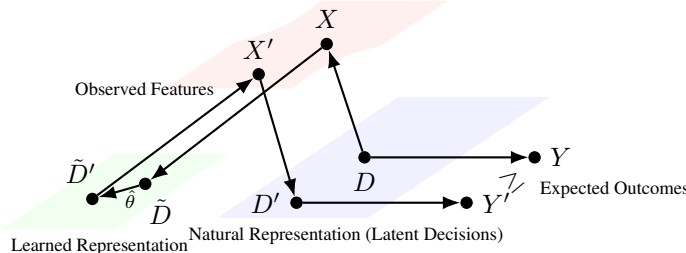

Figure 2: Intervention on learned representation.

## 3 INSTRUMENT GUIDED REPRESENTATION LEARNING: THE LINEAR SETTING

To make matters more concrete, we will start this analysis with the case where the structural equation model that is associated with the causal graph in Figure 1c contains only linear relationships:

$$\begin{aligned}
D &= A \cdot Z + U, & U &\perp\!\!\!\perp Z \\
X &= B \cdot D + B_\perp \cdot V, & V &\perp\!\!\!\perp Z \\
Y &= \theta^\top D + \eta(U, V, \epsilon), & \epsilon &\perp\!\!\!\perp Z
\end{aligned} \tag{5}$$

---

**Algorithm 1** Intervention in Latent Representation Space and evaluation

---

1: **Autoencoder fitting.** Learn encoder $\tilde{e}$ and decoder $\tilde{f}$ of $X$ and using observed data $(Z, X, Y)$.
2: **IV analysis.** Identify causal model $\tilde{h}(\tilde{D})$ using IV regression analysis with instrument $Z$, treatment $\tilde{D} \triangleq \tilde{e}_D(X)$ and outcome $Y$. Calculate average causal derivative $u = \mathbb{E}[\nabla_D \tilde{h}(\tilde{D})]$.
3: **Encode.** Transform $X$ into latent representation $\tilde{D}$ using learned encoder $\tilde{D} = \tilde{e}_D(X)$
4: **Perturb.** Apply perturbation in the latent space: $\tilde{D}_{\alpha u} = \tilde{D} + \alpha u$ where $\alpha$ is a scalar factor controlling perturbation magnitude.
5: **Decode.** Map perturbed latent representation $\tilde{D}_{\alpha u}$ back to input space: $X_{\alpha u} = \tilde{f}(\tilde{D}_{\alpha u})$ (or $X_{\alpha u} = \tilde{f}(\tilde{D}_{\alpha u}, \tilde{e}_V(X))$ if the learned encoder also learns a representation of $V$).
6: **Evaluate.** Apply the true decomposition $e(X_{\alpha u}) = (D_{\alpha u}, V_{\alpha u})$ and evaluate outcome under intervention: $Y_{\alpha u} = h(D_{\alpha u}) + \eta(U, V_{\alpha u}, \epsilon)$.
7: Compare average original outcome $Y$ to average perturbed outcome $Y_{\alpha u}$.

---

where $B$ is an $m \times r$ dimensional matrix that maps the $k$ instrument-driven latent decisions $D$ to the observed high-dimensional treatments $X \in \mathbb{R}^m$ *and is assumed to be full column rank*. $B_\perp$ is a matrix whose column space is orthogonal to the column space of $B$ and is also *assumed to be full column rank*. $U$ corresponds to a random vector of latent unobserved confounders that also affect decisions and outcomes. $\theta$ is an $r$ dimension vector capturing the direct effects of the latent decisions on the outcome. We will assume that the matrix $A$ is of full row rank, i.e., we have more instruments $Z$ than latent decisions $D$, and the instruments vary these latent dimensions in a full-rank manner.

Note that this setting falls under our general model since the function $f(D, V) = BD + B_\perp V$ is invertible. In particular, by the orthogonality of the column space of the two matrices and the fact that they are both full column rank, we have that:

$$e_D(X) \triangleq B^+ X = D \qquad\qquad e_V(X) \triangleq B_\perp^+ X = V \qquad (6)$$

where $B^+$ denotes the Moore-Penrose pseudo-inverse of a matrix and which is a left inverse for full column rank matrices, i.e. $B^+ = (B^\top B)^{-1} B^\top$. Moreover, note that we could have equivalently defined the structural equation for $X$ as:

$$X = B \cdot D + V, \qquad\qquad V \perp\!\!\!\perp Z \qquad (7)$$

We could always split the second part into $B \cdot V + B_\perp V$ and redefine $D \to D + V$, or equivalently redefine $U \to U + V$. The formulation in Equation 5 is chosen for notational convenience.

Our target quantity of interest is the overall effect $\theta$ of the latent factors $D$ on the outcome $Y$. If we could identify the latent factors $D, V$ from the observed variables, then we could simply use the improving intervention direction $u = \theta/\|\theta\|$. In this case, our improving intervention corresponds to $t(X) = B(e_D(X) + \alpha u) + B_\perp V = X + \alpha Bu$, with $D_{\alpha u} = D + \alpha u$ and $V_{\alpha u} = V$, which would lead to an internventional outcome of $Y_{\alpha u} = \theta^\top (D + \alpha u) + \eta(V, U, \epsilon)$, hence:

$$\mathbb{E}[Y_{\alpha u}] = \mathbb{E}[Y] + \alpha \theta^\top u = \mathbb{E}[Y] + \alpha \|\theta\| \qquad (8)$$

Note that in this linear setting, to perform the intervention, it suffices that solely learn a linear encoder $e_D(X) = B^+ X$, since the intervention can be performed implicitly as $t(X) = X + \alpha Bu$, which would not require learning an encoding for $V$. Hence we will take this approach in the remainder of this section. We will show that in this setting it is feasible to identify improving interventions, even though the natural latent decomposition $D$ might not be necessarily identifiable. We will show that we can always identify a representation $\tilde{D}$, such that $\tilde{D}$ is an invertible linear transformation of $D$.

Note that in this setting, a linear regression of $X$ on $Z$ uncovers the matrix $C = B \cdot A$ since our structural equation model implies the regression equation:

$$X = B \cdot A \cdot Z + B_\perp \cdot V + B \cdot U, \quad \mathbb{E}[B_\perp \cdot V + B \cdot U \mid Z] = B_\perp \mathbb{E}[V] + B \cdot \mathbb{E}[U] = 0 \quad (9)$$

Moreover, since $A$ is full row rank, the column space of $C$ can be proven to be the same as the column space of $B$. Thus, if we perform a *thin* singular value decomposition of $C = \mathcal{U}\Sigma\mathcal{V}^\top$, then the $m \times k$ matrix of left eigenvectors $\mathcal{U}$ can be used as matrix $\hat{B}$, as they correspond to an orthonormal basis of the column space of $C$ and, therefore, also of the column space of $B$. Consequently, $\Sigma\mathcal{V}^\top$ can be used

as $\hat{A}$. Subsequently, we can take $\tilde{D} = \hat{B}^\top X = \hat{B}^\top BD$. Since the column space of $\hat{B}$ is the same as the column space of $B$, the square matrix $P = \hat{B}^\top B$ is invertible. An intervention in the direction of $u$ in the learned representation can be thought of as an intervention in the direction of $P^{-1}u$ in the natural representation. An instrumental variable regression estimate, using $Z$ as the instrument, $\tilde{D}$ as the treatment, and $Y$ as the outcome, is characterized as the solution to the moment restriction: $\mathbb{E}[Z(Y - \tilde{\theta}^\top \tilde{D})] = 0$. It can be shown that as long as matrix $A$ is full row rank and the instruments are not co-linear, i.e. $\mathbb{E}[ZZ^\top] \succ 0$, then the above system has a unique solution, $\tilde{\theta} = (P^{-1})^\top \theta$, which is the correct causal effect of interventions on $\tilde{D}$. We will then learned representation space in the direction $u = \tilde{\theta}/\|\tilde{\theta}\|$. The implied intervention in the $X$-space is $t(X) = X + \alpha \hat{B}u$. Algorithm 2 formalizes this procedure and the following theorem formalizes these arguments and provides the outcome improvement guarantee for this intervention.

**Theorem 3.1.** *Under the linear structural equation model in Equation* (5) *and assuming $B, B_\perp$ have full column rank and Assumption 2.2 holds, then the representation and intervention produced by the LIRR algorithm satisfy: $\tilde{D} = PD$, for the invertible matrix $P \triangleq \hat{B}^\top B$. Moreover, $\tilde{\theta} = (P^{-1})^\top \theta$ and the interventional outcome satisfies the guaranteed improvement property:*

$$\mathbb{E}[Y_{\alpha u}] = \mathbb{E}[Y] + \alpha \|(P^{-1})^\top \theta\|$$

---

**Algorithm 2** Linear Instrument Regularized Representation (LIRR) and Intervention

---

1: **Input:** magnitude of intervention $\alpha$
2: Run linear regression of $X$ on $Z \in \mathbb{R}^k$, to estimate a coefficient matrix $C$
3: Calculate the *thin* SVD decomposition of $C = \mathcal{U}\Sigma\mathcal{V}^\top$, keeping only the top $k$ singular values
4: Define $\hat{B} = \mathcal{U}$ and $\hat{A} = \Sigma\mathcal{V}^\top$ and $\tilde{D} = \tilde{e}_D(X) = \hat{B}^\top X$
5: Run linear IV regression solving moment $\mathbb{E}[Z(Y - \tilde{\theta}^\top \tilde{D})] = 0$
6: Let $u = \tilde{\theta}/\|\tilde{\theta}\|$ and perform intervention on learned representation space $\tilde{D}_{\alpha u} = \tilde{D} + \alpha u$
7: Encode back to X-space intervention of $X_{\alpha u} = X + \alpha \hat{B}^\top u$

---

The LIRR algorithm offers substantial improvements over typical approaches to dimensionality reduction when one is faced with high-dimensional treatments and low dimensional instruments. See simulation results in Section 5.

## 4 INSTRUMENT GUIDED REPRESENTATION LEARNING: THE NON-LINEAR SETTING

We will now investigate the general setting introduced in Equation (1). In this non-linear setting, we will require some further assumptions on the latent factors. In particular, we will be assuming that the latent components $D$ are independent of the orthogonal components $V$ that constitute $X$ that are not driven by the instrument. In particular, we will assume the slightly stronger property of joint independence of $Z, U, V$, which implies that $D \perp\!\!\!\perp V$.

**Assumption 4.1** (Joint Independence). Assume that $Z \perp\!\!\!\perp U \perp\!\!\!\perp V$ (jointly independent).

Moreover, we will assume the regularity conditions that the mixing function $f$ is differentiable and that the instrument is supported on an open subset of $\mathbb{R}^k$.

**Assumption 4.2** (Differentiable Decoding Function). $f$ is a differentiable function with uniformly bounded derivatives.

**Assumption 4.3.** $\mathbb{E}[Z] = 0$ and the support of $Z$, $\mathcal{Z}$, is an open subset of $\mathbb{R}^k$.

We will make a completeness assumption on the strength of the instrument, which is a standard assumption for non-parametric instrumental variable identification Cui et al. (2024). We discuss sufficient conditions in the Appendix (Lemma C.4). In particular, it involves characteristic function assumptions that have also been typical in the identifiable latent factor literature Lu et al. (2021).

**Assumption 4.4** (Bounded Completeness). $D$ is bounded complete for Z, that is, for all bounded real functions $h$, we have that:

$$\mathbb{E}[h(D)|Z] = 0 \quad \text{a.s.} \quad \Rightarrow \quad h(D) = 0 \quad \text{a.s.}$$

**Theorem 4.5.** *Suppose that the data generating process follows the SEM described in Equation 1, and satisfies Assumptions 2.1 & 2.2 & 4.1 & 4.2 & 4.3 & 4.4. Let $(\tilde{D}, \tilde{V}) := (\tilde{e}_D(X), \tilde{e}_V(X)) = \tilde{e}(X)$ denote the learned representations. Consider encoder-decoder pairs with perfect reconstruction, i.e. $X = \tilde{f} \circ \tilde{e}(X)$. Then, for the solution $\tilde{e}, \tilde{f}$, and full row rank matrix $\tilde{A}$ that minimizes the objective:*

$$\mathbb{E}[\|\tilde{e}_D(X) - \tilde{A}Z\|^2] \tag{10}$$

*subject to the following constraints: 1) $\tilde{e}$ is a differentiable function with uniformly bounded derivatives. 2) $\tilde{A}$ has full row rank. 3) $\tilde{D} = \tilde{A}Z + \tilde{U}$ with $\tilde{U} \perp\!\!\!\perp Z$ and $\mathbb{E}[\tilde{U}] = 0$. we have that, with probability 1, $\tilde{D} = PD$ and $\tilde{U} = PU$, for $P = \tilde{A}A^+$. Moreover, the matrix $P$ is invertible.*

*Remark* 4.6. Note that the assumptions that $\mathbb{E}[Z] = 0$ and $\mathbb{E}[ZZ'] \succ 0$ are without loss of generality as we can always pre-process $Z$ by centering it and removing co-linear instruments. Moreover, in practice the assumption that $\tilde{D} = \tilde{A}Z + \tilde{U}$, with $\tilde{U} \perp\!\!\!\perp Z$ and $\mathbb{E}[\tilde{U}] = 0$ can be achieved by minimizing a square loss with an intercept, i.e.

$$\min_{e,f,A,c: e,f \text{ invertible}, e \circ f = \text{identity}} \mathbb{E}[\|e_D(X) - AZ - c\|^2]$$

and then defining $\tilde{D} = \tilde{e}_D(X) \triangleq e_D(X) - c$, $\tilde{f} = f + c$.

Subsequently, we identify an intervention as described in Algorithm 1. In particular, we will run an IV analysis, with $Z$ as the instrument, $\tilde{D}$ as the treatment, and $Y$ as the outcome, to estimate a causal model in representation space by finding a solution to the conditional moment restrictions:

$$\mathbb{E}[Y - \tilde{h}(\tilde{D}) \mid Z] = 0 \tag{11}$$

Since $\tilde{D} = PD$ and $\mathbb{E}[Y \mid Z] = \mathbb{E}[h(D) \mid Z]$, by the completeness assumption,

$$\mathbb{E}[h(D) - \tilde{h}(PD) \mid Z] = 0 \Rightarrow h(D) = \tilde{h}(PD) \text{ a.s.} \implies h(P^{-1}\tilde{D}) = \tilde{h}(\tilde{D}) \text{ a.s.}$$

If for instance, $h$ is assumed to be linear, then $\tilde{h}$ is also a linear function and it suffices to run a linear instrumental variable analysis (e.g. two-stage-least-squares). If $h$ is non-linear, then we calculate the average derivative of $\tilde{h}$, and perform the intervention $u$ as described in Algorithm 1. i.e.

$$\tilde{\theta} = \mathbb{E}[\nabla_{\tilde{D}}\tilde{h}(\tilde{D})] = (P^{-1})^\top \mathbb{E}[\nabla_D h(D)], \quad u = \tilde{\theta}/\|\tilde{\theta}\|$$

In finite samples, recently introduced doubly robust methods for estimation of average derivatives of solutions to non-parametric IV problems can be used Bennett et al. (2022; 2023).

**Theorem 4.7.** *Assume that:*

$$Y = h(D) + \eta(U, V, \epsilon), \quad \epsilon \perp\!\!\!\perp \{Z, U, V\}$$

*and that $h$ is twice differentiable with a bounded second derivative. Let $\tilde{e}, \tilde{f}, \tilde{A}$ be an optimal solution that minimize the objective in Equation 10, satisfying the assumptions of Theorem 4.5 with the extra constraint that: 1) $\tilde{D} \perp\!\!\!\perp \tilde{V}$. 2) $\tilde{U} \perp\!\!\!\perp \tilde{V}. \perp\!\!\!\perp Z$ 3) $\tilde{e}$ is an invertible function when restricted to inputs in the image of $f$ and $\tilde{f} \circ \tilde{e}(x) = x$ for all $x \in Im(f)$. Furthermore, assume that the variable $D$ has full support in $\mathbb{R}^r$, i.e. $\mathcal{D} = \mathbb{R}^r$. Then setting $u = \tilde{\theta}/\|\tilde{\theta}\|$, in Algorithm 1, with $\tilde{\theta} = \mathbb{E}[\nabla_{\tilde{D}}\tilde{h}(\tilde{D})]$ and $\tilde{h}$ the solution to the conditional moment restriction problem in Equation (16), we have that:*

$$\mathbb{E}[Y_{\alpha u} - Y] = \alpha \|(P^{-1})^\top \mathbb{E}[\nabla_D h(D)]\| + O(\alpha^2)$$

Hence, for small enough step size $\alpha$, the identified intervention will achieve a positive improvement on the outcome (assuming that $\mathbb{E}[\nabla_D h(D)] \neq 0$).

**Instrument Regularized Auto-Encoder** To achieve the positive improvement as described in Theorem 4.7, then we need to incorporate loss components that are minimized only when i) $e, f$ reconstruct the input $X$, ii) $e_D(X)$ is predicted linearly by $Z$ with a full rank matrix $A$, iii) the residual of this regression $e_D(X) - AZ - c$, which approximates $U$, needs to be independent of $Z$, iv) $Z$ needs to be independent of $e_V(X)$ and v) $e_D(X)$ needs to be independent of $e_V(X)$, $e_D(X) - AZ - c$ needs to be independent of $e_V(X)$, and the variables $(e_D(X) - AZ - c, e_V(X), Z)$

Table 1: Average Test Improvement Comparison of LIRR and PCA on Linear Data (Mean ± Std). DGP 1 corresponds to independent U and V, DGP 2 corresponds to correlated U and independent V, and lastly DGP 3 corresponds to correlated U and V. The average is computed across 100 random seeds, each containing a sample size of 10000 with 80-20 train-test split. The dimension of Z is $k = 4$ is constant and dimension of X is included as $m$ in the first column.

| Size $m$ | Method | DGP 1 | DGP 2 | DGP 3 |
|---|---|---|---|---|
| 50 | LIRR | **3.7283 ± 2.7360** | **5.4706 ± 4.1242** | **5.4944 ± 4.0596** |
| | PCA | 3.1035 ± 3.6229 | 3.1717 ± 4.0468 | 2.5171 ± 4.8519 |
| 100 | LIRR | **2.4189 ± 2.0164** | **4.0806 ± 3.5969** | **3.8931 ± 3.3116** |
| | PCA | 2.1249 ± 2.7203 | 2.4044 ± 3.5741 | 2.5713 ± 3.7491 |
| 500 | LIRR | **1.0355 ± 0.9698** | **1.6996 ± 1.5957** | **1.5934 ± 1.7305** |
| | PCA | 1.0098 ± 1.0786 | 0.9005 ± 1.3995 | 1.1716 ± 1.6904 |

are jointly independent. While we do not explicitly enforce $\tilde{A}$ to be full row rank, we expect this to be satisfied due to the reconstruction loss and the condition that $\tilde{D} \perp\!\!\!\perp \tilde{V}$. Note that in addition to joint independence, we also explicitly enforce pairwise independencies for computational reasons.

Denote the residual as $\delta := e_D(X) - AZ - c$. We introduce the instrument-regularized auto-encoder loss, which incorporates all these elements:

$$
\begin{aligned}
\min_{e,f,A,c} \; & \mathbb{E}\left[\|X - f \circ e(X)\|^2\right] + \lambda \mathbb{E}\left[\|\delta\|^2\right] + \mu_1 \mathcal{R}(\delta, Z) + \mu_2 \mathcal{R}(Z, e_V(X)) \\
& + \mu_3 \big(c_1 \mathcal{R}(e_D(X), e_V(X)) + c_2 \mathcal{R}(\delta, e_V(X)) + c_3 \mathcal{R}(\delta, Z, e_V(X))\big)
\end{aligned}
\tag{IRAE}
$$

$\mathcal{R}(A, B)$ or $\mathcal{R}(A, B, C)$, denotes any regularizer that can be evaluated on a set of $n$ samples and which takes small values when the random variables $A, B$ or $A, B, C$ are jointly independent. Many independence regularizers exist in the literature, and our method is agnostic to the choice. In our experiments, we use the kernel-based independence test statistic (HSIC) Gretton et al. (2007) and its generalization to joint independence (d-HSIC) Pfister et al. (2018).

In experiments, for the purposes of ablation analysis, we will denote with IRAE[0] the variant that contains only the regularization parts that are multiplied by $\lambda$, with IRAE[1] the variant that contains the parts that are multiplied by $\lambda, \mu_1$, with IRAE[2] the variant that contains the parts multiplied by $\lambda, \mu_1, \mu_2$ and IRAE the variant that contains all regularizers.

# 5 EXPERIMENTAL EVALUATION

**Linear setting** We benchmark LIRR (Section 3) against PCA in a linear data-generating setting. PCA is applied to extract the top $k = 4$ components of $X$ as the latent representation, after which steps 4–6 in Algorithm 1 with $\alpha = 1$ compute the improvement $\mathbb{E}[Y_{\alpha u} - Y]$. For each experiment, we randomly generate $A, B, \theta$ in Equation (5) from noraml distributions and test three noise scenarios: (1) independent Gaussian $U$ and $V$, (2) correlated Uniform $U$, independent Gaussian $V$, and (3) correlated Uniform $U$, correlated Gaussian $V$. Average improvements across seeds are reported in Table 1, with detailed procedures in the Appendix. Under independent Gaussian noise, PCA performs comparably to LIRR, but fails under correlated or non-independent noise. LIRR consistently outperforms PCA in DGP 2 and 3, with improvements exceeding one standard deviation from zero except for DGP 3 with $m = 500$. Performance decreases with larger $m$, which could be a result of the curse of dimensionality.

**Non-linear setting** We consider a non-linear data-generating process (Equation (1)) with quadratic $f$ and linear $h$. We benchmark LIRR and IRAE against PCA, vanilla AE, variational autoencoder (VAE), and iVAE. Vanilla AE refers to the autoencoder with only reconstruction loss. VAE maximizes the likelihood $p_f(X)$ with Gaussian latent representation. iVAE utilizes both $Z$ and $X$ in encoding and decoding, maximizing the conditional likelihood of $p_{f,A}(X|Z)$ as information of $Z$ is available in simulations (Khemakhem et al. (2020)). All methods use a bottleneck of $k = 4$ except IRAE[2] and IRAE so that downstream 2SLS will not be ill-posed. IRAE[2] and IRAE use $\text{Dim}(\tilde{D}) = 4$ and $\text{Dim}(\tilde{V}) = 6$. For probabilistic autoencoders (VAE, iVAE), 10 latent samples per observation are compared to the original outcome. Average improvements are evaluated using Algorithm 1. Results (Table 2) show that methods ignoring $Z$ (PCA, vanilla AE, vanilla VAE) yield minimal improvement, while those leveraging $Z$ consistently improve outcomes. IRAE[1] and IRAE achieve the largest gains, with IRAE exceeding one standard deviation above zero.

Table 2: Average Test Improvement Comparison of 9 Methods on Quadratic Data (Mean $\pm$ Std). DGP 1 corresponds to independent U and V, DGP 2 corresponds to correlated U and independent V, and lastly DGP 3 corresponds to correlated U and V. The average is computed across 30 random seeds, each containing a sample size of 10000 with 70-10-20 train-val-test split.

| Method | DGP 1 | DGP 2 | DGP 3 |
|---|---|---|---|
| PCA | $0.1322 \pm 0.3216$ | $0.0545 \pm 0.2994$ | $0.0848 \pm 0.2382$ |
| LIRR | $3.5086 \pm 2.0455$ | $3.4711 \pm 1.9683$ | $3.5682 \pm 2.1296$ |
| Vanilla AE | $0.4138 \pm 2.2000$ | $0.8418 \pm 1.1560$ | $0.7801 \pm 1.7335$ |
| IRAE[0] | $6.1055 \pm 7.1634$ | $2.2898 \pm 6.9957$ | $4.8993 \pm 6.3310$ |
| IRAE[1] | $\mathbf{6.4174 \pm 5.2602}$ | $4.6175 \pm 5.0479$ | $\mathbf{5.8023 \pm 7.1041}$ |
| IRAE[2] | $5.5471 \pm 4.6573$ | $4.5554 \pm 4.0707$ | $5.2145 \pm 4.4358$ |
| IRAE | $5.7740 \pm 4.7664$ | $\mathbf{6.5253 \pm 6.0132}$ | $4.9113 \pm 4.0009$ |
| Vanilla VAE | $0.3651 \pm 0.4629$ | $0.2725 \pm 0.5071$ | $0.2055 \pm 0.3394$ |
| iVAE | $0.2709 \pm 0.3672$ | $0.1192 \pm 0.2503$ | $0.1652 \pm 0.2929$ |

Table 3: Average Test Improvement Comparison of 5 Methods on MNIST Data (Mean $\pm$ Std).

| Sample Size | Image | Vanilla AE | IRAE[0] | IRAE[1] | IRAE[2] | IRAE |
|---|---|---|---|---|---|---|
| 10000 | reconstructed | $-0.51 \pm 0.03$ | $-0.73 \pm 0.04$ | $-0.73 \pm 0.04$ | $\mathbf{-0.33 \pm 0.05}$ | $\mathbf{-0.33 \pm 0.04}$ |
|  | intervened(0.2) | $-0.5 \pm 0.03$ | $-0.07 \pm 0.15$ | $0.07 \pm 0.26$ | $0.72 \pm 0.48$ | $\mathbf{0.87 \pm 0.43}$ |
|  | intervened(1.0) | $-0.47 \pm 0.04$ | $0.04 \pm 0.2$ | $0.15 \pm 0.29$ | $0.92 \pm 0.47$ | $\mathbf{1.1 \pm 0.46}$ |
| 30000 | reconstructed | $-0.51 \pm 0.03$ | $-0.74 \pm 0.04$ | $-0.73 \pm 0.04$ | $\mathbf{-0.33 \pm 0.05}$ | $-0.33 \pm 0.06$ |
|  | intervened(0.2) | $-0.49 \pm 0.03$ | $-0.11 \pm 0.08$ | $-0.17 \pm 0.11$ | $\mathbf{0.38 \pm 0.4}$ | $0.33 \pm 0.44$ |
|  | intervened(1.0) | $-0.44 \pm 0.05$ | $-0.06 \pm 0.12$ | $-0.13 \pm 0.2$ | $\mathbf{0.78 \pm 0.43}$ | $0.69 \pm 0.56$ |
| 60000 | reconstructed | $-0.47 \pm 0.02$ | $-0.71 \pm 0.05$ | $-0.71 \pm 0.04$ | $\mathbf{-0.25 \pm 0.05}$ | $-0.25 \pm 0.06$ |
|  | intervened(0.2) | $-0.46 \pm 0.03$ | $-0.06 \pm 0.26$ | $0.14 \pm 0.3$ | $\mathbf{0.98 \pm 0.42}$ | $0.9 \pm 0.52$ |
|  | intervened(1.0) | $-0.41 \pm 0.06$ | $0.05 \pm 0.29$ | $0.26 \pm 0.36$ | $\mathbf{1.13 \pm 0.44}$ | $1.05 \pm 0.51$ |

**MNIST experiment 1** We examine a case where the outcome depends on the color of the MNIST digits. Two-dimensional instruments $Z$ and confounders $U$ determine three-dimensional RGB features $D$, and the outcome is the sum of R, G, B. Observed data $X$ consists of MNIST pixels. Successful methods should produce intervened images with increased digit brightness. Vanilla AE, IRAE[0], and IRAE[1] use bottleneck size equal to $Z$, while IRAE[2] and IRAE use bottleneck size of 10. The methods are evaluated across 40 seeds, and results are in Table 3 with additional visualizations in the Appendix. As a remark, we note that having multiple dependency penalty terms may be difficult to train. For this reason, initialized the weights of IRAE[2] and IRAE as those in IRAE[1] before further training. Our experiments provide insights into latent space representations. Vanilla AE, lacking specialized latent regularization, mainly reconstructs digit morphology, as shown in Figure 3e, and IV regression on this representation yields no meaningful directional information. Introducing instrument regularization in IRAE[1] encourages the latent space to capture more color information at the expense of digit detail, producing a better improvement than Vanilla AE. Expanding the latent dimension (IRAE[2] and IRAE) allows simultaneous preservation of digit morphology and color information, reducing reconstruction error while enabling IVs to recover the target direction, as illustrated in Figure 3a. IRAE[2] shows slightly less improvement than IRAE due to residual dependence between $D$ and $V$, and adding a dependency penalty in IRAE further improves performance.

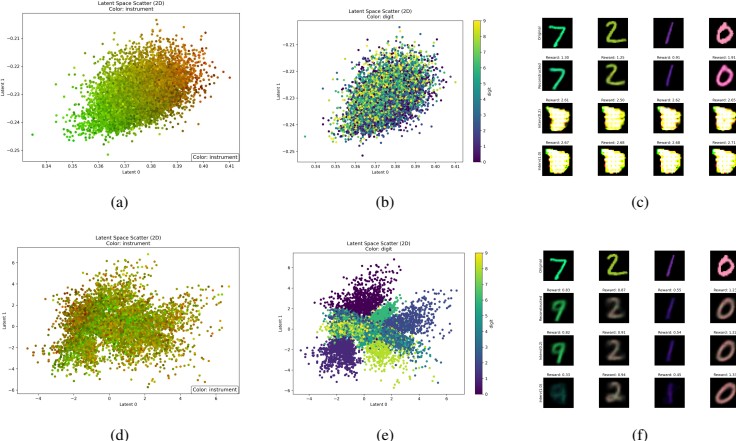

Figure 3: Comparison of IRAE (top row) and Vanilla AE (bottom row) models for Case 1 DGP. (a,d) Scatter plots of latent variables colored by instrument. (b,e) Scatter plots of latent variables colored by digit. (c,f) Original, reconstructed, and intervened ($\alpha = 0.2$ and $\alpha = 1.0$) image for digit 7, 2, 1, and 0.

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

## A    FURTHER RELATED WORK

In this section we provide a more discussion on related work that is not covered in the main text.

**Identifying Representations for Intervention Extrapolation** Similar to our work, Saengkyongam et al. proposed the *Rep4Ex* approach which tries to solve the task of interventional outcome prediction by identifying the SCM. Importantly, although they work with a similar SCM as we do (Equation 1), the level of intervention differs - our work considers interventions on the latent treatment space ($D$), while Saengkyongam et al. considers intervening on $Z$ (using notations in Equation 1). Moreover, our work is motivated by the presence of unobserved confounding between the latent representation of the treatment and the outcome, whereas their work is motivated by the need to extrapolate to unseen interventions, while the treatment that they consider is fully exogenous. Like our approach, they employ autoencoders to learn latent representations from potentially high-dimensional observed features, but use maximum moment restriction (MMR) regularization Muandet et al. (2020) to enforce the constraint $E[e_D(X) - AZ|Z] = 0$. This can be achieved when $E[e_D(X) - AZ] = 0$ and $e_D(X) - AZ \perp\!\!\!\perp Z$, corresponding to our $\lambda$ and $\mu_1$ term in Equation (IRAE). Additionally, while *Rep4Ex* assumes a deterministic mixing function from the latent representation to the observables $X$, our method explicitly handles noisy observations of X through $e_V(X)$, which allows for broader generalization.

**Dimensionality Reduction for High Dimensional Treatments** When learning a representation for the treatment, it is important for the learned representation to capture all causal factors so that the causal relationship is preserved for downstream estimation tasks like treatment effect estimation. Nabi et al.utilize semi-parametric inference theory for structural models to provide a generalized the sufficient dimension reduction approach for learning lower-dimensional representation for treatment, while capturing the relationship between the treatment and the mean counterfactual outcome. Andreu et al. employed a contrastive approach to learn a representation of the high-dimensional treatments. These works studied settings that did not involve the presence of unobserved confounders of the treatment, while we focus on heavily confounded high dimensional structured treatments. Moreover, in these works, the selection of causally relevant factors are guided by the outcome, where as we take an inherently different approach that learns the latent representations using auxiliary information from instrumental variables instead of the treatment.

**Independence Conditions** In our work, we show that independence between certain variables (for more details, see Theorem 4.5) is desirable for identification. We enforce the independence condition by incorporating a Hilbert-Schmidt Independence Criterion (HSIC) Gretton et al. (2007) regularizer. This approach has also been adopted in prior research: for instance, Lopez et al. employed HSIC regularization to mitigate bias in observational datasets for applications in counterfactual policy optimization, while Harada and Kashima use it to learn a representations of the treatment that is independent with the target individual in order to mitigate selection bias.

## B    PROOF OF LINEAR IDENTIFICATION

Before proving the main theorem, we first present some useful lemma.

**Lemma B.1.** *Suppose $A$ is a $n \times k$ matrix with full row rank ($k > n$), and $B$ is a $m \times n$ matrix, with full column rank ($m > n$). Then the columns of $C = BA$ spans the same space as the columns of $B$.*

*Proof of Lemma B.1.* Let $\mathcal{R}(\cdot)$ denote the column space of a matrix.

For any $x \in \mathcal{R}(B)$, there exist vector $y$ such that $x = By$. Since $A$ is full row rank, we know that $AA^+ = I_n$, and $x = By = BAA^+y = C(A^+y)$. Therefore $x \in \mathcal{R}(C)$, so $\mathcal{R}(B) \subseteq \mathcal{R}(C)$.

Similarly, for any $x \in \mathcal{R}(C)$, there exist vector $y$ such that $x = BAy = B(Ay)$. So $x \in \mathcal{R}(B)$, and we have $\mathcal{R}(C) \subseteq \mathcal{R}(B)$.

Together, we have $\mathcal{R}(C) = \mathcal{R}(B)$.

$\square$

Now we proceed to prove Theorem 3.1.

*Proof of Theorem 3.1.* From Equation 9, we have that:

$$X = BAZ + B_\perp V + BU$$

Then taking the conditional expectation over $Z$, we have:

$$\begin{aligned}
\mathbb{E}[X|Z] &= BAZ + \mathbb{E}[B_\perp V + BU] \\
&= BAZ + \mathbb{E}[B_\perp \mathbb{E}[V|Z]] + \mathbb{E}[B\mathbb{E}[U|Z]] \\
&= BAZ + B_\perp \mathbb{E}[V] + B\mathbb{E}[U] \qquad\qquad \text{(Since } V \perp\!\!\!\perp Z \text{ and } U \perp\!\!\!\perp Z) \\
&= BAZ
\end{aligned}$$

Thus $C := BA$ can be uniquely identified as the solution to the linear regression problem, regressing $X$ on $Z$. Consider the SVD decomposition of $C = \mathcal{U}\Sigma\mathcal{V}^\top$. Let $\hat{B} = \mathcal{U}$, and $\hat{A} = \Sigma\mathcal{V}^\top$. Then by Lemma B.1, we have that the columns of $\hat{B}$ spans the same space as the columns of $B$. In other words, there exist an invertible change of basis matrix $P$ such that $B = \hat{B}P$. Since $\hat{B}$ is orthonormal (by construction of SVD), we have that $\hat{B}^T\hat{B} = I_r$, and $P = \hat{B}^T B$. As a result, we also have:

$$\begin{aligned}
D = B^+ X &= (B^T B)^{-1} B^T X \\
&= (P^T \hat{B}^T \hat{B} P)^{-1} P^T \hat{B}^T X \\
&= (P^T P)^{-1} P^T \hat{B}^T X \\
&= P^{-1} \hat{B}^T X = P^{-1}\tilde{D}
\end{aligned}$$

Next, we show that $\tilde{\theta} = (P^{-1})^T\theta$. The LIRR algorithm solves for $\tilde{\theta}$ from the following moment equation:

$$\begin{aligned}
0 &= \mathbb{E}[Z(Y - \tilde{\theta}^T \hat{D})] \\
&= \mathbb{E}[Z(\theta^T D + \eta(V, U, \epsilon) - \hat{\theta}^T P D)] \\
&= \mathbb{E}[Z(\theta^T D - \tilde{\theta}^T P D)] \qquad\qquad \text{(Since } U, V, \epsilon \perp\!\!\!\perp Z \text{ and } \mathbb{E}[\eta(U, v, \epsilon)] = 0) \\
&= \mathbb{E}[ZD^T](\theta - P^T\tilde{\theta}) \\
&= \mathbb{E}[Z(Z^T A^T + U^T)](\theta - P^T\tilde{\theta}) \\
&= \mathbb{E}[ZZ^T]A^T(\theta - P^T\tilde{\theta})
\end{aligned}$$

Since the instruments are not co-linear, we have that $\mathbb{E}[ZZ^T] \succ 0$, i.e. $\mathbb{E}[ZZ^T]$ is invertible. Thus $\mathbb{E}[ZZ^T]A^T(\theta - P^T\tilde{\theta}) = 0$ if and only if $A^T(\theta - P^T\tilde{\theta}) = 0$. Since $A^T$ has full column rank, then by the Rank-Nullity theorem, the null space of $A^T = 0$. Together, this shows that $\tilde{\theta} = (P^{-1})^T\theta$ is the unique solution to the moment condition.

Lastly, we show that the intervened outcome is guaranteed improvement in expectation. Consider an intervention in the direction of $u = \tilde{\theta}/\|\tilde{\theta}\|$ in the $\tilde{D}$ space, this maps to an intervention in the $D$ space as:

$$\begin{aligned}
e_D(t(X)) = B^+ t(X) &= D + \alpha B^+ \hat{B}\tilde{\theta} \\
&= D + \alpha P^{-1}\frac{\tilde{\theta}}{\|\tilde{\theta}\|} = D + \alpha P^{-1}\frac{(P^{-1})^\top\theta}{\|(P^{-1})^\top\theta\|}
\end{aligned}$$

Since, we intervene only in $D$, $e_V(t(X)) = V$. Then, we can compute the intervened outcome:

$$\begin{aligned}
\mathbb{E}[Y_{\alpha u}] &= \mathbb{E}[\theta^T e_D(t(X)) + \eta(e_V(t(X)), U, \epsilon)] \\
&= \mathbb{E}[\theta^T e_D(t(X))] \qquad\qquad (e_V(t(X)) = V, \text{ and } \mathbb{E}[\eta(U, v, \epsilon)] = 0) \\
&= \mathbb{E}\left[\theta^T\left(D + \alpha P^{-1}\frac{(P^{-1})^\top\theta}{\|(P^{-1})^\top\theta\|}\right)\right] \\
&= \mathbb{E}[\theta^T D + \alpha\|(P^{-1})^\top\theta\|] = \mathbb{E}[Y] + \alpha\|(P^{-1})^\top\theta\|
\end{aligned}$$

$\square$

## C   PROOF OF NON-LINEAR IDENTIFICATION

### C.1   THEOREM 4.5

*Proof of Theorem 4.5.* By definition of $(\tilde{D}, \tilde{V})$, we have:

$$(\tilde{D}, \tilde{V}) = \tilde{e}(X) = \tilde{e} \circ f(D, V) =: q(D, V)$$

Denote with $q_1(D, V)$ the $\tilde{D}$ component of the output of $q$ and $q_2$ the $\tilde{V}$ component.

Since we have that $\tilde{D} = \tilde{A}Z + \tilde{U}$, with $\tilde{U} \perp\!\!\!\perp Z$ and $\mathbb{E}[\tilde{U}] = 0$, we can write:

$$\mathbb{E}[\tilde{D} \mid Z = z] = \mathbb{E}[\tilde{A}Z + \tilde{U} \mid Z = z] = \tilde{A}z$$

Moreover:

$$
\begin{aligned}
\mathbb{E}[\tilde{D} \mid Z = z] &= \mathbb{E}[q_1(D, V) \mid Z = z] \\
&= \mathbb{E}[q_1(Az + U, V) \mid Z = z] \\
&= \mathbb{E}[q_1(Az + U, V)] & (Z \perp\!\!\!\perp \{U, V\}) \\
&= \mathbb{E}_U[\mathbb{E}_V[q_1(Az + U, V)]] & (U \perp\!\!\!\perp V) \\
&= \mathbb{E}_U[\tilde{q}_1(Az + U)] & (\tilde{q}_1(d) \triangleq \mathbb{E}_V[q_1(d, V)]) \\
&= \mathbb{E}[\tilde{q}_1(Az + U)]
\end{aligned}
$$

Thus we can conclude that:

$$\tilde{A}z = \mathbb{E}[\tilde{D} \mid Z = z] = \mathbb{E}[\tilde{q}_1(Az + U)]$$

Since this holds for all $z \in \mathcal{Z}$ and since $\mathcal{Z}$ is an open set, we can take the derivative with respect to $z$, to derive:

$$\forall z \in \mathcal{Z} : \tilde{A} = \partial_z \mathbb{E}[\tilde{q}_1(Az + U)]$$

Since $q_1$ is continuously differentiable with bounded derivatives, the same holds for $\tilde{q}_1$ and therefore we can exchange the order of differentiation and expectation:

$$\tilde{A} = \mathbb{E}[\partial_z \tilde{q}_1(Az + U)]$$

Letting $\hat{q}_1^{(1)}$ denote the Jacobian of the function $\tilde{q}_1(d)$, we can write by the chain rule:

$$
\begin{aligned}
\tilde{A} &= \mathbb{E}[\tilde{q}_1^{(1)}(Az + U)A] \\
&= \mathbb{E}[\tilde{q}_1^{(1)}(Az + U)]A \\
&= \mathbb{E}[\tilde{q}_1^{(1)}(Az + U) \mid Z = z]A & (Z \perp\!\!\!\perp U) \\
&= \mathbb{E}[\tilde{q}_1^{(1)}(AZ + U) \mid Z = z]A \\
&= \mathbb{E}[\tilde{q}_1^{(1)}(D) \mid Z = z]A
\end{aligned}
$$

Since $A$ is full row rank, we have that $AA^+$ is invertible. Thus we can write:

$$\tilde{A}A^+ = \mathbb{E}[\tilde{q}^{(1)}(D) \mid Z = z]$$

or equivalently:

$$\forall z \in \mathcal{Z} : \mathbb{E}[\tilde{q}_1^{(1)}(D) - \tilde{A}A^+ \mid Z = z] = 0$$

By the bounded completeness assumption and since both $\tilde{A}A^+$ and $\tilde{q}_1^{(1)}$ are bounded, the latter implies that:

$$\forall d \in \mathcal{D} : \tilde{q}_1^{(1)}(d) = \tilde{A}A^+$$

or equivalently that:

$$\tilde{q}_1(d) = \tilde{A}A^+d + \tilde{\nu}$$

for some constant vector $\nu$. Moreover,

$$\mathbb{E}[\tilde{D}] = \mathbb{E}[\tilde{q}_1(D)]$$
$$= \tilde{A}A^+\mathbb{E}[D] + \tilde{\nu}$$
$$= \tilde{A}A^+A\mathbb{E}[Z] + \tilde{A}A^+\mathbb{E}[U] + \tilde{\nu}$$
$$= \tilde{\nu}$$

But we also have $\mathbb{E}[\tilde{D}] = A\mathbb{E}[Z] + \mathbb{E}[\tilde{U}] = 0$. Hence, we have that $\tilde{\nu} = 0$. Thus:

$$\forall d \in \mathcal{D} : \tilde{q}_1(d) = \tilde{A}A^+d$$

Next, we argue that $\tilde{A}A^+$ is an invertible matrix. Note that:

$$\mathbb{E}[\tilde{D}Z^\top] = \mathbb{E}[(\tilde{A}Z + \tilde{U})Z^\top]$$
$$= \tilde{A}\mathbb{E}[ZZ^\top] \qquad\qquad (\tilde{U} \perp\!\!\!\perp Z, \mathbb{E}[Z] = 0, \mathbb{E}[\tilde{U}] = 0)$$

Moreover:

$$\mathbb{E}[\tilde{D}Z^\top] = \mathbb{E}[q_1(D,V)Z^\top]$$
$$= \mathbb{E}[q_1(AZ + U, V)Z^\top]$$
$$= \mathbb{E}[\mathbb{E}[q_1(AZ + U, V) \mid Z, U]Z^\top]$$
$$= \mathbb{E}[\tilde{q}_1(AZ + U)Z^\top] \qquad\qquad (Z \perp\!\!\!\perp U \perp\!\!\!\perp V)$$
$$= \mathbb{E}[\tilde{q}_1(D)Z^\top]$$
$$= \mathbb{E}[(\tilde{A}A^+D)Z^\top]$$
$$= \tilde{A}A^+\mathbb{E}[DZ^\top]$$
$$= \tilde{A}A^+\mathbb{E}[(AZ + U)Z^\top]$$
$$= \tilde{A}A^+A\mathbb{E}[ZZ^\top] \qquad\qquad (Z \perp\!\!\!\perp U, \mathbb{E}[U] = 0)$$

Thus we have concluded that:

$$\tilde{A}\mathbb{E}[ZZ^\top] = \mathbb{E}[\tilde{D}Z^\top] = \tilde{A}A^+A\mathbb{E}[ZZ^\top]$$

Since $\mathbb{E}[ZZ^\top]$ is assumed to be invertible, the latter implies that:

$$\tilde{A} = \tilde{A}A^+A$$

By Lemma C.2 in Appendix C.3, since $\tilde{A}$ and $A$ have full row rank, the row span of $\tilde{A}$ is equal to the row span of $A$ and the matrix $\tilde{A}A^+$ is invertible.

We have thus concluded that:

$$\forall d \in \mathcal{D} : \tilde{q}_1(d) = \tilde{A}A^+d$$

and $\tilde{A}A^+$ is invertible.

Consider any solution with perfect encoder-decoder pair $(\tilde{e}, \tilde{f})$, and $\tilde{A}$ that satisfies the conditions of the theorem and minimizes the objective function:

$$\mathbb{E}[\|\tilde{e}_D(X) - \tilde{A}Z\|^2] = \mathbb{E}[\|\tilde{D} - \tilde{A}Z\|^2]$$

For any feasible solution, we can decompose this objective into two components by centering around

$$\mu_{\tilde{A}}(d) \triangleq \tilde{A}A^+d$$

i.e.:

$$\mathbb{E}[\|\tilde{D} - \tilde{A}Z\|^2] = \mathbb{E}[\|\tilde{D} - \mu_{\tilde{A}}(D) + \mu_{\tilde{A}}(D) - \tilde{A}Z\|^2]$$
$$= \mathbb{E}[\|\tilde{D} - \mu_{\tilde{A}}(D)\|^2 + \|\mu_{\tilde{A}}(D) - \tilde{A}Z\|^2 + 2\mathbb{E}[(\tilde{D} - \mu_{\tilde{A}}(D))^\top(\mu_{\tilde{A}}(D) - \tilde{A}Z)]$$

Consider the inner product term. Since we have that:

$$\mathbb{E}[\tilde{D} - \mu_{\tilde{A}}(D) \mid D, Z] = \mathbb{E}[q_1(D, V) - \tilde{q}_1(D) \mid D, Z]$$
$$= \mathbb{E}[q_1(D, V) \mid D, Z] - \tilde{q}_1(D)$$
$$= \mathbb{E}[q_1(D, V) \mid D, Z] - \mathbb{E}[q_1(D, V) \mid D]$$

Since $Z \perp\!\!\!\perp U \perp\!\!\!\perp V$, we have by Lemma C.3 in Appendix C.3 that $Z \perp\!\!\!\perp V \mid \mathbb{1}\{AZ + U = d\}$:

$$\mathbb{E}[q_1(D, V) \mid D = d, Z] = \mathbb{E}[q_1(d, V) \mid D = d, Z] = \mathbb{E}[q_1(d, V) \mid D = d] = \mathbb{E}[q_1(D, V) \mid D = d]$$

Thus:

$$\mathbb{E}[\tilde{D} - \mu_{\tilde{A}}(D) \mid D, Z] = 0$$

From this we conclude that for any feasible solution $\tilde{e}, \tilde{f}, \tilde{A}$, we have that the objective can be decomposed as:

$$\mathbb{E}[\|\tilde{e}_D(X) - \tilde{A}Z\|^2] = \mathbb{E}[\|\tilde{D} - \mu_{\tilde{A}}(D)\|^2] + \mathbb{E}[\|\mu_{\tilde{A}}(D) - \tilde{A}Z\|^2]$$
$$= \mathbb{E}[\|q_1(D, V) - \mu_{\tilde{A}}(D)\|^2] + \mathbb{E}[\|\mu_{\tilde{A}}(D) - \tilde{A}Z\|^2]$$

Suppose that with positive probability, we have that $q_1(D, V) \neq \mu_{\tilde{A}}(D) = \tilde{A}A^+ D$. Then we have that:

$$\mathbb{E}[\|q_1(D, V) - \mu_{\tilde{A}}(D)\|^2] > 0$$

In this case, we will provide an alternative feasible solution, which achieves smaller objective than $\tilde{e}, \tilde{f}, \tilde{A}$. Consider the solution:

$$\tilde{e}'(x) = (\tilde{A}A^+ e_D(x), e_V(x))$$
$$\tilde{f}'(d, v) = f((\tilde{A}A^+)^{-1} d, v)$$

Note that we used the fact that for any feasible solution, we have already shown that $\tilde{A}A^+$ is invertible. Moreover, note that for this solution we have that, $\tilde{e}', \tilde{f}'$ is invertible, since $e, f$ is invertible and $\tilde{A}A^+$ is invertible. Finally,

$$\tilde{f}' \circ \tilde{e}'(x) = x$$
$$\tilde{e}'(f(d, v)) = (\tilde{A}A^+ e_D(f(d, v)), e_V(f(d, v))) = (\tilde{A}A^+ d, v)$$

Thus:

$$\tilde{D} = \tilde{e}'_D(X) = \tilde{e}'_D(f(D, V)) = \tilde{A}A^+ D = \tilde{A}A^+ AZ + \tilde{A}A^+ U = \tilde{A}Z + \tilde{A}A^+ U$$

Where we used the fact that we have already shown (in the proof of Theorem 4.5) that $\tilde{A}A^+ A = \tilde{A}$. Thus, we also have that:

$$\tilde{D} = \tilde{A}Z + \tilde{U}$$

where $\tilde{U} = \tilde{A}A^+ U$ and satisfies $\tilde{U} \perp\!\!\!\perp Z$ and $\mathbb{E}[\tilde{U}] = 0$.

Therefore, this new solution is a feasible solution. Moreover, since under this solution we have that $\tilde{D} = \tilde{A}A^+ D = \mu_{\tilde{A}}(D)$, the first part of the objective vanishes and the objective takes the value:

$$\mathbb{E}[\|\mu_{\tilde{A}}(D) - \tilde{A}Z\|^2] < \mathbb{E}[\|q_1(D, V) - \mu_{\tilde{A}}(D)\|^2] + \mathbb{E}[\|\mu_{\tilde{A}}(D) - \tilde{A}Z\|^2]$$

contradicting the optimality of the original solution.

Thus we have derived that for any optimal feasible solution, it must hold that with probability 1:

$$\tilde{D} = q_1(D, V) = \mu_{\tilde{A}}(D) = \tilde{A}A^+ D \tag{12}$$

with $\tilde{A}A^+$ an invertible matrix. Moreover, this implies that $\tilde{U} = \tilde{D} - \tilde{A}Z = \tilde{A}A^+ U$. $\qquad\square$

## C.2 PROOF OF POSITIVE IMPROVEMENT

*Proof of Theorem 4.7.* In this proof, we show that intervention in the direction of average derivatives of $\tilde{h}$ guarantees positive improvement for sufficiently small $\alpha$, assuming that $h$ is twice differentiable. If we perform the intervention $\tilde{D} + \alpha u$, then we have by Lemma C.1 that:

$$D_{\alpha u}, V_{\alpha u} = (D + \alpha P^{-1}u, q_2^{-1}(\tilde{D} + \alpha u, \tilde{V}))$$

Since $\mathcal{D} = \mathbb{R}^r$ and since $P$ is an invertible matrix, we have that $\tilde{\mathcal{D}} = \mathbb{R}^r$. Thus, for all $d \in \tilde{\mathcal{D}}$, we also have that $d + au \in \hat{\mathcal{D}}$. By Lemma C.1, have that for all $d \in \hat{\mathcal{D}}$:

$$\text{Law}(q_2^{-1}(d, \tilde{V})) = \text{Law}(q_2^{-1}(d + \alpha u, \tilde{V}))$$

By Theorem 4.5, we have that, with probability 1, $\tilde{D} = PD$ and $\tilde{U} = PU$. Moreover, by assumption, we have that $\tilde{V} \perp\!\!\!\perp \tilde{U} \perp\!\!\!\perp Z$, which implies

$$\tilde{V} \perp\!\!\!\perp \{\tilde{A}Z + \tilde{U}, P^{-1}\tilde{U}\} \implies \tilde{V} \perp\!\!\!\perp \{\tilde{D}, U\},$$

By Lemma C.1, we also have that:

$$\text{Law}(q_2^{-1}(d, \tilde{V}) \mid \tilde{D} = d, U) = \text{Law}(q_2^{-1}(d + \alpha u, \tilde{V}) \mid \tilde{D} = d, U)$$
$$\implies \text{Law}(q_2^{-1}(\tilde{D}, \tilde{V}) \mid \tilde{D}, U) = \text{Law}(q_2^{-1}(\tilde{D} + \alpha u, \tilde{V}) \mid \tilde{D}, U)$$

which by the definition of $V$ and $V_\alpha$ is equivalent to:;

$$\text{Law}(V \mid \tilde{D}, U) = \text{Law}(V_{\alpha u} \mid \tilde{D}, U) \implies \text{Law}(V \mid U) = \text{Law}(V_{\alpha u} \mid U)$$

By the outcome structural equation

$$Y = h(D) + \eta(U, V, \epsilon)$$

we have that:

$$Y_{\alpha u} = h(D + \alpha P^{-1}u) + \eta(U, V_{\alpha u}, \epsilon_Y)$$

and that:

$$\mathbb{E}[Y_{\alpha u} - Y] = \mathbb{E}[h(D + \alpha P^{-1}u) - h(D)] + \mathbb{E}[\eta(U, V_{\alpha u}, \epsilon) - \eta(U, V, \epsilon)]$$

Since $\epsilon \perp\!\!\!\perp \{Z, U, V\}$ and since $V_{\alpha u}$ is a measurable function of these random variables, we have that $\epsilon \perp\!\!\!\perp \{V_{\alpha u}, V, U\}$. Letting $\tilde{\eta}(u, v) = \mathbb{E}_\epsilon[\eta(u, v, \epsilon)]$, we can write:

$$\mathbb{E}[Y_{\alpha u} - Y] = \mathbb{E}[h(D + \alpha P^{-1}u) - h(D)] + \mathbb{E}[\tilde{\eta}(U, V_{\alpha u}) - \tilde{\eta}(U, V)]$$
$$= \mathbb{E}[h(D + \alpha P^{-1}u) - h(D)] + \mathbb{E}[\mathbb{E}[\tilde{\eta}(U, V_{\alpha u}) - \tilde{\eta}(U, V) \mid U]]$$
$$= \mathbb{E}[h(D + \alpha P^{-1}u) - h(D)] \qquad (\text{Law}(V \mid U) = \text{Law}(V_{\alpha u} \mid U))$$

By a first-order Taylor expansion and since $h$ is twice differentiable with bounded first and second derivatives:

$$\mathbb{E}[Y_{\alpha u} - Y] = \mathbb{E}[\alpha \nabla_D h(D)^\top P^{-1}u] + O(\alpha^2) = \alpha \|(P^{-1})^\top \mathbb{E}[\nabla_D h(D)]\| + O(\alpha^2)$$

$\square$

## C.3 AUXILIARY LEMMAS

**Lemma C.1.** *Suppose the assumptions of Theorem 4.5 hold, and additionally impose the following constraints on the learned functions $\tilde{e}, \tilde{f}, \tilde{A}$ that minimize the objective in Equation 10:*

- $\tilde{D} \perp\!\!\!\perp \tilde{V}$

- $\tilde{e}$ *is an invertible function when restricted to inputs in the image of $f$ and $\tilde{f} \circ \tilde{e}(x) = x$ for all $x \in Im(f)$.*

*Let $q \triangleq \tilde{e} \circ f$ and $q^{-1} \triangleq e \circ \tilde{f}$. Let $q_2$ denote the $\tilde{V}$-component of the output of $q$ and $q_2^{-1}$ the V-component of the output of $q^{-1}$. Then we have that $(\tilde{D}, \tilde{V}) = q(D, V)$ and $(D, V) = q^{-1}(\tilde{D}, \tilde{V})$, almost surely. Moreover, it must also hold with probability 1 that:*

- *$\tilde{V} = q_2(D, V) = (\tilde{e} \circ f)_2(D, V)^3$ with the property that for all $d, d' \in \mathcal{D}$:*
$$Law(q_2(d, V)) = Law(q_2(d', V)).$$

- *$V = q_2^{-1}(D, V) = (e \circ \tilde{f})_2(\tilde{D}, \tilde{V})$ with the property that for all $d, d' \in \tilde{\mathcal{D}} \triangleq \{Pd : d \in \mathcal{D}\}$:*
$$Law(q_2^{-1}(d, \tilde{V})) = Law(q_2^{-1}(d', \tilde{V})).$$

*Proof of Lemma C.1.* In this proof, we argue about the properties of the second part of the function $q$. Note that since $\tilde{D} \perp\!\!\!\perp \tilde{V}$ and since $\tilde{D} = PD$, for some invertible $P$, with probability 1, we have that $\tilde{V} \perp\!\!\!\perp D$. Thus:
$$q_2(D, V) \equiv \tilde{V} \perp\!\!\!\perp D$$
Since, $D \perp\!\!\!\perp V$, this implies that $Law(q_2(d, V)) = Law(q_2(d', V))$ for all $d, d' \in \mathcal{D}$.

Since $\tilde{e}$ is a bijection when restricted to inputs that are outputs of $f$ and since $f$ is an injection, we have that $\tilde{e} \circ f$ is an injection. Thus there exists a well-defined inverse function $q^{-1} = e \circ \tilde{f}$, such that $D, V = q^{-1}(D, V)$. Let $q_2^{-1}$ be the $V$ component of its output. Since $D \perp\!\!\!\perp V$ and $\tilde{D} = PD$, we have that:
$$V \equiv q_2^{-1}(\tilde{D}, \tilde{V}) \perp\!\!\!\perp \tilde{D}$$
Since $\tilde{D} \perp\!\!\!\perp \tilde{V}$, this implies that $Law(q_2^{-1}(d, \tilde{V})) = Law(q_2^{-1}(d', \tilde{V}))$ for all $d, d' \in \tilde{\mathcal{D}}$ $\qquad \square$

**Lemma C.2.** *Suppose $A$ and $B$ are $r \times k$ matrices with full row rank. If $A = AB^+B$, then $rowspan(A) = rowspan(B)$ and $AB^+$ is invertible.*

*Proof.* Consider the thin SVDs of $B = U_B \Sigma_B V_B^\top$ and $A = U_A \Sigma_A V_A^\top$. Then
$$B^+ B = V_B \Sigma_B^{-1} U_B^\top U_B \Sigma_B V_B^\top = V_B V_B^\top$$
is the projection onto the row space of $B$. Then, we have:
$$A = AB^+B \quad \Leftrightarrow \quad AV_B V_B^\top = A \tag{13}$$

First, we prove by contradiction that $rowspan(A) = rowspan(B)$. Suppose $x \in rowspan(A) = span(V_A)$, but $x \notin rowspan(B) = span(V_B)$. Let $V_B^\perp$ denote an orthogonal completion of $V_B$, then
$$x \notin span(V_B) \Rightarrow x = V_B V_B^\top x + V_B^\perp (V_B^\perp)^\top x$$
where $V_B^\perp (V_B^\perp)^\top x \neq 0$, which implies $(V_B^\perp)^\top x \neq 0$ as $V_B^\perp$ is orthogonal. Hence, we have the following:
$$\|x\|^2 = x^\top x = x^\top V_B V_B^\top x + x^\top V_B^\perp (V_B^\perp)^\top x = \|V_B^\top x\|^2 + \|(V_B^\perp)^\top x\|^2 > \|V_B^\top x\|^2$$
However, we also have that $AV_B V_B^\top x - Ax = 0$, which implies $u \triangleq V_B V_B^\top x - x \in$ null-space$(A) = span(V_A^\perp)$. Thus, it should be orthogonal with $x \in span(V_A)$.
$$0 = x^\top u = x^\top (V_B V_B^\top x - x) = \|V_B^\top x\|^2 - \|x\|^2 \neq 0$$
This yields a contradiction! Thus, $rowspan(A) \subseteq rowspan(B)$. Since, both matrices have full row rank, then $A$ and $B$ have the same row space.

Now we show that $AB^+$ is invertible:
$$AB^+ = U_A \Sigma_A V_A^\top V_B \Sigma_B^{-1} U_B^\top$$
Since $A, B$ are full row rank, $U_A, U_B, \Sigma_A, \Sigma_B$ are $r \times r$ invertible matrices. So it suffices to show that $V_A^\top V_B$ is invertible. Since $span(V_A) = span(V_B)$, there exists an invertible change-of-basis matrix $P$ such that
$$V_B = V_A P \Rightarrow V_A^\top V_B = V_A^\top V_A P = I_r P = P \Rightarrow AB^+ \text{ is invertible.}$$

$\qquad \square$

---

³With $(\tilde{e} \circ f)_2$ we denote the $V$ component of the output of the function $\tilde{e} \circ f_0$.

**Lemma C.3.** *If $U \perp\!\!\!\perp V \perp\!\!\!\perp Z$ (jointly independent), then $V \perp\!\!\!\perp Z \mid f(Z, U)$, for any measurable function $U$.*

Let $W = f(Z, U)$. Then:

$$
\begin{aligned}
p(v \mid w, z) &= \int_u p(v \mid w, z, u) p(u \mid w, z) du \\
&= \int_u p(v \mid z, u) p(u \mid w, z) du \\
&= \int_u p(v) p(u \mid w, z) du \qquad\qquad (V \perp\!\!\!\perp Z \perp\!\!\!\perp U) \\
&= p(v) \int_u p(u \mid w, z) du \\
&= p(v)
\end{aligned}
$$

**Lemma C.4** (Sufficient Conditions for Bounded Completeness). *Consider $D = A \cdot Z + U$, $U \perp\!\!\!\perp Z$. $D$ is bounded complete for $Z$ if the following holds:*

- *The measure of $AZ$ is continuous and is supported on $\mathbb{R}^r$.*

- *The density of $U$ is continuous.*

- *The characteristic function of the distribution of $U$ is infinitely often differentiable and does not vanish on the real line.*

*Proof of Lemma C.4.* This result follows as a Corollary of Theorem 2.1 in D'Haultfoeuille, where we consider the special case of linear mappings from $Z$ to $D$. □

Another set of sufficient conditions is given by Saengkyongam et al., requiring only that $AZ$ be supported on an open subset of $\mathbb{R}^r$ rather than on the full space.

**Lemma C.5** (Sufficient Conditions for Bounded Completeness 2). *Consider $D = A \cdot Z + U$, $U \perp\!\!\!\perp Z$. $D$ is bounded complete for $Z$ if the following holds:*

- *The measure of $AZ$ is continuous and is supported on an open subset of $\mathbb{R}^r$.*

- *The density of $U$ is analytic.*

- *The characteristic function of the distribution of $U$ has no zeros.*

*Proof of Lemma C.5.* This result follows part of the proof of Theorem 6 in Saengkyongam et al.. □

## D  NON-LINEAR SETTING WITH COVARIATES

We will expand the nonlinear setting to include covariates. Specifically, the data generating process follows:

$$
\begin{aligned}
D &= A_W \cdot Z + U, & U &\perp\!\!\!\perp Z \mid W \\
X &= f_W(D, V), & V &\perp\!\!\!\perp Z \mid W \qquad\qquad (14) \\
Y &= h_W(D) + \eta(U, V, \epsilon), & \epsilon &\perp\!\!\!\perp Z \mid W
\end{aligned}
$$

Where $W$ is dimension $p$ observed covariates or exogenous variables. Denote the domain of $W$ be $\mathcal{W}$. All of $Z, U, V, \epsilon$ can be dependent on $W$.

The assumption required mostly follows the nonlinear setting, with additional conditioning on $W$.

**Assumption D.1** (Invertible Encoding). For every $w \in \mathcal{W}$, the function $f_w$ is invertible. Given $W = w$, write the encoding function $e_w(X) = f_w^{-1}(X) = (D, V)$, i.e. there is a one-to-one correspondence between the high-dimensional treatment $X$ and the characteristics $(D, V)$ that describe the treatment given covariates are $W = w$.

**Assumption D.2** (Full-Rank Latents). Assume that the matrix $A_w$ has full row-rank and $\mathbb{E}[ZZ^\top \mid W = w] \succ 0$ for every $w \in \mathcal{W}$.

**Assumption D.3** (Conditional Joint Independence). Assume that $Z \perp\!\!\!\perp U \perp\!\!\!\perp V \mid W$ (conditionally jointly independent).

**Assumption D.4** (Differentiable Decoding Function). For every $w \in \mathcal{W}$, $f_w$ is a differentiable function with uniformly bounded derivatives.

**Assumption D.5.** $\mathbb{E}[Z \mid W] = 0$, $\mathbb{E}[U \mid W] = 0$, $\mathbb{E}[\eta(U, V, \epsilon) \mid W] = 0$ and the support of $Z$, $\mathcal{Z}$, is an open subset of $\mathbb{R}^k$.

We discussed the sufficient conditions for bounded completeness in D.6, which is a direct result of Theorem 2.1 in D'Haultfoeuille.

**Assumption D.6** (Bounded Completeness). $D$ is bounded complete for Z, that is, for all bounded real functions $h$, we have that:

$$\mathbb{E}[h_W(D) \mid Z, W = w] = 0 \quad \text{a.s.} \quad \Rightarrow \quad h_w(D) = 0 \quad \text{a.s.}$$

**Theorem D.7.** *Suppose that the data generating process follows the SEM described in Equation 14, and satisfies Assumptions D.1 & D.2 & D.3 & D.4 & D.5 & D.6. Let $(\tilde{D}_W, \tilde{V}_W) := (\tilde{e}_{W,D}(X), \tilde{e}_{W,V}(X)) = \tilde{e}_W(X)$ denote the learned representations. Consider encoder-decoder pairs with perfect reconstruction, i.e. $X = \tilde{f}_W \circ \tilde{e}_W(X)$. Then, for the solution $\tilde{e}_W, \tilde{f}_W$, and full row rank matrix $\tilde{A}_W$ that minimizes the objective function*

$$\mathbb{E}[\|\tilde{e}_{W,D}(X) - \tilde{A}_W Z\|^2] \tag{15}$$

*subject to the following constraints:*

- *$\tilde{e}_w$ is a differentiable function with uniformly bounded derivatives for all $w \in \mathcal{W}$.*

- *$\tilde{A}_w$ has full row rank for all $w \in \mathcal{W}$.*

- *$\tilde{D}_w = \tilde{A}_w Z + \tilde{U}_w$ for all $w \in \mathcal{W}$ with $\tilde{U}_W \perp\!\!\!\perp Z \mid W$ and $\mathbb{E}[\tilde{U}_W \mid W] = 0$.*

*we have that, with probability 1, $\tilde{D}_W = P_W D$ and $\tilde{U}_W = P_W U$, for $P_W = \tilde{A}_W A_W^+$. Moreover, the matrix $P_W$ is invertible.*

**Lemma D.8.** *Suppose the assumptions of Theorem D.7 hold, and additionally impose the following constraints on the learned functions $\tilde{e}_W, \tilde{f}_W, \tilde{A}_W$ that minimize the objective in Equation 15:*

- *$\tilde{D}_W \perp\!\!\!\perp \tilde{V}_W \mid W$*

- *$\tilde{e}_w$ is an invertible function when restricted to inputs in the image of $f_w$ and $\tilde{f}_w \circ \tilde{e}_w(x) = x$ for all $x \in Im(f)$ and corresponding $w \in \mathcal{W}$.*

*Let $q_W \triangleq \tilde{e}_W \circ f_W$ and $q_W^{-1} \triangleq e_W \circ \tilde{f}_W$. Let $q_{W,1}$ denote the $\tilde{D}$-component of the output of $q_W$ and $q_{W,1}^{-1}$ the $D$-component of the output of $q_W^{-1}$.*

*Then we have that $(\tilde{D}_W, \tilde{V}_W) = q_W(D, V)$ and $(D, V) = q_W^{-1}(\tilde{D}_W, \tilde{V}_W)$, almost surely. Moreover, it must also hold with probability 1 that:*

- *$\tilde{V}_W = q_{W,1}(D, V) = (\tilde{e}_W \circ f_W)_2(D, V)$ with the property that for given $w$ and all $d, d' \in \mathcal{D}_w := support(D \mid W = w)$:*

$$Law(q_{W,2}(d, V) \mid W = w) = Law(q_{W,2}(d', V) \mid W = w).$$

- *$V = q_{W,2}^{-1}(D, V) = (e_W \circ \tilde{f}_W)_2(\tilde{D}_W, \tilde{V}_W)$ with the property that for given $w$ and all $d, d' \in \tilde{\mathcal{D}}_w := \{P_w d : d \in \mathcal{D}_w\}$:*

$$Law(q_{w,2}^{-1}(d, \tilde{V}_W) \mid W = w) = Law(q_{w,2}^{-1}(d', \tilde{V}_W) \mid W = w).$$

*Remark* D.9. Note that the assumptions that $\mathbb{E}[Z \mid W] = 0$ and $D = A_W Z + U$ are without loss of generality as we can always pre-process Z by centering it. In practice, the assumption that $\tilde{D}_W = \tilde{A}_W Z + \tilde{U}$, with $\tilde{U}_W \perp\!\!\!\perp Z \mid W$ and $\mathbb{E}[\tilde{U}_W \mid W] = 0$ can be achieved by minimizing a square loss with an covariate-specific intercept $C_W$, i.e.

$$\min_{e,f,A,C:e,f\text{invertible},e\circ f=\text{identity}} \mathbb{E}[\|e_{W,D}(X) - A_W Z - C_W\|^2]$$

and then defining $\tilde{D}_W = \tilde{e}_D(X) - C_W \triangleq e_{W,D}(X)$, $\tilde{f}_W = f_W + C_W$.

The intervention is carried out in close accordance with Algorithm 1. In particular, we will run an IV analysis, with $Z$ as the instrument, $\tilde{D}$ as the treatment, $W$ as the covariates, and $Y$ as the outcome, to estimate a causal model in representation space by finding a solution to the conditional moment restrictions:

$$\mathbb{E}[Y - \tilde{h}_W(\tilde{D}_W) \mid Z, W = w] = 0 \tag{16}$$

Note that since $\tilde{D}_W = P_W D$ and since $\mathbb{E}[Y \mid Z, W = w] = \mathbb{E}[h_W(D) \mid Z, W = w]$, we have by the completeness assumption that:

$$\mathbb{E}[h_W(D) - \tilde{h}_W(P_W D) \mid Z, W = w] = 0 \Rightarrow h_w(D) = \tilde{h}_w(P_w D) \text{ a.s.}$$
$$\Rightarrow h_w(P_w^{-1}\tilde{D}) = \tilde{h}_w(\tilde{D}_w) \text{ a.s.}$$

**Theorem D.10.** *Assume that:*

$$Y = h_W(D) + \eta(U, V, \epsilon), \quad \epsilon \perp\!\!\!\perp \{Z, U, V\} \mid W$$

*and that $h_W$ is twice differentiable with a bounded second derivative. Let $\tilde{e}_W, \tilde{f}_W, \tilde{A}_W$ be an optimal solution as prescribed in Lemma D.8 with the extra constraint that:*

$$\tilde{U}_W \perp\!\!\!\perp \tilde{V}_W \perp\!\!\!\perp Z \mid W \qquad \text{(conditional joint independence)}$$

*and assume that the assumptions of Lemma D.8 are satisfied. Furthermore, assume that the variable $D$ has full support in $\mathbb{R}^r$ conditional on $W = w$, for all $w \in \mathbb{R}^p$. Then setting $u_w = \tilde{\theta}_w / \|\tilde{\theta}_w\|$, in Algorithm 1, with $\tilde{\theta}_w = \mathbb{E}[\nabla_{\tilde{D}_W} \tilde{h}_W(\tilde{D}_W) \mid W = w]$ and $\tilde{h}$ the solution to the conditional moment restriction problem in Equation (16), we have that:*

$$\mathbb{E}[Y_{\alpha u} - Y \mid W] = \alpha \|(P_W^{-1})^\top \mathbb{E}[\nabla_D h_W(D) \mid W]\| + O(\alpha^2)$$

Hence, for small enough step size $\alpha$, the identified intervention will achieve a positive improvement on the outcome (assuming that $\mathbb{E}[\nabla_D h_W(D) \mid W] \neq 0$).

**Instrument Regularized Auto-Encoder** To achieve the positive improvement as described in Theorem D.10, then we need to incorporate loss components that are minimized only when

- $e_W, f_W$ reconstruct the input $X$,
- $e_{W,D}(X)$ is predicted linearly by $Z$ given $W$ with a full rank matrix $A_W$,
- The residual of this regression $e_{W,D}(X) - A_W Z - C_W$, which approximates $U$, is independent of $Z$ given $W$,
- $Z$ is independent of $e_{W,V}(X)$ given $W$,
- Conditioning on $W$, we have that:

$$e_{W,D} \perp\!\!\!\perp e_{W,V}(X) \mid W$$
$$e_{W,D}(X) - A_W Z - C_W \perp\!\!\!\perp e_{W,V}(X) \mid W$$
$$(e_{W,D}(X) - A_W Z - C_W) \perp\!\!\!\perp e_{W,V}(X) \perp\!\!\!\perp Z \mid W$$

While we do not explicitly enforce $\tilde{A}_W$ to be full row rank, we expect this to be satisfied due to the reconstruction loss and the condition that $\tilde{D}_W \perp\!\!\!\perp \tilde{V}_W \mid W$. Moreover, note that instead of only conditional joint independence of $Z, \tilde{U}_W, \tilde{V}_W$ we also explicitly enforce pairwise independencies for computational reasons.

We introduce the instrument-regularized auto-encoder loss, which incorporates all these elements:

$$\min_{e,f,A,C} \mathbb{E}\left[\|X - f_W \circ e_W(X)\|^2\right] + \lambda\mathbb{E}\left[\|e_{W,D}(X) - A_W Z - C_W\|^2\right]$$
$$+ \mu_1 \mathcal{R}_W(e_{W,D}(X) - A_W Z - C_W, Z)$$
$$+ \mu_2 \mathcal{R}_W(Z, e_{W,V}(X)) \qquad\qquad \text{(IRAE with covar)}$$
$$+ \mu_3\big(c_1 \mathcal{R}_W(e_{W,D}(X), e_{W,V}(X))$$
$$+ c_2 \mathcal{R}_W(e_{W,D}(X) - A_W Z - C_W, e_{W,V}(X))$$
$$+ c_3 \mathcal{R}_W(e_{W,D}(X) - A_W Z - C_W, Z, e_{W,V}(X))\big)$$

$\mathcal{R}_W(A,B)$ or $\mathcal{R}_W(A,B,C)$, denotes any regularizer that can be evaluated on a set of $n$ samples and which takes small values when the random variables $A, B$ or $A, B, C$ are jointly independent condition on $W$. For example, the procedure introduced in Pogodin et al. (2024). Computing conditional independence statistics is typically more challenging than evaluating unconditional independence, and may introduce additional computational complexity.

# E    PROOF OF NON-LINEAR IDENTIFICATION WITH COVARIATES

## E.1    AUXILIARY LEMMAS

**Lemma E.1.** *If* $U \perp\!\!\!\perp V \perp\!\!\!\perp Z \mid W$ *(conditionally jointly independent), then* $V \perp\!\!\!\perp Z \mid f(Z,U,W), W$, *for any measurable function* $U$.

Let $S = f(Z, U, W)$. Then:

$$p(v \mid s, z, w) = \int_u p(v \mid s, z, u, w) p(u \mid s, z, w) du$$
$$= \int_u p(v \mid z, u, w) p(u \mid s, z, w) du$$
$$= \int_u p(v \mid w) p(u \mid s, z, w) du \qquad (V \perp\!\!\!\perp Z \perp\!\!\!\perp U \mid W)$$
$$= p(v \mid w) \int_u p(u \mid s, z, w) du$$
$$= p(v \mid w)$$

**Lemma E.2** (Sufficient Conditions for Bounded Completeness). *Consider* $D = A_W \cdot Z + U$, $\quad U \perp\!\!\!\perp Z \mid W$. *$D$ is bounded complete for $Z$ if the following holds:*

- $Z \perp\!\!\!\perp U \mid W$

- *The measure of $A_W Z$ is continuous and is supported on $\mathbb{R}^r$.*

- *The conditional density of $U \mid W = w$ is continuous for all $w \in \mathcal{W}$.*

- *The conditional characteristic function of the distribution of $U \mid W = w$ is infinitely often differentiable and does not vanish on the real line for all $w \in \mathcal{W}$.*

*Proof of Lemma E.2.* This result follows as a Corollary of Theorem 2.1 in D'Haultfoeuille, where we consider the special case of linear mappings from $Z$ to $D$. $\qquad\square$

## E.2    THEOREM D.7 AND LEMMA D.8

*Proof of Theorem 4.5.* By definition of $(\tilde{D}_W, \tilde{V}_W)$, and given $W = w$, we have:

$$(\tilde{D}_W, \tilde{V}_W) = \tilde{e}_w(X) = \tilde{e}_w \circ f_w(D, V) =: q_w(D, V)$$

Denote with $q_{w,1}(D,V)$ the $\tilde{D}_w$ component of the output of $q$ and $q_{w,2}$ the $\tilde{V}_w$ component.

Since we have that $\tilde{D}_w = \tilde{A}_w Z + \tilde{U}_w$, with $\tilde{U}_W \perp\!\!\!\perp Z \mid W$ and $\mathbb{E}[\tilde{U}_W \mid W] = 0$, we can write:

$$\mathbb{E}[\tilde{D}_W \mid Z = z, W = w] = \mathbb{E}[\tilde{A}_w Z + \tilde{U}_W \mid Z = z, W = w] = \tilde{A}_w z$$

Moreover:

$$
\begin{aligned}
\mathbb{E}[\tilde{D}_W \mid Z = z, W = w] &= \mathbb{E}[q_{w,1}(D, V) \mid Z = z, W = w] \\
&= \mathbb{E}[q_{w,1}(A_w z + U, V) \mid Z = z, W = w] \\
&= \mathbb{E}[q_{w,1}(A_w z + U, V) \mid W = w] && (Z \perp\!\!\!\perp \{U, V\} \mid W) \\
&= \mathbb{E}_U[\mathbb{E}_V[q_{w,1}(A_w z + U, V) \mid W = w] \mid W = w] && (U \perp\!\!\!\perp V \mid W) \\
&= \mathbb{E}_U[\tilde{q}_{w,1}(A_w z + U) \mid W = w] && (\tilde{q}_{w,1}(d) \triangleq \mathbb{E}_V[q_{w,1}(d, V) \mid W = w]) \\
&= \mathbb{E}[\tilde{q}_{w,1}(A_w z + U) \mid W = w]
\end{aligned}
$$

Thus we can conclude that:

$$\tilde{A}_w z = \mathbb{E}[\tilde{D}_W \mid Z = z, W = w] = \mathbb{E}[\tilde{q}_{w,1}(A_w z + U) \mid W = w]$$

Since this holds for all $z \in \mathcal{Z}$ and since $\mathcal{Z}$ is an open set, we can take the derivative with respect to $z$, to derive:

$$\forall z \in \mathcal{Z} : \tilde{A}_w = \partial_z \mathbb{E}[\tilde{q}_{w,1}(A_w z + U) \mid W = w]$$

Since $q_{w,1}$ is continuously differentiable with bounded derivatives, the same holds for $\tilde{q}_{w,1}$ and therefore we can exchange the order of differentiation and expectation:

$$\tilde{A}_w = \mathbb{E}[\partial_z \tilde{q}_{w,1}(A_w z + U) \mid W = w]$$

Letting $\tilde{q}_{w,1}^{(1)}$ denote the Jacobian of the function $\tilde{q}_{w,1}(d)$, we can write by the chain rule:

$$
\begin{aligned}
\tilde{A}_w &= \mathbb{E}[\tilde{q}_{w,1}^{(1)}(A_w z + U) A_w \mid W = w] \\
&= \mathbb{E}[\tilde{q}_{w,1}^{(1)}(A_w z + U) \mid W = w] A_w \\
&= \mathbb{E}[\tilde{q}_{w,1}^{(1)}(A_w z + U) \mid Z = z, W = w] A_w && (Z \perp\!\!\!\perp U \mid W) \\
&= \mathbb{E}[\tilde{q}_{w,1}^{(1)}(A_W Z + U) \mid Z = z, W = w] A_w \\
&= \mathbb{E}[\tilde{q}_{w,1}^{(1)}(D) \mid Z = z, W = w] A_w
\end{aligned}
$$

Since $A_w$ is full row rank, we have that $A_w A_w^+$ is invertible. Thus we can write:

$$\tilde{A}_w A_w^+ = \mathbb{E}[\tilde{q}_w^{(1)}(D) \mid Z = z, W = w]$$

or equivalently:

$$\forall z \in \mathcal{Z} : \mathbb{E}[\tilde{q}_{w,1}^{(1)}(D) - \tilde{A}_w A_w^+ \mid Z = z, W = w] = 0$$

By the bounded completeness assumption and since both $\tilde{A}_w A_w^+$ and $\tilde{q}_{w,1}^{(1)}$ are bounded, the latter implies that:

$$\forall d \in \mathcal{D} : \tilde{q}_{w,1}^{(1)}(d) = \tilde{A}_w A_w^+$$

or equivalently that:

$$\tilde{q}_{w,1}(d) = \tilde{A}_w A_w^+ d + \tilde{\nu}$$

for some constant vector $\nu$. Moreover,

$$
\begin{aligned}
\mathbb{E}[\tilde{D}_W \mid W = w] &= \mathbb{E}[\tilde{q}_{w,1}(D)] \\
&= \tilde{A}_w A_w^+ \mathbb{E}[D \mid W = w] + \tilde{\nu} \\
&= \tilde{\nu}
\end{aligned}
$$

But we also have $\mathbb{E}[\tilde{D}_W \mid W = w] = 0$. Hence, we have that $\tilde{\nu} = 0$. Thus:

$$\forall d \in \mathcal{D} : \tilde{q}_{w,1}(d) = \tilde{A}_w A_w^+ d$$

Next, we argue that $\tilde{A}_w A_w^+$ is an invertible matrix. Note that:

$$\mathbb{E}[\tilde{D}_W Z^\top \mid W = w] = \mathbb{E}[(\tilde{A}_w Z + \tilde{U}_W) Z^\top \mid W = w]$$
$$= \tilde{A}_w \mathbb{E}[Z Z^\top \mid W = w] \quad (\tilde{U}_W \perp\!\!\!\perp Z \mid W, \mathbb{E}[Z \mid W] = 0, \mathbb{E}[\tilde{U}_W \mid W] = 0)$$

Moreover:

$$\mathbb{E}[\tilde{D}_W Z^\top \mid W = w] = \mathbb{E}[q_{w,1}(D, V) Z^\top \mid W = w]$$
$$= \mathbb{E}[q_{w,1}(A_w Z + U, V) Z^\top \mid W = w]$$
$$= \mathbb{E}[\mathbb{E}[q_{w,1}(A_w Z + U, V) \mid Z, U, W = w] Z^\top \mid W = w]$$
$$= \mathbb{E}[\tilde{q}_{w,1}(A_w Z + U) Z^\top \mid W = w] \qquad (Z \perp\!\!\!\perp U \perp\!\!\!\perp V \mid W)$$
$$= \mathbb{E}[\tilde{q}_{w,1}(D) Z^\top \mid W = w]$$
$$= \mathbb{E}[(\tilde{A}_w A_w^+ D) Z^\top \mid W = w]$$
$$= \tilde{A}_w A_w^+ \mathbb{E}[D Z^\top \mid W = w]$$
$$= \tilde{A}_w A_w^+ \mathbb{E}[(A_w Z + U) Z^\top \mid W = w]$$
$$= \tilde{A}_w A_w^+ A_w \mathbb{E}[Z Z^\top \mid W = w]$$
$$\qquad (Z \perp\!\!\!\perp U \mid W, \mathbb{E}[Z \mid W] = 0, \mathbb{E}[U \mid W] = 0)$$

Thus we have concluded that:

$$\tilde{A}_w \mathbb{E}[Z Z^\top \mid W = w] = \mathbb{E}[\tilde{D}_W Z^\top \mid W = w] = \tilde{A}_w A_w^+ A_w \mathbb{E}[Z Z^\top \mid W = w]$$

Since $\mathbb{E}[Z Z^\top \mid W = w]$ is assumed to be invertible, the latter implies that:

$$\tilde{A}_w = \tilde{A}_w A_w^+ A_w$$

By Lemma C.2 in Appendix C.3, since $\tilde{A}_w$ and $A_w$ have full row rank, the row span of $\tilde{A}_w$ is equal to the row span of $A_w$ and the matrix $\tilde{A}_w A_w^+$ is invertible.

We have thus concluded that:

$$\forall d \in \mathcal{D} : \tilde{q}_1(d) = \tilde{A}_w A_w^+ d$$

and $\tilde{A}_w A_w^+$ is invertible.

Consider any solution with perfect encoder-decoder pair $(\tilde{e}_W, \tilde{f}_W)$, and $\tilde{A}_W$ that satisfies the conditions of the theorem and minimizes the objective function:

$$\mathbb{E}[\|\tilde{e}_{W,D}(X) - \tilde{A} Z\|^2] = \mathbb{E}[\|\tilde{D}_W - \tilde{A}_W Z\|^2]$$

Given $W = w$, for any feasible solution, we can decompose this objective into two components by centering around

$$\mu_{\tilde{A}_w}(d) \triangleq \tilde{A}_w A_w^+ d$$

i.e.:

$$\mathbb{E}[\|\tilde{D}_W - \tilde{A}_W Z\|^2] = \mathbb{E}[\|\tilde{D}_W - \mu_{\tilde{A}_W}(D) + \mu_{\tilde{A}_W}(D) - \tilde{A}_W Z\|^2]$$
$$= \mathbb{E}[\|\tilde{D}_W - \mu_{\tilde{A}_W}(D)\|^2 + \|\mu_{\tilde{A}_W}(D) - \tilde{A}_W Z\|^2]$$
$$+ 2\mathbb{E}[(\tilde{D}_W - \mu_{\tilde{A}_W}(D))^\top (\mu_{\tilde{A}_W}(D) - \tilde{A}_W Z)]$$

Consider the inner product term. Since we have that:

$$\mathbb{E}[\tilde{D}_W - \mu_{\tilde{A}_W}(D) \mid D, Z, W] = \mathbb{E}[q_{W,1}(D, V) - \tilde{q}_{W,1}(D) \mid D, Z, W]$$
$$= \mathbb{E}[q_{W,1}(D, V) \mid D, Z, W] - \tilde{q}_{W,1}(D)$$
$$= \mathbb{E}[q_{W,1}(D, V) \mid D, Z, W] - \mathbb{E}[q_{W,1}(D, V) \mid D, W]$$

Since $Z \perp\!\!\!\perp U \perp\!\!\!\perp V \mid W$, we have by Lemma E.1 that $Z \perp\!\!\!\perp V \mid \{\mathbb{1}\{A_w Z + U = d\}, \mathbb{1}\{W = w\}\}$:

$$\begin{aligned}
\mathbb{E}[q_{W,1}(D, V) \mid D = d, Z, W = w] &= \mathbb{E}[q_{W,1}(d, V) \mid D = d, Z, W = w] \\
&= \mathbb{E}[q_{W,1}(d, V) \mid D = d, W = w] \\
&= \mathbb{E}[q_{W,1}(D, V) \mid D = d, W = w]
\end{aligned}$$

Thus:

$$\mathbb{E}[\tilde{D}_W - \mu_{\tilde{A}_W}(D) \mid D, Z, W] = 0$$

From this we conclude that for any feasible solution $\tilde{e}_W, \tilde{f}_W, \tilde{A}_W$, we have that the objective can be decomposed as:

$$\begin{aligned}
\mathbb{E}[\|\tilde{e}_{W,D}(X) - \tilde{A}_W Z\|^2] &= \mathbb{E}[\|\tilde{D}_W - \mu_{\tilde{A}_W}(D)\|^2] + \mathbb{E}[\|\mu_{\tilde{A}_W}(D) - \tilde{A}_W Z\|^2] \\
&= \mathbb{E}[\|q_{W,1}(D, V) - \mu_{\tilde{A}_W}(D)\|^2] + \mathbb{E}[\|\mu_{\tilde{A}_W}(D) - \tilde{A}_W Z\|^2]
\end{aligned}$$

Suppose that with positive probability, we have that $q_{W,1}(D, V) \neq \mu_{\tilde{A}_W}(D) = \tilde{A}_W A_W^+ D$. Then we have that:

$$\mathbb{E}[\|q_{W,1}(D, V) - \mu_{\tilde{A}_W}(D)\|^2] > 0$$

In this case, we will provide an alternative feasible solution, which achieves smaller objective than $\tilde{e}_W, \tilde{f}_W, \tilde{A}_W$. Consider the solution:

$$\begin{aligned}
\tilde{e}'_W(x) &= (\tilde{A}_W A_W^+ e_{W,D}(x), e_{W,V}(x)) \\
\tilde{f}'_W(d, v) &= f((\tilde{A}_W A_W^+)^{-1} d, v)
\end{aligned}$$

Note that we used the fact that for any feasible solution, we have already shown that $\tilde{A}_W A_W^+$ is invertible. Moreover, note that for this solution we have that, $\tilde{e}'_W, \tilde{f}'_W$ is invertible, since $e_W, f_W$ is invertible and $\tilde{A}_W A_W^+$ is invertible. Finally,

$$\begin{aligned}
\tilde{f}'_W \circ \tilde{e}'_W(x) &= x \\
\tilde{e}'_W(f(d, v)) &= (\tilde{A}_W A_W^+ e_{W,D}(f_W(d, v)), e_{W,V}(f_W(d, v))) = (\tilde{A}_W A_W^+ d, v)
\end{aligned}$$

Thus:

$$\begin{aligned}
\tilde{D}_W = \tilde{e}'_{W,D}(X) &= \tilde{e}'_{W,D}(f_W(D, V)) \\
&= \tilde{A}_W A_W^+ D \\
&= \tilde{A}_W A_W^+ A_W Z + \tilde{A}_W A_W^+ U \\
&= \tilde{A}_W Z + \tilde{A}_W A_W^+ U
\end{aligned}$$

Where we used the fact that we have already shown (in the proof of Theorem D.7) that $\tilde{A}_w A_w^+ A_w = \tilde{A}_w$ for all $w \in \mathcal{W}$. Thus, we also have that:

$$\tilde{D}_W = \tilde{A}_W Z + \tilde{U}_W$$

where $\tilde{U}_W = \tilde{A}_W A_W^+ U$ and satisfies $\tilde{U}_W \perp\!\!\!\perp Z \mid W$ and $\mathbb{E}[\tilde{U}_W \mid W] = 0$.

Therefore, this new solution is a feasible solution. Moreover, since under this solution we have that $\tilde{D}_W = \tilde{A}_W A_W^+ D = \mu_{\tilde{A}_W}(D)$, the first part of the objective vanishes and the objective takes the value:

$$\mathbb{E}[\|\mu_{\tilde{A}_W}(D) - \tilde{A}_W Z\|^2] < \mathbb{E}[\|q_{W,1}(D, V) - \mu_{\tilde{A}_W}(D)\|^2] + \mathbb{E}[\|\mu_{\tilde{A}_W}(D) - \tilde{A}_W Z\|^2]$$

contradicting the optimality of the original solution.

Thus we have derived that for any optimal feasible solution, it must hold that with probability 1:

$$\tilde{D}_W = q_{W,1}(D, V) = \mu_{\tilde{A}_W}(D) = \tilde{A}_W A_W^+ D \tag{17}$$

with $\tilde{A}_W A_W^+$ an invertible matrix. Moreover, this implies that $\tilde{U}_W = \tilde{D}_W - \tilde{A}_W Z = \tilde{A}_W A_W^+ U$. $\quad\square$

*Proof of Lemma D.8.* In this proof, we argue about the properties of the second part of the function $q_W$. Note that since $\tilde{D}_W \perp\!\!\!\perp \tilde{V}_W \mid W$ and since $\tilde{D}_W = P_W D$, for some invertible $P_W$, with probability 1, we have that $\tilde{V}_W \perp\!\!\!\perp D \mid W$. Thus:

$$q_{w,2}(D, V) \equiv \tilde{V}_w \perp\!\!\!\perp D \mid W = w$$

Since, $D \perp\!\!\!\perp V \mid W$, this implies that $\text{Law}(q_{w,2}(d, V) \mid W = w) = \text{Law}(q_{w,2}(d', V) \mid W = w)$ for all $d, d' \in \mathcal{D}_w$.

Fixed $W = w$. Since $\tilde{e}_w$ is a bijection when restricted to inputs that are outputs of $f_w$ and since $f_w$ is an injection, we have that $\tilde{e}_w \circ f_w$ is an injection. Thus there exists a well-defined inverse function $q_w^{-1} = e_w \circ \tilde{f}_w$, such that $D, V = q_w^{-1}(D, V)$. Let $q_{w,2}^{-1}$ be the $V$ component of its output. Since $D \perp\!\!\!\perp V \mid W$ and $\tilde{D}_W = P_W D$, we have that:

$$V \equiv q_{w,2}^{-1}(\tilde{D}_w, \tilde{V}_w) \perp\!\!\!\perp \tilde{D} \mid W = w$$

Since $\tilde{D}_W \perp\!\!\!\perp \tilde{V}_W \mid W$, this implies that $\text{Law}(q_{w,2}^{-1}(d, \tilde{V}_W) \mid W = w) = \text{Law}(q_{w,2}^{-1}(d', \tilde{V}_W) \mid W = w)$ for all $d, d' \in \tilde{\mathcal{D}}_w$ $\qquad \square$

### E.3 PROOF OF POSITIVE IMPROVEMENT

*Proof of Theorem D.10.* In this proof, we show that intervention in the direction of average derivatives of $\tilde{h}_W$ guarantees positive improvement for sufficiently small $\alpha$, assuming that $h_W$ is twice differentiable. If we perform the intervention $\tilde{D}_W + \alpha u_W$, then we have by Lemma D.8 that:

$$D_{\alpha u_W}, V_{\alpha u_W} = (D + \alpha P^{-1} u_W, q_2^{-1}(\tilde{D}_W + \alpha u_W, \tilde{V}_W))$$

Since $\mathcal{D}_w = \mathbb{R}^r$ and since $P_w$ is an invertible matrix, we have that $\tilde{\mathcal{D}}_w = \mathbb{R}^r$ for every $w \in \mathcal{W}$. Thus, for all $d \in \tilde{\mathcal{D}}_W$, we also have that $d + a u_W \in \tilde{\mathcal{D}}$. By Lemma C.1, have that for all $d \in \tilde{\mathcal{D}}_w$:

$$\text{Law}(q_{W,2}^{-1}(d, \tilde{V}_W) \mid W) = \text{Law}(q_{W,2}^{-1}(d + \alpha u_W, \tilde{V}_W) \mid W)$$

By Theorem D.7, we have that, with probability 1, $\tilde{D}_W = P_W D$ and $\tilde{U}_W = P_W U$. Moreover, by assumption, we have that $\tilde{V}_W \perp\!\!\!\perp \tilde{U}_W \perp\!\!\!\perp Z \mid W$, which implies

$$\tilde{V}_W \perp\!\!\!\perp \{\tilde{A}_W Z + \tilde{U}_W, P^{-1} \tilde{U}_W\} \mid W \implies \tilde{V}_W \perp\!\!\!\perp \{\tilde{D}_W, U\} \mid W$$

we also have that:

$$\text{Law}(q_{W,2}^{-1}(d, \tilde{V}_W) \mid U, W = w, \tilde{D}_W = d) = \text{Law}(q_{W,2}^{-1}(d + \alpha u_W, \tilde{V}_W) \mid U, W = w, \tilde{D}_W = d)$$

$$\implies \text{Law}(q_{W,2}^{-1}(\tilde{D}_W, \tilde{V}_W) \mid \tilde{D}_W, U, W = w) = \text{Law}(q_{W,2}^{-1}(\tilde{D}_W + \alpha u_W, \tilde{V}_W) \mid \tilde{D}_W, U, W = w)$$

which by the definition of $V$ and $V_\alpha$ is equivalent to::

$$\text{Law}(V \mid \tilde{D}_W, U, W) = \text{Law}(V_{\alpha u} \mid \tilde{D}_W, U, W) \implies \text{Law}(V \mid U, W) = \text{Law}(V_{\alpha u} \mid U, W)$$

By the outcome structural equation

$$Y = h_W(D) + \eta(U, V, \epsilon)$$

we have that:

$$Y_{\alpha u} = h_W(D + \alpha P_W^{-1} u_W) + \eta(U, V_{\alpha u_W}, \epsilon_Y)$$

and that:

$$\mathbb{E}[Y_{\alpha u_W} - Y \mid W] = \mathbb{E}[h_W(D + \alpha P_W^{-1} u_W) - h_W(D) \mid W] + \mathbb{E}[\eta(U, V_{\alpha u_W}, \epsilon) - \eta(U, V, \epsilon) \mid W]$$

Since $\epsilon \perp\!\!\!\perp \{Z, U, V\} \mid W$ and since $V_{\alpha u_W}$ is a measurable function of these random variables, we have that $\epsilon \perp\!\!\!\perp \{V, V_{\alpha u}, U\} \mid W$. Letting $\tilde{\eta}(u, v) = \mathbb{E}_\epsilon[\eta(u, v, \epsilon) \mid W]$, we can write:

$$\mathbb{E}[Y_{\alpha u} - Y \mid W] = \mathbb{E}[h_W(D + \alpha P_W^{-1} u_W) - h_W(D) \mid W] + \mathbb{E}[\tilde{\eta}(U, V_{\alpha u_W}) - \tilde{\eta}(U, V) \mid W]$$

$$= \mathbb{E}[h_W(D + \alpha P_W^{-1} u_W) - h_W(D) \mid W] + \mathbb{E}[\mathbb{E}[\tilde{\eta}(U, V_{\alpha u}) - \tilde{\eta}(U, V) \mid U, W] \mid W]$$

$$= \mathbb{E}[h_W(D + \alpha P_W^{-1} u) - h_W(D) \mid W] \quad (\text{Law}(V \mid U, W) = \text{Law}(V_{\alpha u} \mid U, W))$$

By a first-order Taylor expansion and since $h$ is twice differentiable with bounded first and second derivatives:

$$\mathbb{E}[Y_{\alpha u} - Y \mid W] = \mathbb{E}[\alpha \nabla_D h_W(D)^\top P^{-1} u_W \mid W] + O(\alpha^2)$$

$$= \alpha \| (P_W^{-1})^\top \mathbb{E}[\nabla_D h_W(D) \mid W] \| + O(\alpha^2)$$

$\qquad \square$

# F  FURTHER DETAILS ON EXPERIMENTAL EVALUATION

## F.1  LINEAR

This section provides details of the linear experiments briefly described in Section 5 of the main paper. While the main paper presents summary statistics of average improvements and key findings, here we included detailed data generating equations and histograms of the improvements across runs.

The data are generated using the three following cases.

---

**Linear DGP 1 Independent Gaussian U and V**

Draw DGP parameters

$$A \sim \{N(0, 0.1^2)\}^{r \times k} \qquad B \sim \{N(0, 1)\}^{m \times r} \qquad \theta \sim \{N(0, 1)\}^{r \times 1}$$

Then generate $n$ samples as:

$$Z_i \sim \mathcal{N}(0, I_k) \qquad\qquad \text{(instrument)}$$

$$U_i \sim \mathcal{N}(0, 20^2 \cdot I_r) \qquad\qquad \text{(confounder 1)}$$

$$V_i \sim \mathcal{N}(0, 10^2 \cdot I_m) \qquad\qquad \text{(confounder 2)}$$

$$\eta_i(U_i, V_i) = \sum_{j=1}^{r} U_{ij} + 0.2 \cdot \varepsilon_i, \quad \varepsilon_i \sim \mathcal{N}(0, 1) \qquad \text{(confounder 3)}$$

$$D_i = AZ_i + U_i \qquad\qquad \text{(latent representation)}$$

$$X_i = BD_i + V_i \qquad\qquad \text{(observed representation)}$$

$$Y_i = \theta^\top D + \eta_i(U_i, V_i)$$

With dimensions $n = 10000$, $r = k = 4$, where $i \in \{1, 2, \ldots, n\}$ indexes the samples.

---

---

**Linear DGP 2 Correlated Uniform U and Independent Gaussian V**

Draw DGP parameters

$$A \sim \{N(0, 0.1^2)\}^{r \times k} \qquad B \sim \{N(0,1)\}^{m \times r} \qquad \theta \sim \{N(0,1)\}^{r \times 1}$$
$$E \sim \{N(0,1)\}^{h \times r}$$

Then generate $n$ samples as:

$$Z_i \sim \mathcal{N}(0, I_k) \qquad \text{(instrument)}$$

$$U_i \sim E \cdot \{\text{Unif}(-1, -1)\}^h \qquad \text{(correlated Uniform confounder 1)}$$

$$V_i \sim \mathcal{N}(0, 10^2 \cdot I_m) \qquad \text{(confounder 2)}$$

$$\eta_i(U_i, V_i) = \sum_{j=1}^{r} U_{ij} + 0.2 \cdot \varepsilon_i, \quad \varepsilon_i \sim \mathcal{N}(0,1) \quad \text{(confounder 3)}$$

$$D_i = AZ_i + U_i \qquad \text{(latent representation)}$$

$$X_i = BD_i + V_i \qquad \text{(observed representation)}$$

$$Y_i = \theta^\top D_i + \eta_i(U_i, V_i)$$

With dimensions $n = 10000, r = k = 4, h = 3$, where $i \in \{1, 2, \ldots, n\}$ indexes the samples.

---

**Linear DGP 3 Correlated Uniform U and Correlated Gaussian V**

Draw DGP parameters

$$A \sim \{N(0, 0.1^2)\}^{r \times k} \qquad B \sim \{N(0,1)\}^{m \times r} \qquad \theta \sim \{N(0,1)\}^{r \times 1}$$
$$E \sim \{N(0,1)\}^{h_1 \times r} \qquad F \sim \{N(0,1)\}^{h_2 \times r}$$

Then generate $n$ samples as:

$$Z_i \sim \mathcal{N}(0, I_k) \qquad \text{(instrument)}$$

$$U_i \sim E \cdot \{\text{Unif}(-1, -1)\}^h \qquad \text{(correlated Uniform confounder 1)}$$

$$V_i \sim F \cdot \mathcal{N}(0, 5^2 \cdot I_{h_2}) \qquad \text{(correlated Gaussian confounder 2)}$$

$$\eta_i(U_i, V_i) = \sum_{j=1}^{r} U_{ij} + 0.2 \cdot \varepsilon_i, \quad \varepsilon_i \sim \mathcal{N}(0,1) \quad \text{(confounder 3)}$$

$$D_i = AZ_i + U_i \qquad \text{(latent representation)}$$

$$X_i = BD_i + V_i \qquad \text{(observed representation)}$$

$$Y_i = \theta^\top D_i + \eta_i(U_i, V_i)$$

With dimensions $n = 10000, r = k = 4, h_1 = 3, h_2 = 5$, where $i \in \{1, 2, \ldots, n\}$ indexes the samples.

---

To determine the true outcome after perturbation, We used the formula

$$Y_{\alpha u} = \theta^T (B^\dagger X_{\alpha u}).$$

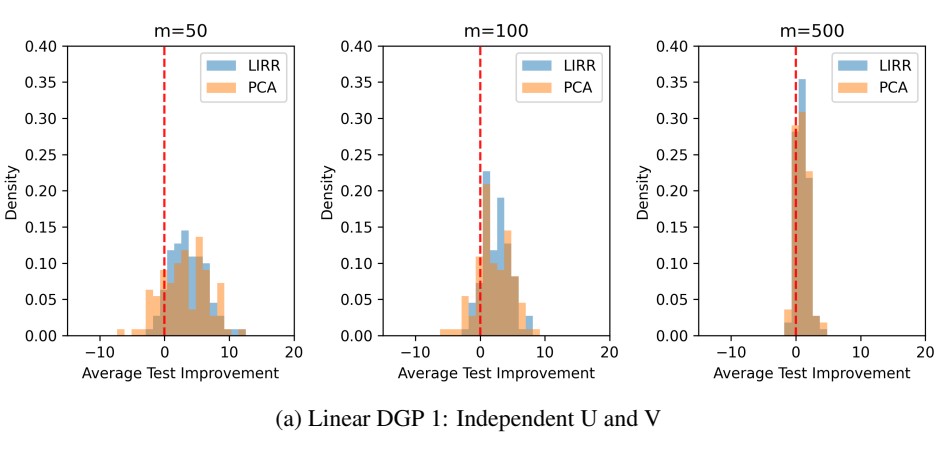

(a) Linear DGP 1: Independent U and V

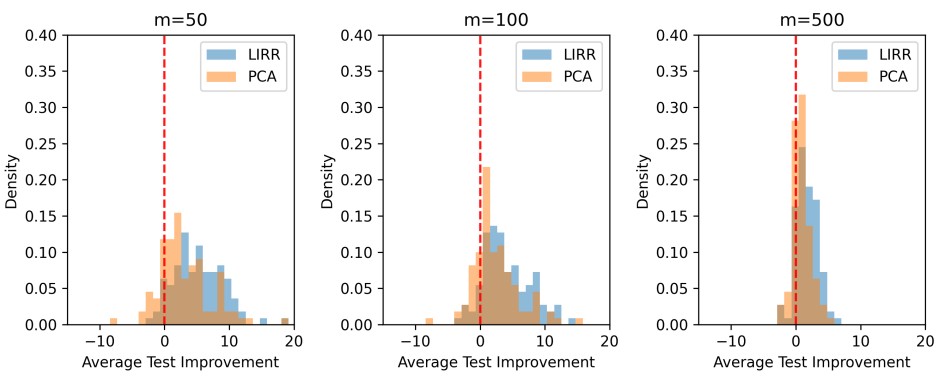

(b) Linear DGP 2: Correlated U and Independent V

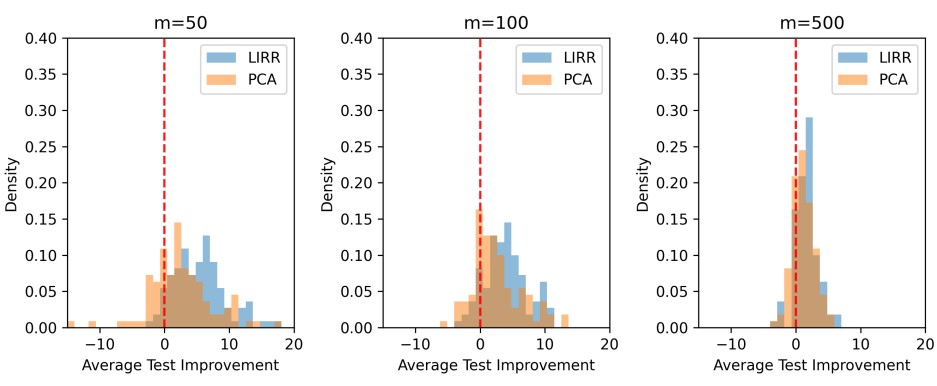

(c) Linear DGP 3: Correlated U and V

Figure 4: Distribution of Average Improvement for Linear Experiment

Each experiment was repeated 100 times with different random seeds, each containing a sample size of 10000 with 80-20 train-test split. We also varied the dimensionality of X, $m$, to examine the dimension effects while holding the dimension of Z constant ($k = 4$).

In addition to the summary statistics included in the main paper, we also plotted the distribution of average test improvements across seeds in Figure 4. We can observe that the test improvements of LIRR are shifted more to the right compared to the baseline PCA method.

## F.2 QUADRATIC

This section provides details of the nonlinear experiments briefly described in Section 5 of the main paper. While the main paper presents summary statistics of average improvements and key findings, here we included detailed data generating equations, model hyperparameter, and histograms of the improvements across runs.

The data are generated using the following 3 cases.

---

**Quadratic DGP 1 Independent Gaussian U, V**

Draw DGP parameters

$$A \sim \{N(0,1)\}^{r \times k} \qquad B \sim \{N(0,1)\}^{m \times (2*r + r*(r-1)/2)} \qquad \theta \sim \{N(0,1)\}^{r \times 1}$$

Then generate samples as:

$$Z_i \sim \mathcal{N}(0, I_k) \qquad \text{(instrument)}$$

$$U_i \sim \mathcal{N}(0, 0.2^2 \cdot I_r) \qquad \text{(confounder 1)}$$

$$V_i \sim \mathcal{N}(0, 0.2^2 \cdot I_m) \qquad \text{(confounder 2)}$$

$$\eta_i(U_i, V_i) = \sum_{j=1}^{r} U_{ij} + 0.2 \cdot \varepsilon_i, \quad \varepsilon_i \sim \mathcal{N}(0,1) \qquad \text{(confounder 3)}$$

$$D_i = AZ_i + U_i \qquad \text{(latent representation)}$$

$$X_i = B \cdot [D_{i1}, D_{i2}, ..., D_{i1}D_{i2}, ...D_{ir}^2] + V_i \qquad \text{(observed representation)}$$

$$Y_i = \theta^\top D + \eta_i(U_i, V_i)$$

With dimensions $n = 10000$, $r = k = 4$, where $i \in \{1, 2, \ldots, n\}$ indexes the samples.

---

**Quadratic DGP 2 Correlated Uniform U and Independent Gaussian V**

Draw DGP parameters

$$A \sim \{N(0,1)\}^{r \times k} \qquad B \sim \{N(0,1)\}^{m \times (2*r+r*(r-1)/2)} \qquad \theta \sim \{N(0,1)\}^{r \times 1}$$
$$E \sim \{N(0,1)\}^{h \times r}$$

Then generate samples as:

$$Z_i \sim \mathcal{N}(0, I_k) \qquad\qquad\qquad \text{(instrument)}$$

$$U_i \sim E \cdot \{\text{Unif}(-0.2, -0.2)\}^h \qquad\qquad \text{(correlated Uniform confounder 1)}$$

$$V_i \sim \mathcal{N}(0, 0.2^2 \cdot I_m) \qquad\qquad \text{(confounder 2)}$$

$$\eta_i(U_i, V_i) = \sum_{j=1}^{r} U_{ij} + 0.2 \cdot \varepsilon_i, \quad \varepsilon_i \sim \mathcal{N}(0,1) \qquad \text{(confounder 3)}$$

$$D_i = AZ_i + U_i \qquad\qquad\qquad \text{(latent representation)}$$

$$X_i = B \cdot [D_{i1}, D_{i2}, ..., D_{i1}D_{i2}, ...D_{ir}^2] + V_i \quad \text{(observed representation)}$$

$$Y_i = \theta^\top D_i + \eta_i(U_i, V_i)$$

With dimensions $n = 10000$, $r = k = 4, h = 3$, where $i \in \{1, 2, \ldots, n\}$ indexes the samples.

---

**Quadratic DGP 3 Correlated Uniform U and Correlated Gaussian V**

Draw DGP parameters

$$A \sim \{N(0,1)\}^{r \times k} \qquad B \sim \{N(0,1)\}^{m \times (2*r+r*(r-1)/2)} \qquad \theta \sim \{N(0,1)\}^{r \times 1}$$
$$E \sim \{N(0,1)\}^{h_1 \times r} \qquad F \sim \{N(0,1)\}^{h_2 \times r}$$

Then generate samples as:

$$Z_i \sim \mathcal{N}(0, I_k) \qquad\qquad\qquad \text{(instrument)}$$

$$U_i \sim E \cdot \{\text{Unif}(-0.2, -0.2)\}^h \qquad\qquad \text{(correlated Uniform confounder 1)}$$

$$V_i \sim F \cdot \mathcal{N}(0, 0.05^2 \cdot I_{h_2}) \qquad\qquad \text{(correlated Gaussian confounder 2)}$$

$$\eta_i(U_i, V_i) = \sum_{j=1}^{r} U_{ij} + 0.2 \cdot \varepsilon_i, \quad \varepsilon_i \sim \mathcal{N}(0,1) \qquad \text{(confounder 3)}$$

$$D_i = AZ_i + U_i \qquad\qquad\qquad \text{(latent representation)}$$

$$X_i = B \cdot [D_{i1}, D_{i2}, ..., D_{i1}D_{i2}, ...D_{ir}^2] + V_i \quad \text{(observed representation)}$$

$$Y_i = \theta^\top D_i + \eta_i(U_i, V_i)$$

With dimensions $n = 10000$, $r = k = 4, h_1 = 3, h_2 = 5$, where $i \in \{1, 2, \ldots, n\}$ indexes the samples.

All encoder architectures incorporate a Random Fourier Feature layer, followed by three feedforward layers and a final linear projection. Decoders consist of three feedforward layers and a final linear projection layer. For our IRAE[2] and IRAE models, we set the bottleneck dimension to 10, larger

Table 4: Training Parameters for Quadratic Simulations

|  | Vanilla AE | IRAE[0] | IRAE[1] | IRAE[2] | IRAE | VAE | iVAE |
|---|---|---|---|---|---|---|---|
| **Architecture** | | | | | | | |
| Encoder dimensions | | | $100 \rightarrow 50 \rightarrow 20$ | | | | |
| Decoder dimensions | | | $20 \rightarrow 50 \rightarrow 100$ | | | | |
| RFF bandwidth $\sigma$ | | | 20 | | | | |
| Bottleneck dimension | 4 | 4 | 4 | 10 | 10 | 4 | 4 |
| **Optimization** | | | | | | | |
| Optimizer | | | RMSprop | | | | |
| Learning rate | | | $5 \times 10^{-4}$ | | | $1 \times 10^{-4}$ | $5 \times 10^{-4}$ |
| Alpha | | | 0.9 | | | | |
| Epsilon | | | $1 \times 10^{-8}$ | | | | |
| Weight decay | | | $1 \times 10^{-6}$ | | | | |
| Momentum | | | None | | | | |
| **Regularization Parameters** | | | | | | | |
| $\lambda$ | 0 | 1 | 1 | 1 | 1 | NA | NA |
| $\mu_1$ | 0 | 0 | 1 | 1 | 1 | NA | NA |
| $\mu_2$ | 0 | 0 | 0 | 1 | 1 | NA | NA |
| $\mu_3$ | 0 | 0 | 0 | 0 | 1 | NA | NA |
| weight for kl term | NA | NA | NA | NA | NA | 3 | 3 |
| | | | $c_1 = 1.0, c_2 = c_3 = 0.0$ | | | | |
| **Training Protocol (with early stopping of patience 20)** | | | | | | | |
| | | | 1000 epochs | | | | |

than the instrumental variable dimension $r = k = 4$. By construction, Vanilla and IRAE[1] has bottleneck equal to $k = 4$. To determine the true outcome after perturbation, we used the formula

$$Y_{\alpha u} = \theta^T((B^\dagger X_{\alpha u})[: r]),$$

where $[: r]$ index into the first order terms (excluding the quadratic and cross terms) of $D$.

The hyperparameters used in the training procedure are described in Table 4.

Additional plots corresponding to Table 4 are included in Figure 5.

## F.3 MNIST EXPERIMENT 1

This section provides details of the MNIST experiments briefly described in Section 5 of the main paper. Here we included detailed data generating equations, model hyperparameter, and plots for IRAE[0], IRAE[1], IRAE[2] that were not included in the main paper.

The data for MNIST experiment is generated using *Case 1 DGP*.

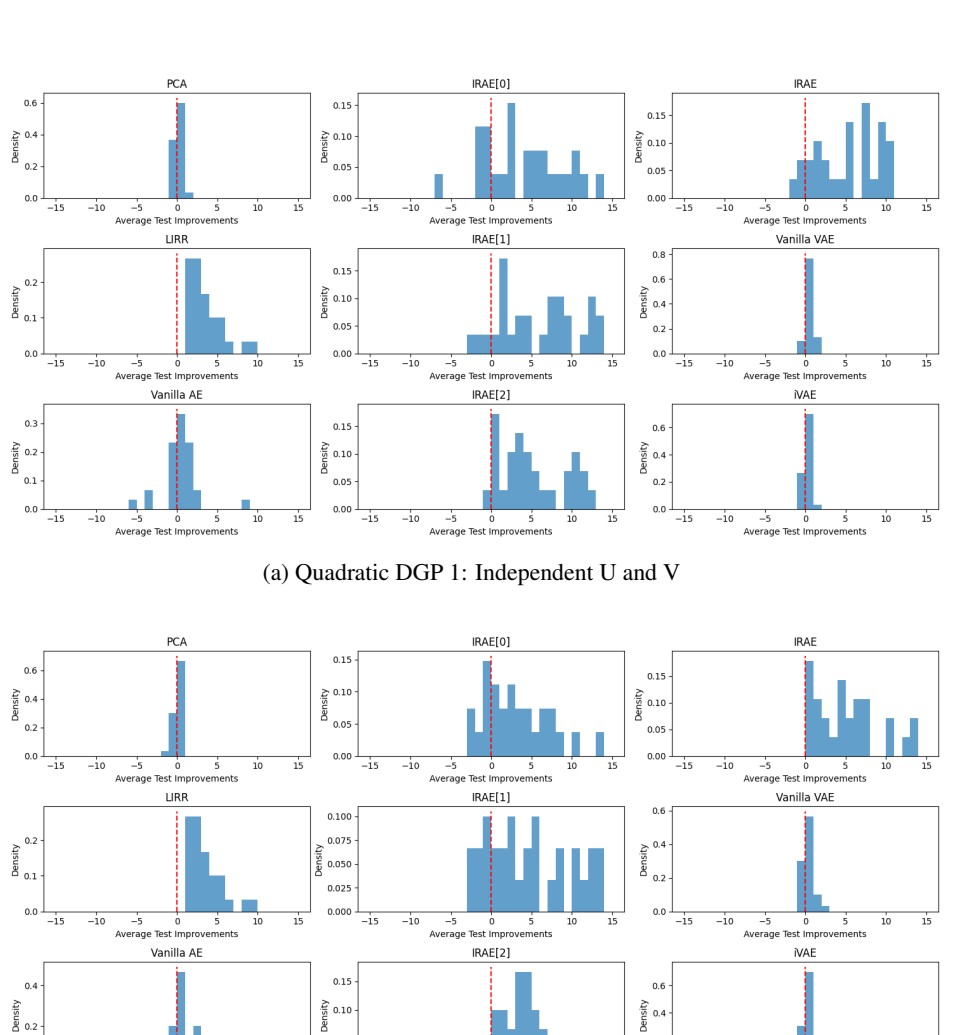

(a) Quadratic DGP 1: Independent U and V

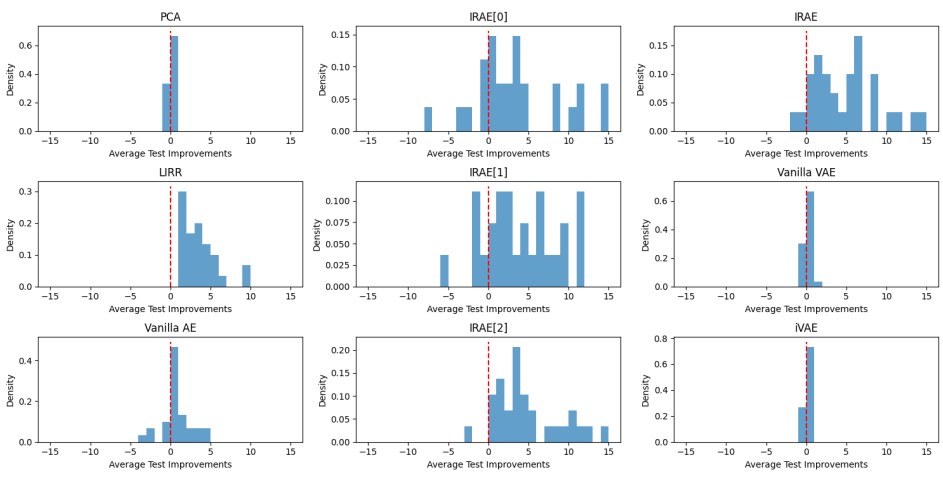

(b) Quadratic DGP 2: Correlated U and Independent V

(c) Quadratic DGP 3: Correlated U and V

Figure 5: Distribution of Average Improvement for Quadratic Experiment

---

**Case 1 DGP**

Draw DGP parameters $\alpha, \beta \sim \text{Unif}(0.1, 0.7)$. Then generate samples as:

$$G_i \in [0, 1]^{28 \times 28} \qquad \text{(grayscale MNIST image)}$$

$$Z_i, \ U_i \sim \mathcal{N}(0, I_2), \qquad Z_i \perp\!\!\!\perp U_i \qquad \text{(instrument \& confounder)}$$

$$r_i = \text{clip}(0.5 + \alpha\, Z_{i1} + \beta\, U_{i1},\, 0,\, 1) \qquad \text{(red channel)}$$

$$g_i = \text{clip}(0.5 + \alpha\, Z_{i2} + \beta\, U_{i2},\, 0,\, 1) \qquad \text{(green channel)}$$

$$b_i = \text{clip}\left(0.5 + \alpha\, \frac{Z_{i1} + Z_{i2}}{2},\, 0,\, 1\right) \qquad \text{(blue channel)}$$

$$X_i(k, \ell, c) = G_i(k, \ell) \cdot (r_i, g_i, b_i)_c, \qquad \begin{aligned} &c \in \{R, G, B\}, \\ &(k, \ell) \in \{1, \ldots, 28\}^2 \end{aligned} \qquad \text{(colour image)}$$

$$Y_i = r_i + g_i + b_i. \qquad \text{(outcome, details below)}$$

Returns the tuples $(Z_i, X_i, Y_i)$.

---

All encoders consist of three Conv2D layers, followed by additional feedforward layers, and conclude with a linear projection. Decoders mirror this architecture in reverse order. For our IRAE[2] and IRAE models, we set the bottleneck dimension to 10 which is larger than $k = 2$. For vanilla and IRAE[0], IRAE[1], the bottle neck is 2. The autoencoder with multiple HSIC regularization terms presents greater training challenges due to the complexity of term. To address this, we initialized IRAE[2] and IRAE with weights from the simpler IRAE[1] model. All of models are trained with 60k training samples and evaluated on 10k test set. More training details can be found in Table 5.

Table 5: Training Parameters for MNIST Simulations

| Parameter | Vanilla AE | IRAE[0] | IRAE[1] | IRAE[2] | IRAE |
|---|---|---|---|---|---|
| **Architecture** | | | | | |
| Kernel Size | | | 3 | | |
| Encoder channels | | | $16 \to 32 \to 64$ | | |
| Decoder channels | | | $64 \to 32 \to 16$ | | |
| Bottleneck dimension | 2 | 2 | 2 | 10 | 10 |
| **Optimization** | | | | | |
| Optimizer | | | Adam (default parameters in torch) | | |
| Learning rate | | | $1 \times 10^{-3}$ | | |
| Weight initialization | None | None | None | From IRAE[1] | From IRAE[1] |
| **Loss Weights** | | | | | |
| $\lambda$ | 0 | 10 | 10 | 10 | 10 |
| $\mu_1$ | 0 | 0 | 10 | 10 | 10 |
| $\mu_2$ | 0 | 0 | 0 | 10 | 10 |
| $\mu_3$ | 0 | 0 | 0 | 0 | 10 |
| | | | $c_1 = 0.8, c_2 = 0.15, c_3 = 0.05$ | | |
| Weights $\lambda, \mu_2, \mu_3$ took value 1 whenever non-zero, instead of 10 for experiments in the Case 2-4 DGPs. | | | | | |
| **Training Epochs (with early stopping of patience 5)** | | | | | |
| | 50 | 50 | 50 | 50* | 50* |

* Additional epochs after initializing with weights from IRAE[1]

*Remark* F.1 (Calculation of Outcome from Image). To calculate expected $Y_{\alpha u}$, we first perform 2-mean clustering on the image pixels and extract the red, green, blue values from the center of the colored cluster. Then, we take the sum of these values as $Y$. Note that is this similar to taking the

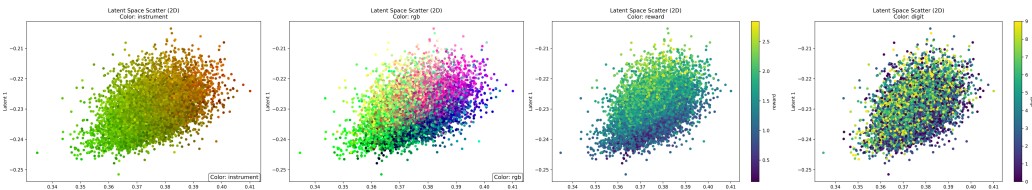

Figure 6: Alignment of recovered latent variables with instrument, true representation [R,G,B], reward (Y) and digit for the **IRAE** model (Case 1 DGP). Data points with similar instrument, color, and reward are grouped together in the latent space.

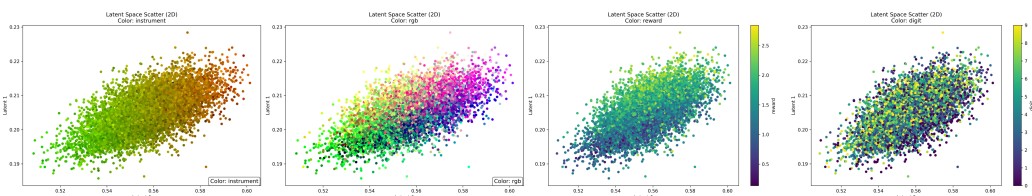

Figure 7: Alignment of recovered latent variables with instrument, true representation [R,G,B], reward (Y) and digit for the **IRAE[2]** model (Case 1 DGP).

average colors over the gray scale mask so the colors would be slightly smaller than the original colors. We tested the methods on the original image and the result is 0.2 smaller on average.

*Remark* F.2 (Calculation of Outcome Improvement). When calculating the outcome improvement of the intervention, take the difference between the kmeans calculation described in the previous paragraph applied to the image produced by the intervention and we subtract the outcome of the kmeans calculation when applied to the original image.

*Remark* F.3. We use a linear kernel for HSIC in order to perform benchmarking at a large scale in fast speed, which may not capture all nonlinear dependencies in this complex image representation setting. More complex independence statistics based on domain knowledge, could perhaps lead to more disentanglement, albeit they might also be harder to train. In subsequent section experiments we also examine a pairwise RBF Kernel based HSIC and we find that it does not lead to improved performance as compared to the linear kernel.

*Remark* F.4. We observe that this example does not perfectly align with the formulation in Equation (1). Here, the number of instruments is 2, which is fewer than the natural representation of D of 3 colors. We may be able to interpret the learned representation as a 2-dimensional subspace of the 3-dimensional color representation, but the mapping from Z to D is still not immediately invertible as assumed in the theory. Additionally, while our theoretical analysis assumes a mapping from color $D$ to outcome directly, our calculation employs k-means clustering on $X$ instead. Nevertheless, this example demonstrates that our method performs robustly even in settings beyond those covered by our theoretical guarantees, and offers potential future directions of theoretical investigation.

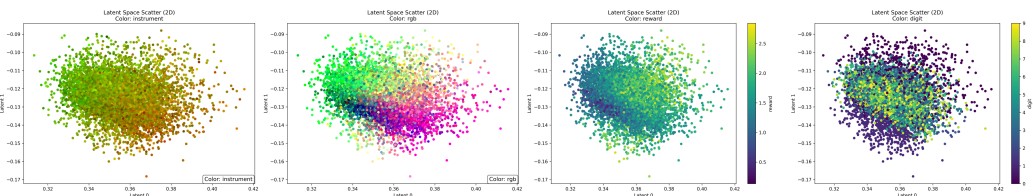

Figure 8: Alignment of recovered latent variables with instrument, true representation [R,G,B], reward (Y) and digit for the **IRAE[1]** model (Case 1 DGP).

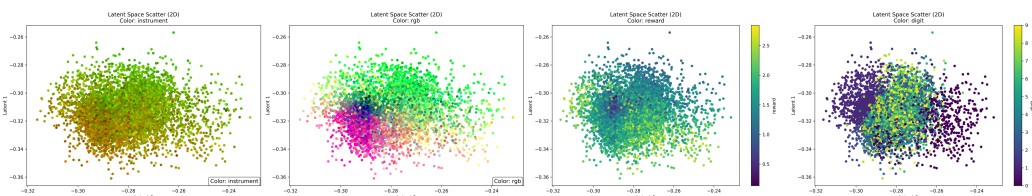

Figure 9: Alignment of recovered latent variables with instrument, true representation [R,G,B], reward (Y) and digit for the **IRAE[0]** model (Case 1 DGP).

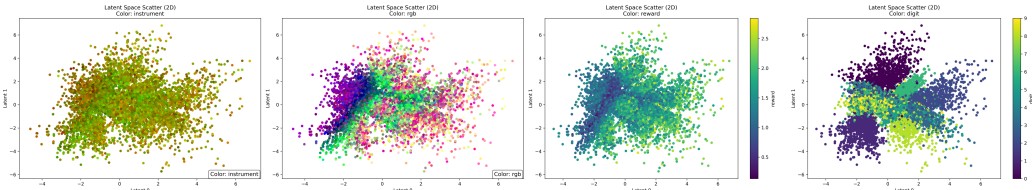

Figure 10: Alignment of recovered latent variables with instrument, true representation [R,G,B], reward (Y) and digit for the **Vanilla AE** model (Case 1 DGP).

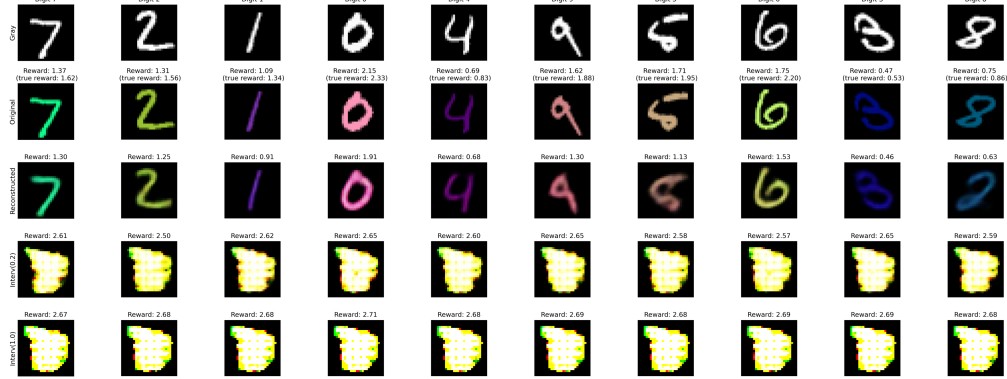

Figure 11: **IRAE** on Case 1 DGP for one random seed (random seed 22), with a Conv AutoEncoder, linear HSIC as independence criterion, **latent dimension 10**, regularization weights $\lambda = \mu_1 = \mu_2 = 10$ and training for 50 epochs with early stopping (patience 5 epochs) warm start from IRAE[1].

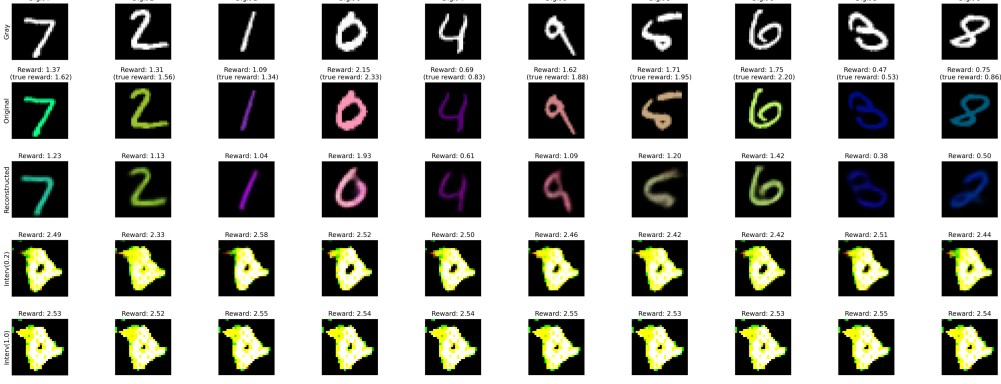

Figure 12: **IRAE[2]** on Case 1 DGP for one random seed (random seed 22), with a Conv AutoEncoder, linear HSIC as independence criterion, **latent dimension 10**, regularization weights $\lambda = \mu_1 = \mu_2 = 10$ and training for 50 epochs with early stopping (patience 5 epochs) warm start from IRAE[1].

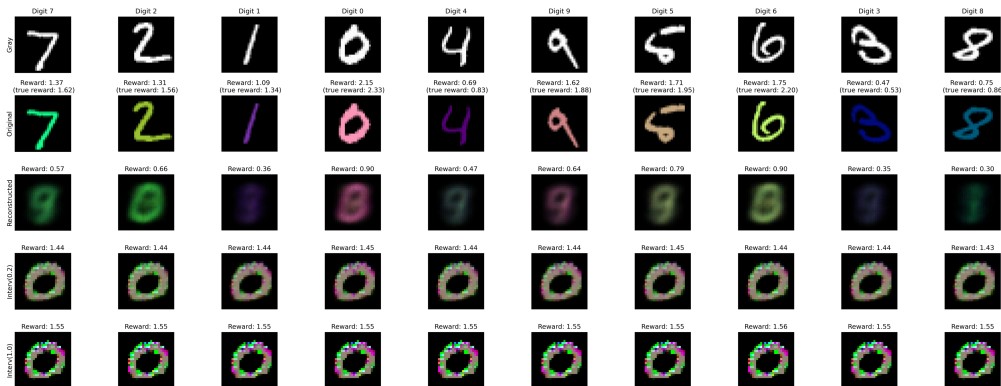

Figure 13: **IRAE[1]** on Case 1 DGP for one random seed (random seed 22), with a Conv AutoEncoder, linear HSIC as independence criterion, **latent dimension 2**, regularization weights $\lambda = \mu_1 = 10$ and training for 50 epochs with early stopping (patience 5 epochs) from scratch

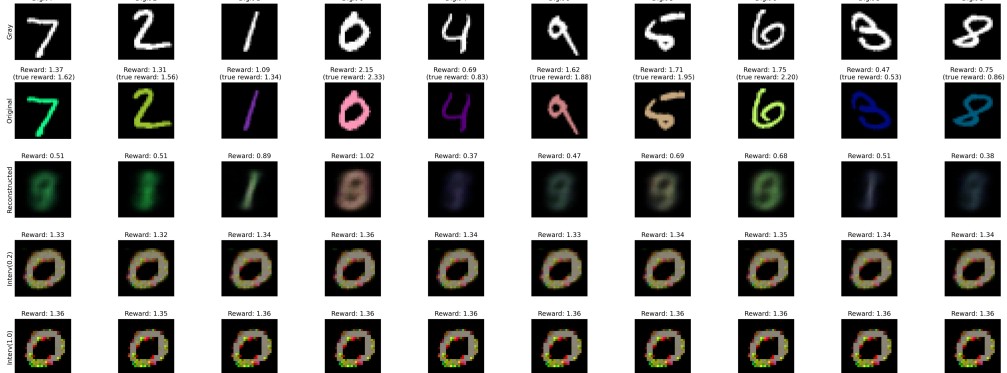

Figure 14: **IRAE[0]** on Case 1 DGP for one random seed (random seed 22), with a Conv AutoEncoder, linear HSIC as independence criterion, **latent dimension 2**, regularization weights $\lambda = \mu_1 = 10$ and training for 50 epochs with early stopping (patience 5 epochs) from scratch

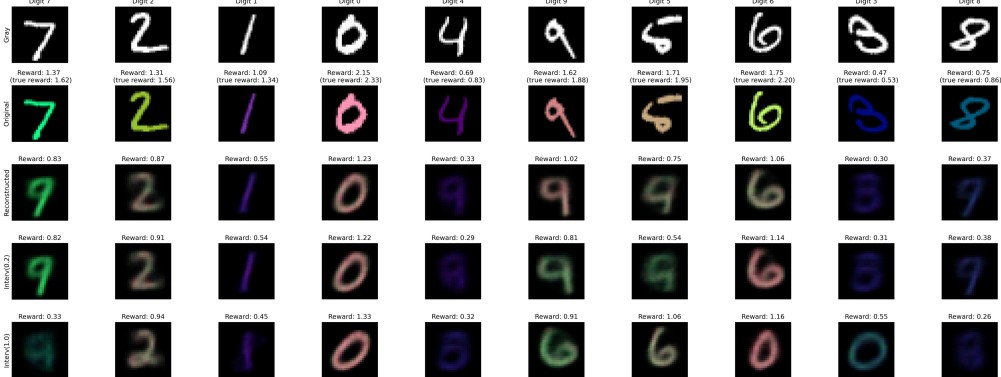

Figure 15: **Vanilla AE** on Case 1 DGP for one random seed (random seed 22), with a Conv AutoEncoder, **latent dimension 2**, regularization weights $\lambda = \mu_1 = 10$ and training for 50 epochs with early stopping (patience 5 epochs) from scratch

Table 6: Summary of parameters explored in MNIST Experiment 2

| Setting Category | Options | Description |
|---|---|---|
| **Data Generating Process** | DGP2 | Three Instruments |
| **Autoencoder Architecture** | Dense | **Encoder:** Dense layer $3 \times 28 \times 28 \to 512$, followed by linear projection to latent dimension **Decoder:** Linear layer from latent dimension to 512, followed by dense layer $512 \to 3 \times 28 \times 28$ |
| | Convolution | **Encoder:** Three Conv2D layers with channel $16 \to 32 \to 64$ of kernel size 3, followed by a dense layer of size 256 and linear projection to latent dimension **Decoder:** Linear layer from latent dimension to size 256, followed by dense layer and three Conv2D layers with channel $64 \to 32 \to 16$ of kernel size 3 |
| **Latent Dimension IRAE[2] and IRAE** | 10 32 | Used for IRAE[2] and IRAE models Used for IRAE[2] and IRAE models |
| **Regularization Type** | Linear HSIC Pairwise HSIC | Applied as independence measure on the entire vector Applied between pairwise coordinates |
| **Weight Initialization IRAE[2] and IRAE** | Without warmstart With warmstart | Training from randomly initialized weights for 50 epochs Initializing with weights transferred from a pre-trained IRAE[1] model, and training for additional 50 epochs |

## F.4 MNIST EXPERIMENT 2

Building on the results from our MNIST experiments in Section 5, we conducted a more comprehensive evaluation by exploring additional hyperparameter configurations and data generating processes. Given that independence test statistics are often complex and challenging to train, we systematically investigated various model architectures, independence test statistics calculation, and initialization strategies to identify optimal configurations. To align with our theoretical requirements outlined in Theorem 2.2, we evaluated our approach on a supplementary dataset with three instruments, denoted as *Case 2 DGP*.

Our findings reveal that simpler dense architectures perform at least as well as, and often better than, more complex convolutional neural networks for this task. Furthermore, we observed that larger bottleneck dimensions in IRAE[2] and IRAE models better preserve the original digit morphology in treated images — a potentially valuable property when morphological features is confounded the outcome variable.

The full set of hyperparameters explored are included in Table 6. All of models are trained with 60k training samples and evaluated on 10k test set, for 40 random seeds. Regularization weights are 0 or 1. All models are trained with 50 epochs after initialization with early stopping of patience 5.

---

**Case 2 DGP**

Draw DGP parameters $\alpha, \beta \sim \text{Unif}(0.1, 0.7)$. Then generate samples as:

$$G_i \in [0,1]^{28 \times 28} \qquad \text{(grayscale MNIST image)}$$

$$Z_i, \, U_i \sim \mathcal{N}(0, I_3), \qquad Z_i \perp\!\!\!\perp U_i \qquad \text{(instrument \& confounder)}$$

$$r_i = \text{clip}(0.5 + \alpha\, Z_{i1} + \beta\, U_{i1},\, 0,\, 1) \qquad \text{(red channel)}$$

$$g_i = \text{clip}(0.5 + \alpha\, Z_{i2} + \beta\, U_{i2},\, 0,\, 1) \qquad \text{(green channel)}$$

$$b_i = \text{clip}(0.5 + \alpha\, Z_{i3} + \beta\, U_{i3},\, 0,\, 1) \qquad \text{(blue channel)}$$

$$X_i(k, \ell, c) = G_i(k, \ell) \cdot (r_i, g_i, b_i)_c, \qquad \begin{matrix} c \in \{R, G, B\}, \\ (k, \ell) \in \{1, \ldots, 28\}^2 \end{matrix} \qquad \text{(colour image)}$$

$$Y_i = r_i + g_i + b_i. \qquad \text{(outcome)}$$

Returns the tuples $(Z_i, X_i, Y_i)$.

---

We highlight some findings from our exploration of the performance of our proposed methods across various hyperparameter dimensions:

**Architecture:** We found that simple dense layers can achieve better performance than convolutional architectures for this task, suggesting that Conv2D layers may be unnecessarily complex for this particular example.

**Data Generating Process:** Our experimental results demonstrate that the relative performance of our methods remains consistent across both DGP1 and DGP2.

**Latent Dimension:** When using larger latent dimensions (32), both the reconstructed and treated images preserved more of the original digit morphology although the improvement is smaller (c.f. Figures 17 to 22). This may be a desired property in some cases, especially in the case that the digit morphology is a confounder (not tested in our experiment) and has a direct effect on the outcome.

**Regularization Type:** While pairwise HSIC may theoretically capture more nonlinear dependencies, we found that it was often more difficult to train in practice. Linear HSIC consistently yielded better performance with greater training stability.

**Weight Initialization:** Dense architectures performed well without warm start initialization, while convolutional architectures benefited significantly from weight transfer. This difference likely stems from the higher complexity and larger parameter space of convolutional networks.

Overall, the best improvement model stems from the IRAE method with all regularizers, a Dense architecture, latent = 10, linear HSIC with no warm start.

| Arch. | Latent Dim | Reg Type | Warm Start | image | Vanilla AE | IRAE[0] | IRAE[1] | IRAE[2] | IRAE |
|---|---|---|---|---|---|---|---|---|---|
| dense | 10 | linear | False | reconstructed | -0.46 (0.02) | -0.67 (0.02) | -0.67 (0.02) | **-0.27 (0.01)** | -0.28 (0.02) |
| | | | | intervened(0.2) | -0.45 (0.02) | **1.4 (0.12)** | **1.4 (0.1)** | 1.39 (0.15) | 1.39 (0.2) |
| | | | | intervened(1.0) | -0.37 (0.02) | 1.54 (0.11) | 1.54 (0.09) | **1.57 (0.12)** | 1.54 (0.18) |
| | | | True | reconstructed | -0.46 (0.02) | -0.67 (0.02) | -0.67 (0.02) | **-0.36 (0.14)** | -0.39 (0.16) |
| | | | | intervened(0.2) | -0.45 (0.02) | **1.4 (0.12)** | **1.4 (0.1)** | 1.17 (0.53) | 1.12 (0.58) |
| | | | | intervened(1.0) | -0.37 (0.02) | **1.54 (0.11)** | **1.54 (0.09)** | 1.32 (0.5) | 1.18 (0.6) |
| | | pairwise | False | reconstructed | -0.46 (0.02) | -0.67 (0.02) | -0.68 (0.01) | **-0.3 (0.03)** | -0.35 (0.03) |
| | | | | intervened(0.2) | -0.45 (0.02) | **1.4 (0.12)** | **1.4 (0.14)** | -0.09 (0.37) | 0.12 (0.53) |
| | | | | intervened(1.0) | -0.37 (0.02) | **1.54 (0.11)** | 1.53 (0.13) | 0.09 (0.57) | 0.35 (0.58) |
| | | | True | reconstructed | -0.46 (0.02) | -0.67 (0.02) | -0.68 (0.01) | **-0.33 (0.1)** | -0.57 (0.22) |
| | | | | intervened(0.2) | -0.45 (0.02) | **1.4 (0.12)** | **1.4 (0.14)** | 1.31 (0.24) | 0.87 (0.78) |
| | | | | intervened(1.0) | -0.37 (0.02) | **1.54 (0.11)** | 1.53 (0.13) | 1.49 (0.15) | 1.12 (0.59) |
| | 32 | linear | False | reconstructed | -0.46 (0.02) | -0.67 (0.02) | -0.67 (0.02) | **-0.14 (0.02)** | **-0.14 (0.01)** |
| | | | | intervened(0.2) | -0.45 (0.02) | **1.4 (0.12)** | **1.4 (0.1)** | 0.74 (0.34) | 0.66 (0.38) |
| | | | | intervened(1.0) | -0.37 (0.02) | **1.54 (0.11)** | **1.54 (0.09)** | 1.43 (0.31) | 1.35 (0.38) |
| | | | True | reconstructed | -0.46 (0.02) | -0.67 (0.02) | -0.67 (0.02) | **-0.26 (0.12)** | **-0.26 (0.15)** |
| | | | | intervened(0.2) | -0.45 (0.02) | **1.4 (0.12)** | **1.4 (0.1)** | 1.08 (0.36) | 1.03 (0.5) |
| | | | | intervened(1.0) | -0.37 (0.02) | **1.54 (0.11)** | **1.54 (0.09)** | 1.29 (0.42) | 1.19 (0.49) |
| | | pairwise | False | reconstructed | -0.46 (0.02) | -0.67 (0.02) | -0.68 (0.01) | **-0.13 (0.01)** | -0.19 (0.02) |
| | | | | intervened(0.2) | -0.45 (0.02) | **1.4 (0.12)** | **1.4 (0.14)** | -0.15 (0.05) | -0.23 (0.09) |
| | | | | intervened(1.0) | -0.37 (0.02) | **1.54 (0.11)** | 1.53 (0.13) | -0.2 (0.18) | -0.26 (0.3) |
| | | | True | reconstructed | -0.46 (0.02) | -0.67 (0.02) | -0.68 (0.01) | **-0.19 (0.05)** | -0.31 (0.15) |
| | | | | intervened(0.2) | -0.45 (0.02) | **1.4 (0.12)** | **1.4 (0.14)** | 0.07 (0.41) | 0.02 (0.47) |
| | | | | intervened(1.0) | -0.37 (0.02) | **1.54 (0.11)** | 1.53 (0.13) | 0.42 (0.65) | 0.4 (0.68) |
| conv | 10 | linear | False | reconstructed | -0.37 (0.02) | -0.6 (0.06) | -0.6 (0.05) | **-0.21 (0.03)** | -0.22 (0.03) |
| | | | | intervened(0.2) | -0.36 (0.03) | 0.21 (0.34) | 0.4 (0.4) | **0.98 (0.23)** | 0.91 (0.32) |
| | | | | intervened(1.0) | -0.31 (0.07) | 0.4 (0.56) | 0.69 (0.58) | **1.25 (0.55)** | 1.18 (0.57) |
| | | | True | reconstructed | -0.37 (0.02) | -0.6 (0.06) | -0.6 (0.05) | **-0.2 (0.04)** | **-0.2 (0.03)** |
| | | | | intervened(0.2) | -0.36 (0.03) | 0.21 (0.34) | 0.4 (0.4) | **1.0 (0.45)** | 0.9 (0.51) |
| | | | | intervened(1.0) | -0.31 (0.07) | 0.4 (0.56) | 0.69 (0.58) | **0.89 (0.75)** | 0.64 (0.75) |
| | | pairwise | False | reconstructed | -0.37 (0.02) | -0.6 (0.06) | -0.6 (0.05) | **-0.22 (0.05)** | -0.27 (0.08) |
| | | | | intervened(0.2) | -0.36 (0.03) | 0.21 (0.34) | 0.04 (0.45) | 0.47 (0.42) | **0.49 (0.47)** |
| | | | | intervened(1.0) | -0.31 (0.07) | 0.4 (0.56) | 0.12 (0.57) | 0.86 (0.47) | **0.89 (0.55)** |
| | | | True | reconstructed | -0.37 (0.02) | -0.6 (0.06) | -0.6 (0.05) | **-0.26 (0.07)** | -0.3 (0.08) |
| | | | | intervened(0.2) | -0.36 (0.03) | 0.21 (0.34) | 0.04 (0.45) | **0.82 (0.47)** | 0.54 (0.51) |
| | | | | intervened(1.0) | -0.31 (0.07) | 0.4 (0.56) | 0.12 (0.57) | **1.08 (0.55)** | 0.75 (0.64) |
| | 32 | linear | False | reconstructed | -0.37 (0.02) | -0.6 (0.06) | -0.6 (0.05) | **-0.1 (0.03)** | -0.11 (0.04) |
| | | | | intervened(0.2) | -0.36 (0.03) | 0.21 (0.34) | 0.4 (0.4) | **0.7 (0.33)** | 0.56 (0.38) |
| | | | | intervened(1.0) | -0.31 (0.07) | 0.4 (0.56) | 0.69 (0.58) | **1.26 (0.39)** | 1.11 (0.5) |
| | | | True | reconstructed | -0.37 (0.02) | -0.6 (0.06) | -0.6 (0.05) | **-0.1 (0.03)** | **-0.1 (0.04)** |
| | | | | intervened(0.2) | -0.36 (0.03) | 0.21 (0.34) | 0.4 (0.4) | 1.05 (0.49) | **1.13 (0.38)** |
| | | | | intervened(1.0) | -0.31 (0.07) | 0.4 (0.56) | 0.69 (0.58) | **1.11 (0.57)** | 1.09 (0.64) |
| | | pairwise | False | reconstructed | -0.37 (0.02) | -0.6 (0.06) | -0.6 (0.05) | **-0.11 (0.03)** | -0.14 (0.06) |
| | | | | intervened(0.2) | -0.36 (0.03) | **0.21 (0.34)** | 0.04 (0.45) | 0.02 (0.26) | -0.02 (0.31) |
| | | | | intervened(1.0) | -0.31 (0.07) | **0.4 (0.56)** | 0.12 (0.57) | 0.21 (0.49) | 0.1 (0.55) |
| | | | True | reconstructed | -0.37 (0.02) | -0.6 (0.06) | -0.6 (0.05) | **-0.14 (0.08)** | -0.19 (0.07) |
| | | | | intervened(0.2) | -0.36 (0.03) | 0.21 (0.34) | 0.04 (0.45) | 0.4 (0.56) | **0.5 (0.45)** |
| | | | | intervened(1.0) | -0.31 (0.07) | 0.4 (0.56) | 0.12 (0.57) | 0.68 (0.68) | **0.82 (0.59)** |

Figure 16: Experimental results for the **Case 2** data generating process. Mean improvement and standard deviation of improvement is reported. *reconstructed* refers to the mean outcome improvement of the reconstructed image from the autoencoder with no intervention in the latents, as compared to the original image. *intervened($\alpha$)* refers to the mean outcome improvement of the image produced by intervening on the latents in direction $\alpha \cdot u$, where $u = \theta/\|\theta\|$ and $\theta$ is estimated by 2SLS in latent space.

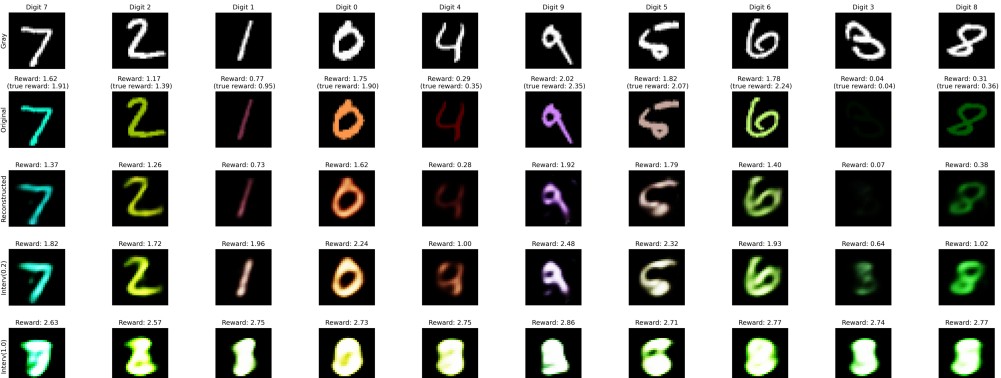

Figure 17: **IRAE** on Case 2 DGP for one random seed (random seed 22), with a Dense AutoEncoder, linear HSIC as independence criterion, **latent dimension 32**, regularization weights $\lambda = \mu_1 = \mu_2 = \mu_3 = 1$ and training for 50 epochs with early stopping (patience 5 epochs) from scratch (no warm start from IRAE1).

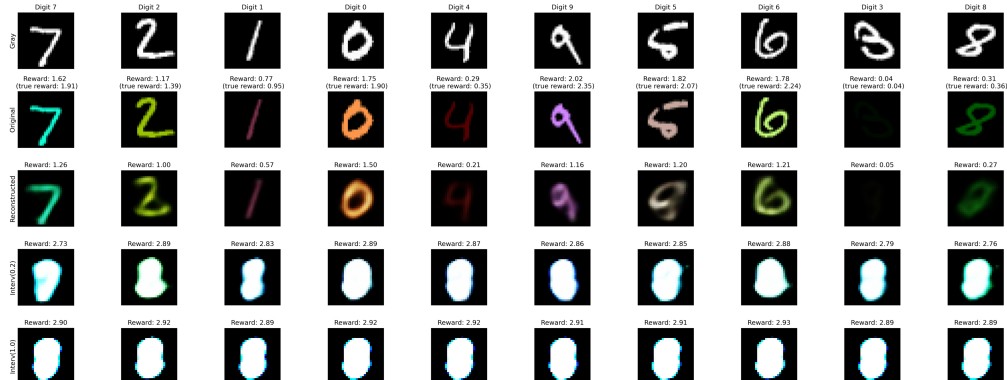

Figure 18: **IRAE** on Case 2 DGP for one random seed (random seed 22), with a Dense AutoEncoder, linear HSIC as independence criterion, **latent dimension 10**, regularization weights $\lambda = \mu_1 = \mu_2 = \mu_3 = 1$ and training for 50 epochs with early stopping (patience 5 epochs) from scratch (no warm start from IRAE1).

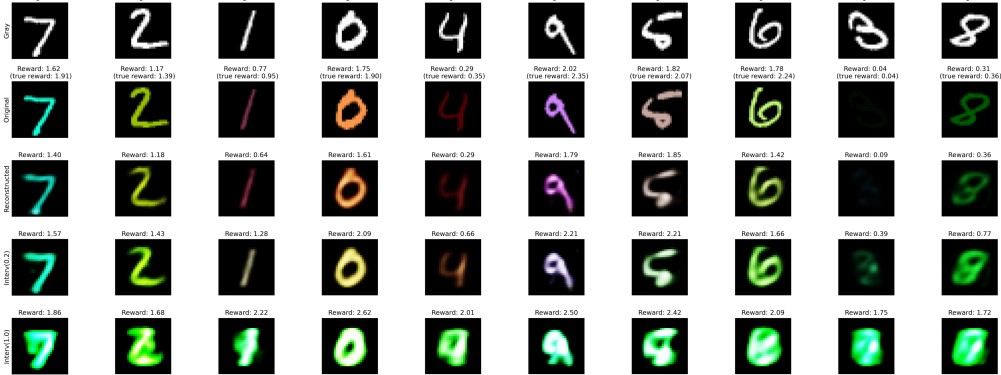

Figure 19: **IRAE[2]** on Case 2 DGP for one random seed (random seed 22), with a Dense AutoEncoder, linear HSIC as independence criterion, **latent dimension 32**, regularization weights $\lambda = \mu_1 = \mu_2 = 1$ and training for 50 epochs with early stopping (patience 5 epochs) from scratch (no warm start from IRAE[1]).

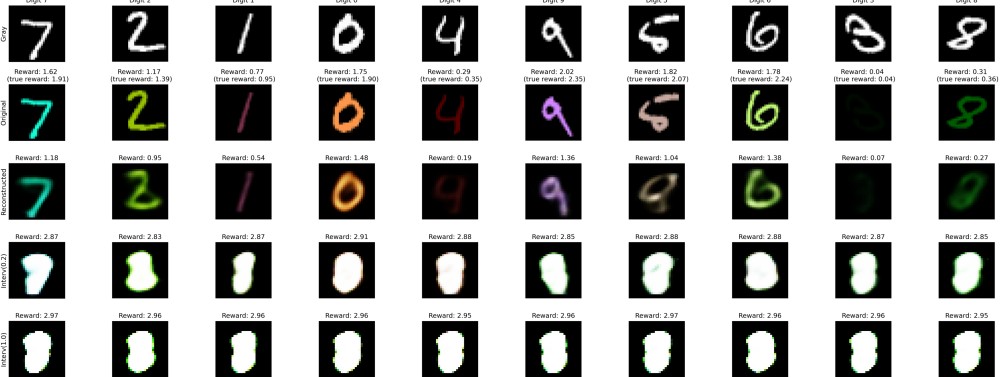

Figure 20: **IRAE[2]** on Case 2 DGP for one random seed (random seed 22), with a Dense AutoEncoder, linear HSIC as independence criterion, **latent dimension 10**, regularization weights $\lambda = \mu_1 = \mu_2 = 1$ and training for 50 epochs with early stopping (patience 5 epochs) from scratch (no warm start from IRAE[1]).

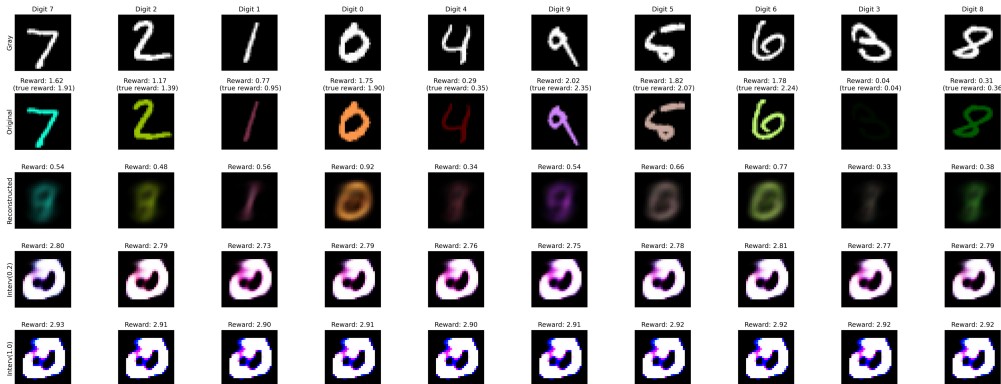

Figure 21: **IRAE[1]** on Case 2 DGP for one random seed (random seed 22), with a Dense AutoEncoder, linear HSIC as independence criterion, **latent dimension 3 = number of instruments**, regularization weights $\lambda = \mu_1 = 1$ and $\mu_2 = \mu_3 = 0$ and training for 50 epochs with early stopping (patience 5 epochs).

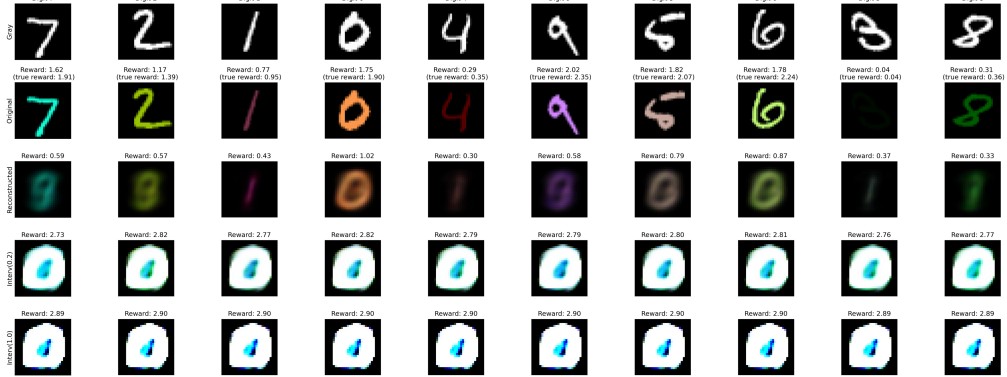

Figure 22: **IRAE[0]** on Case 2 DGP for one random seed (random seed 22), with a Dense AutoEncoder, linear HSIC as independence criterion, **latent dimension 3 = number of instruments**, regularization weights $\lambda = \mu_1 = 1$ and $\mu_2 = \mu_3 = 0$ and training for 50 epochs with early stopping (patience 5 epochs).

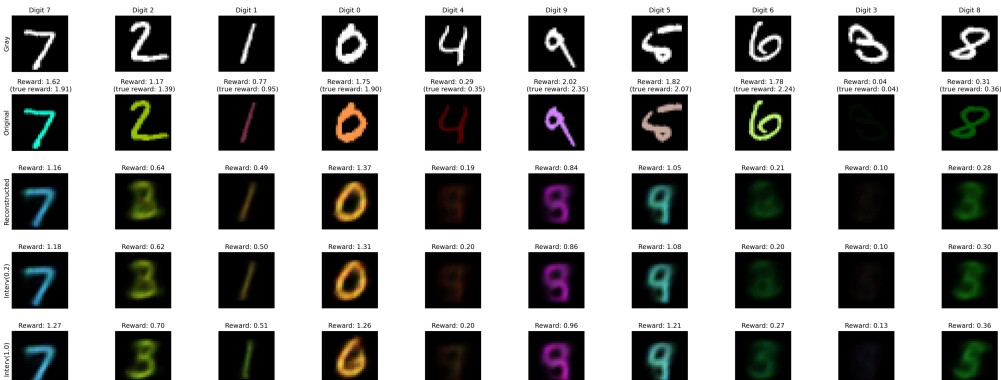

Figure 23: **Vanilla AE** on Case 2 DGP for one random seed (random seed 22), with a Dense AutoEncoder, linear HSIC as independence criterion, **latent dimension 3 = number of instruments**, regularization weights $\lambda = \mu_1 = \mu_2 = \mu_3 = 0$ and training for 50 epochs with early stopping (patience 5 epochs).

## F.5    CASE 3: CONFOUNDED OUTCOME

We examine the following confounded outcome generating process, where the instruments now affect the colors in a more convoluted intertwined manner. We denote this as *Case 3 DGP*.

All of models are trained with 60k training samples and evaluated on 10k test set, for 40 random seeds. Regularization weights are 0 or 1. All models are trained with 50 epochs after initialization with early stopping of patience 5.

---

**Case 3 DGP**

Draw DGP parameters $\alpha, \beta \sim \text{Unif}(0.1, 0.7)$. Then generate samples as:

$$G_i \in [0, 1]^{28 \times 28} \qquad \text{(grayscale MNIST image)}$$

$$Z_i, \, U_i \sim \mathcal{N}(0, I_3), \qquad Z_i \perp\!\!\!\perp U_i \qquad \text{(instrument \& confounder)}$$

$$r_i = \text{clip}(0.5 + \alpha\, Z_{i1} + \beta\, U_{i1},\, 0,\, 1) \qquad \text{(red channel)}$$

$$g_i = \text{clip}(0.5 + \alpha\, Z_{i2} + \beta\, U_{i2},\, 0,\, 1) \qquad \text{(green channel)}$$

$$b_i = \text{clip}(0.5 + \alpha\, Z_{i3} + \beta\, U_{i3},\, 0,\, 1) \qquad \text{(blue channel)}$$

$$X_i(k, \ell, c) = G_i(k, \ell) \cdot (r_i, g_i, b_i)_c, \qquad \begin{array}{l} c \in \{R, G, B\}, \\ (k, \ell) \in \{1, \dots, 28\}^2 \end{array} \qquad \text{(colour image)}$$

$$Y_i = r_i + g_i + b_i - U_{i1} - U_{i2} - U_{i3}. \qquad \text{(confounded outcome)}$$

Returns the tuples $(Z_i, X_i, Y_i)$.

---

In this confounding setting, we found that IRAE[0], IRAE[1], IRAE[2], IRAE still led to improved outcome, whereas Vanilla AE did not.

| Arch | Latent Dim | Reg Type | Warm Start | image | Vanilla AE | IRAE[0] | IRAE[1] | IRAE[2] | IRAE |
|------|-----------|----------|-----------|-------|-----------|---------|---------|---------|------|
| dense | 10 | linear | False | reconstructed | -0.46 (0.02) | -0.67 (0.02) | -0.67 (0.02) | **-0.27 (0.01)** | **-0.27 (0.02)** |
| | | | | intervened(0.2) | -0.45 (0.02) | **1.4 (0.12)** | **1.4 (0.1)** | 1.38 (0.15) | 1.36 (0.15) |
| | | | | intervened(1.0) | -0.37 (0.03) | 1.54 (0.11) | 1.54 (0.09) | 1.57 (0.12) | **1.58 (0.08)** |
| | 32 | linear | False | reconstructed | -0.46 (0.02) | -0.67 (0.02) | -0.67 (0.02) | **-0.14 (0.02)** | **-0.14 (0.02)** |
| | | | | intervened(0.2) | -0.45 (0.02) | **1.4 (0.12)** | **1.4 (0.1)** | 0.74 (0.34) | 0.63 (0.35) |
| | | | | intervened(1.0) | -0.37 (0.03) | **1.54 (0.11)** | **1.54 (0.09)** | 1.42 (0.32) | 1.34 (0.39) |
| conv | 10 | linear | False | reconstructed | -0.37 (0.02) | -0.6 (0.06) | -0.6 (0.05) | **-0.21 (0.03)** | -0.22 (0.03) |
| | | | | intervened(0.2) | -0.36 (0.03) | 0.21 (0.34) | 0.4 (0.4) | **0.98 (0.23)** | 0.83 (0.41) |
| | | | | intervened(1.0) | -0.31 (0.07) | 0.4 (0.56) | 0.69 (0.58) | 1.25 (0.54) | **1.28 (0.51)** |
| | 32 | linear | False | reconstructed | -0.37 (0.02) | -0.6 (0.06) | -0.6 (0.05) | **-0.1 (0.03)** | **-0.1 (0.03)** |
| | | | | intervened(0.2) | -0.36 (0.03) | 0.21 (0.34) | 0.4 (0.4) | **0.7 (0.33)** | 0.64 (0.38) |
| | | | | intervened(1.0) | -0.31 (0.07) | 0.4 (0.56) | 0.69 (0.58) | **1.26 (0.39)** | 1.11 (0.42) |

Figure 24: Experimental results for the **Case 3** data generating process. Mean improvement and standard deviation of improvement is reported.

## F.6    CASE 4: CONFOUNDED DGP WITH ONE OUTCOME RELEVANT DIMENSION

We examine the following confounded outcome generating process, where the instruments now affect the colors in a more convoluted intertwined manner. Moreover, only the red channel is relevant for the outcome and the outcome is confounded. We denote this as *Case 4 DGP*.

All of models are trained with 60k training samples and evaluated on 10k test set, for 40 random seeds. Regularization weights are 0 or 1. All models are trained with 50 epochs after initialization with early stopping of patience 5.

| Arch | Latent Dim | Reg Type | Warm Start | image | Vanilla AE | IRAE[0] | IRAE[1] | IRAE[2] | IRAE |
|------|-----------|----------|-----------|-------|-----------|---------|---------|---------|------|
| dense | 10 | linear | False | reconstructed | -0.16 (0.01) | -0.22 (0.01) | -0.22 (0.01) | **-0.09 (0.01)** | **-0.09 (0.01)** |
| | | | | intervened(0.2) | -0.15 (0.01) | **0.51 (0.03)** | 0.5 (0.03) | **0.51 (0.02)** | **0.51 (0.02)** |
| | | | | intervened(1.0) | -0.1 (0.02) | **0.55 (0.01)** | 0.55 (0.02) | **0.55 (0.01)** | **0.55 (0.01)** |
| | 32 | linear | False | reconstructed | -0.16 (0.01) | -0.22 (0.01) | -0.22 (0.01) | **-0.05 (0.01)** | **-0.05 (0.01)** |
| | | | | intervened(0.2) | -0.15 (0.01) | **0.51 (0.03)** | 0.5 (0.03) | 0.5 (0.01) | 0.49 (0.03) |
| | | | | intervened(1.0) | -0.1 (0.02) | **0.55 (0.01)** | 0.55 (0.02) | 0.54 (0.01) | 0.54 (0.01) |
| conv | 10 | linear | False | reconstructed | -0.13 (0.01) | -0.2 (0.02) | -0.2 (0.02) | **-0.07 (0.03)** | **-0.07 (0.03)** |
| | | | | intervened(0.2) | -0.13 (0.01) | 0.26 (0.12) | 0.28 (0.14) | **0.45 (0.03)** | 0.44 (0.05) |
| | | | | intervened(1.0) | -0.11 (0.04) | 0.42 (0.19) | 0.44 (0.21) | **0.54 (0.01)** | 0.53 (0.02) |
| | 32 | linear | False | reconstructed | -0.13 (0.01) | -0.2 (0.02) | -0.2 (0.02) | **-0.04 (0.03)** | **-0.04 (0.02)** |
| | | | | intervened(0.2) | -0.13 (0.01) | 0.26 (0.12) | 0.28 (0.14) | **0.45 (0.04)** | **0.45 (0.05)** |
| | | | | intervened(1.0) | -0.11 (0.04) | 0.42 (0.19) | 0.44 (0.21) | **0.53 (0.01)** | 0.51 (0.08) |

Figure 25: Experimental results for the **Case 4** data generating process. Mean improvement and standard deviation of improvement is reported.

---

**Case 4 DGP**

Draw DGP parameters $\alpha, \beta \sim \text{Unif}(0.1, 0.7)$. Then generate samples as:

$$G_i \in [0,1]^{28 \times 28} \qquad \text{(grayscale MNIST image)}$$

$$Z_i, \ U_i \sim \mathcal{N}(0, I_3), \qquad Z_i \perp\!\!\!\perp U_i \qquad \text{(instrument \& confounder)}$$

$$r_i = \text{clip}\big(0.5 + \alpha\,(Z_{i1} - Z_{i2}) + \beta\,U_{i1},\ 0,\ 1\big) \qquad \text{(red channel)}$$

$$g_i = \text{clip}\big(0.5 + \alpha\,(Z_{i2} - Z_{i3}) + \beta\,U_{i2},\ 0,\ 1\big) \qquad \text{(green channel)}$$

$$b_i = \text{clip}\big(0.5 + \alpha\,(Z_{i3} - Z_{i1}) + \beta\,U_{i3},\ 0,\ 1\big) \qquad \text{(blue channel)}$$

$$X_i(k, \ell, c) = G_i(k, \ell) \cdot (r_i, g_i, b_i)_c, \qquad \begin{matrix} c \in \{R, G, B\}, \\ (k, \ell) \in \{1, \ldots, 28\}^2 \end{matrix} \qquad \text{(colour image)}$$

$$Y_i = r_i - U_{i1}. \qquad \text{(confounded outcome)}$$

Returns the tuples $(Z_i, X_i, Y_i)$.

---

We demonstrate in this data generating process the importance of running an instrumental variable regression in the latent space. We see below that if instead we had run OLS regressing the outcome on the identified latent factors, then the direction would be erroneous and the interventional images will not be moving the image towards more red colors.

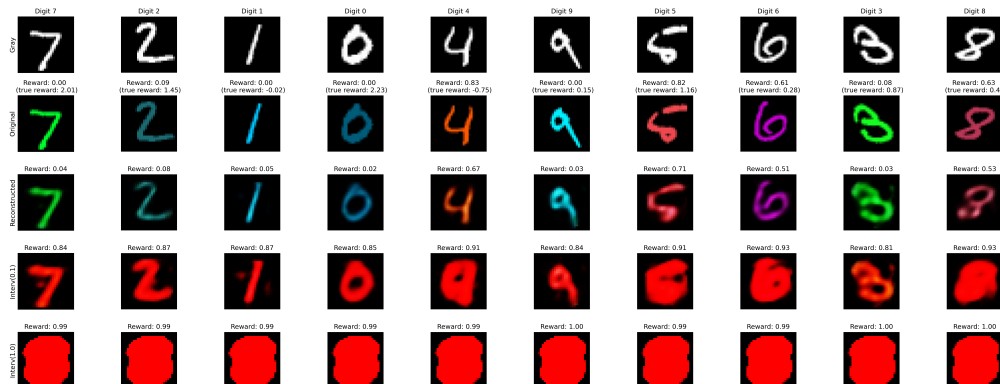

Figure 26: **IRAE** on **Case 4 DGP** for one random seed, with a Dense AutoEncoder, linear HSIC as independence criterion, **latent dimension 32**, regularization weights $\lambda = \mu_1 = \mu_2 = \mu_3 = 1$ and training for 50 epochs with early stopping (patience 5 epochs) from scratch (no warm start from IRAE1). Interventional images are intervened in the **direction identified by 2SLS** in the latent space with instrument $Z$, treatment $D$ and outcome $Y$. The outcome is larger when the color of the image is changed to red.

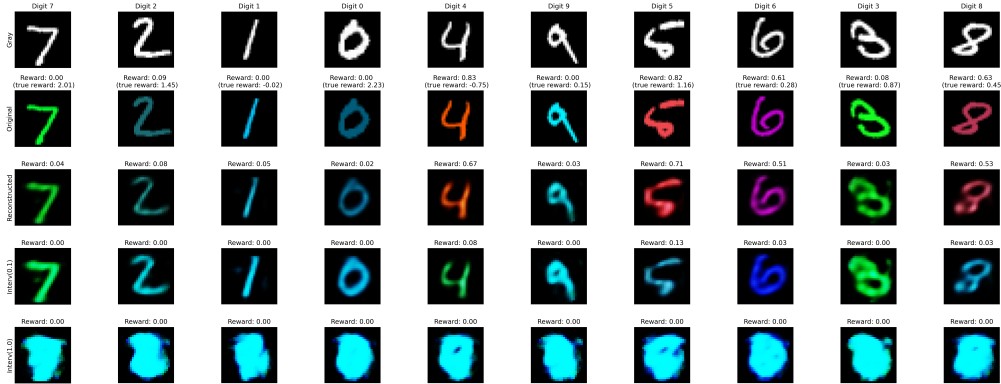

Figure 27: **IRAE** on **Case 4 DGP** for one random seed, with a Dense AutoEncoder, linear HSIC as independence criterion, **latent dimension 32**, regularization weights $\lambda = \mu_1 = \mu_2 = \mu_3 = 1$ and training for 50 epochs with early stopping (patience 5 epochs) from scratch (no warm start from IRAE[1]). Interventional images are intervened in the **direction identified by OLS**$(Y \sim D)$ in the latent space. The outcome is larger when the color of the image is changed to red.

## G  LLMS USE

Large language models (LLMs) were used to refine writing and generate preliminary code templates for experiments. All theoretical proofs, analyses, and substantive research contributions were conducted by the authors.

