# OpenReview forum: "Learning Treatment Representations for Downstream Instrumental Variable Regression"
_ICLR.cc/2026/Conference — Submitted to ICLR 2026_

### Official Review · Reviewer_Pw1E · 2025-10-17

**Soundness:** 3
**Presentation:** 3
**Contribution:** 3
**Rating:** 8
**Confidence:** 3

**Summary:**

This paper addresses the inability of IV models to handle high-dimensional treatments when instruments are scarce. The authors identify that existing two-step approaches that apply unsupervised dimension reduction (e.g., PCA) before IV regression violate the exclusion restriction and introduce omitted variable bias. To resolve this, they propose Instrument-Guided Representation Learning (IGRL), which incorporates instrumental variables directly into the representation learning process to ensure learned features capture only Z-driven variation in X. The approach provides both theoretical guarantees that the learned representations identify outcome-prediction-optimizing directions and empirical evidence of improvements over conventional two-stage methods.

**Strengths:**

- Learning representations of high-dimensional treatments is a difficult but relevant problem in many disciplines, e.g., when the treatment is a text
- The proposed method comes with theoretical improvement guarantees and promising empirical results
- The idea seems novel and rigorously executed

**Weaknesses:**

- The current version of the method does not allow for incorporating observed covariates, which could strengthen improvements in practice. Is this a straightforward extension or would this require more elaborate theory?
- As is common for IV methods, the proposed method depends on assumptions that may be difficult to verify in practice
- The specific setting (instruments + high-dimensional treatment) seems somewhat niche. A few clear motivating examples might strengthen the story of the paper.

**Questions:**

- I have one general question regarding the representation learning approach: Usually, we lose identifiability of the causal target quantity when using a treatment representation due to violation of the consistency assumption (no hidden variability within the treatment). Could you elaborate on how exactly your method addresses this problem? The results seem reasonable as they only provide improvement guarantees and no identifiable treatment effect.

---

> ### Author Response · Authors · 2025-11-25
>
> **W1: Incorporating observed covariates**
>
> We thank the reviewer for the comment. As noted in the footnote on page 3, we discuss the case of observed covariates in Appendix D. Our framework naturally extends to settings with observed confounders: the encoder, decoder, and predictor can incorporate covariates as inputs, and the independence regularizer can be replaced with a conditional independence regularizer.
>
> **W2: Testability of assumptions**
>
> While our method relies on several assumptions, most are standard in IV settings, such as instrument relevance, unconfoundedness, and bounded completeness. As in classical IV analysis, these assumptions can often be partially assessed using statistical tests or supported by domain knowledge. Our work follows this common practice, and we also discuss extensions that relax certain assumptions (e.g., conditioning on observed covariates) in Appendix D. Moreover, we note that most of our extra assumptions which are needed for the identifiability statement are a relaxation of the prior work of Rep4Ex [1], which considered a related structural causal model. Finally, we also tested scenarios where some of our key assumptions are violated and provided evidence for the robustness of our method in our experiments, as described below.
>
>
> *Violation of invertible encoding*
>
> In all MNIST experiments (Section 5 and Appendix E), the invertibility assumption is slightly violated because the observed images contain continuous color gradients, which cannot be perfectly mapped onto the latent color representation. We approximate this by performing 2-mean clustering to extract the dominant color (e.g., black background). Despite this, the learned latent captures the relevant variation for downstream IV regression, and the method performs well.
>
> *Violation of full-rank condition*
>
> In MNIST experiment 1 (Section 5), the treatment D is three-dimensional (R, G, B) while the instrument Z is two-dimensional (chosen for visualization). The true matrix A therefore cannot have full row rank. Even so, the method finds relevant two-dimensional representations that capture all information from Z, and still achieves strong performance in practice.
>
> *Violation of linearity of latent with respect to instruments*
>
> To keep color values valid, we clip entries of D to [0,1]. Most values vary linearly with Z as intended, though a few are clipped, creating slight deviations. These minor violations have little impact on downstream results.
>
> *Violation of other assumptions*
>
> The remaining assumptions correspond to standard regularity conditions in nonparametric IV, and are not tested separately.
>
> These results illustrate that the method is robust to small deviations from the assumptions, ensuring practical applicability even when conditions are not perfectly met.
>
> Reference:
>
> [1] Saengkyongam, S., Rosenfeld, E., Ravikumar, P., Pfister, N., & Peters, J. (2023). Identifying representations for intervention extrapolation. arXiv preprint arXiv:2310.04295.

---

> ### Author Response · Authors · 2025-11-25
>
> **W3: Motivating Example**
>
> Here is a more detailed description of the clinical data problem described in line 39:
>
> In clinical settings, treatment information is often recorded for administrative and billing purposes. Researchers thus observe detailed billing codes—such as medication charges, room and board, equipment use, and specialist services—rather than the underlying clinical decisions. Treatment choices are heavily confounded by patient health status and other unobserved factors. For example, severe fractures are more likely to be treated surgically, whereas milder fractures—or cases in which patients prefer to avoid surgery—may be managed with casting or observation. Such unmeasured clinical variation introduces substantial confounding when estimating causal effects.
>
> Prior work has addressed unobserved confounding by leveraging hospital capacity constraints or characteristics as instruments. However, these instruments are often limited to a small, hand-selected set of operational metrics (e.g., [2, 3, 4]), which constrains the number of treatment variables that can be studied.
>
> Given detailed billing codes, it is natural to hypothesize that the underlying treatment decisions generate these observed patterns. For instance, a decision to perform surgery may produce a combination of billing items including preoperative diagnostics, operating-room time, anesthesia, and postoperative care. Instead of hand-selecting treatment variables, one can apply dimension-reduction techniques to the observed billing codes to approximate the underlying treatment decision. This reduces dimensionality, matching the number of available instruments and enabling identification of causal effects.
>
>
> Reference:
>
> [2] Athey, S., & Wager, S. (2021). Policy learning with observational data. Econometrica, 89(1), 133-161.
>
> [3] Jing Dong, Pengyi Shi, Fanyin Zheng, and Xin Jin. Capacity management in networks: A structural estimation approach for hospital inpatient wards.
>
> [4] Jimmy Qin, Carri W Chan, Jing Dong, Shunichi Homma, and Siqin Ye. Waiting online versus in-person: An empirical study on outpatient clinic visit incompletion. 2023.
>
>
> **Q1: Identifiability and Consistency assumption**
>
> We agree that using learned treatment representations generally breaks the standard consistency assumption, and thus full identification of all treatment effects is not possible. Our method does not aim to identify every causal direction in the original treatment space. Instead, our framework focuses on the subset of directions that are identifiable given the instruments. The learned latent representation isolates exactly those components of the treatment for which the instruments provide valid variation. In this sense, our approach “flips” the usual causal question: rather than starting with a fixed intervention and asking whether it is identifiable, we learn which interventional directions can be identified and provide downstream improvement guarantees.

---

> > ### Comment · Reviewer_Pw1E · 2025-11-27
> >
> > Thank you for the detailed rebuttal. I have no further questions and will maintain my score.

---

### Official Review · Reviewer_9SdE · 2025-10-30

**Soundness:** 2
**Presentation:** 2
**Contribution:** 2
**Rating:** 2
**Confidence:** 4

**Summary:**

The paper proposes a method for representation learning of high-dimensional treatments that allows for a valid downstream instrumental variable treatment effect estimation setting. To do so, the paper proposes to incorporate IVs in the representation learning step. The procedure can be interpreted as a regularization of the unsupervised representation learning step to eliminate spurious backdoor paths. The paper compares the presented method with common statistical dimension reduction techniques, such as PCA, and ML methods, such as VAEs.

**Strengths:**

- The problem setting is interesting and novel to the best of my knowledge

**Weaknesses:**

- Line 67: The paper states that causal representation learning is uncommon in causal inference, but common in causal discovery. This is not true.
The paper thus lacks a sufficient discussion of related work on representation learning in causal inference tasks.
- The method requires very strong untestable assumptions, limiting its applicability in practice.
- The paper neither contains a discussion on the method, its limitations, or the results, nor a conclusion, leaving it an unfinished work. Potentially, these important parts of a research paper are neglected because of the page limit. However, this cannot be seen as an excuse.
- Mathematical statements are neither proven in the main paper nor are proofs referenced.
- The paper is lacking a proper motivation with precise use case examples. The problem statement seems a bit far-fetched and unrealistic. For example, what exactly are the available instruments in line 39? In my opinion, it is rather rare in practice to have suitable instruments
- Many references are incomplete or not stated correctly. This is not in line with proper scientific work.
- Many language/grammar errors. Sentences are incomplete, making it difficult for the reader to follow the line of thought.

**Questions:**

- Line 95: The data does not include any observed confounders. Why is this reasonable? If we assume all confounders are unobserved, we introduce a lot of  uncertainty in the modelling
- Why exactly do naïve representation learning approaches introduce omitted variable bias? This needs to be explained further.
- Assumption 2.1/ structural equation for X: This assumption is very strong. Why would such an encoding exist? How can we validate it?
- Experiments: Why is the proposed method not compared against other treatment representation learning methods? Even if they are not designed for IV
analysis, the comparison is of high value to assess the usefulness of the proposed method. A comparison with PCA is insufficient.

---

> ### Author Response · Authors · 2025-11-25
>
> We thank the reviewer for the detailed feedback and helpful suggestions.
>
> **W1: Related work**
>
> We believe that the majority of disentangled representation learning focuses on causal structure in the latent space, while another line of work aims at learning representations for downstream causal tasks (e.g. [1], [2]) that we discussed in the extended related work in Appendix A. The original intent of this description is to highlight that while our method leverages some independence of the latent structure, our method does not aim to find causal structure in latent space.
>
> Reference:
>
> [1] Nabi, R., McNutt, T., & Shpitser, I. (2022, August). Semiparametric causal sufficient dimension reduction of multidimensional treatments. In Uncertainty in Artificial Intelligence (pp. 1445-1455). PMLR.
>
> [2] Andreu, O. C., Vlontzos, A., O'Riordan, M., & Gilligan-Lee, C. M. (2024). Contrastive representations of high-dimensional, structured treatments. arXiv preprint arXiv:2411.19245.
>
> **W2: Testability of Assumptions**
>
> While our method relies on several assumptions, most are standard in IV settings, such as instrument relevance, unconfoundedness, and bounded completeness. As in classical IV analysis, these assumptions can often be partially assessed using statistical tests or supported by domain knowledge. Our work follows this common practice, and we also discuss extensions that relax certain assumptions (e.g., conditioning on observed covariates) in Appendix D. Moreover, we note that most of our extra assumptions which are needed for the identifiability statement are a relaxation of the prior work of Rep4Ex [3], which considered a related structural causal model. Finally, we also tested scenarios where some of our key assumptions are violated and provided evidence for the robustness of our method in our experiments, as described below.
>
>
> *Violation of invertible encoding*
>
> In all MNIST experiments (Section 5 and Appendix E), the invertibility assumption is slightly violated because the observed images contain continuous color gradients, which cannot be perfectly mapped onto the latent color representation. We approximate this by performing 2-mean clustering to extract the dominant color (e.g., digit color). Despite this, the learned latent captures the relevant variation for downstream IV regression, and the method performs well.
>
> *Violation of full-rank condition*
>
> In MNIST experiment 1 (Section 5), the treatment D is three-dimensional (R, G, B) while the instrument Z is two-dimensional (chosen for visualization). The true matrix A therefore cannot have full row rank. Even so, the method finds relevant two-dimensional representations that capture all information from Z, and still achieves strong performance in practice.
>
> *Violation of linearity of latent with respect to instruments*
>
> To keep color values valid, we clip entries of D to [0,1]. Most values vary linearly with Z as intended, though a few are clipped, creating slight deviations. TThese minor violations have little impact on downstream results.
>
> *Violation of other assumptions*
>
> The remaining assumptions correspond to standard regularity conditions in nonparametric IV, and are not tested separately.
>
> These results illustrate that the method is robust to small deviations from the assumptions, ensuring practical applicability even when conditions are not perfectly met.
>
> Reference:
>
>
> [3] Saengkyongam, S., Rosenfeld, E., Ravikumar, P., Pfister, N., & Peters, J. (2023). Identifying representations for intervention extrapolation. arXiv preprint arXiv:2310.04295.
>
> **W3: Proof/Conclusion**
>
> The formal proofs are included in Appendix Sections B, C, and D. In the revised manuscript, we will add explicit references in the main text to guide readers to these proofs.
>
> While a short discussion is included in Section 5, we will add a separate discussion section to highlight the performance differences between IRAE variants and discuss settings in which each regularization term is most helpful, providing practical guidelines for users. Moreover, we will also add a conclusion section summarizing our contributions, highlighting potential limitations, and reflecting on practical applicability.

---

> ### Author Response · Authors · 2025-11-25
>
> **W4: Motivating Example**
>
> Please find a more detailed description of the clinical data problem described in line 39 below:
>
> In clinical settings, treatment information is often recorded for administrative and billing purposes. Researchers thus observe detailed billing codes—such as medication charges, room and board, equipment use, and specialist services—rather than the underlying clinical decisions. Treatment choices are heavily confounded by patient health status and other unobserved factors. For example, severe fractures are more likely to be treated surgically, whereas milder fractures—or cases in which patients prefer to avoid surgery—may be managed with casting or observation. Such unmeasured clinical variation introduces substantial confounding when estimating causal effects.
>
> Prior work has addressed unobserved confounding by leveraging hospital capacity constraints or characteristics as instruments. However, these instruments are often limited to a small, hand-selected set of operational metrics (e.g., [4, 5, 6]), which constrains the number of treatment variables that can be studied.
>
> Given detailed billing codes, it is natural to hypothesize that the underlying treatment decisions generate these observed patterns. For instance, a decision to perform surgery may produce a combination of billing items including preoperative diagnostics, operating-room time, anesthesia, and postoperative care. Instead of hand-selecting treatment variables, one can apply dimension-reduction techniques to the observed billing codes to approximate the underlying treatment decision. This reduces dimensionality, matching the number of available instruments and enabling identification of causal effects.
>
> In the preliminary data description we obtained, we have instruments such as day of the week and time of arrival, which approximate hospital capacity constraints and have been used in prior work cited in our manuscript. In addition, we also have physician IDs, which is commonly used as an instrument, as discussed in Brookhart et al. [7]. We look forward to sharing our results in the future.
>
> Reference:
>
> [4] Athey, S., & Wager, S. (2021). Policy learning with observational data. Econometrica, 89(1), 133-161.
>
> [5] Jing Dong, Pengyi Shi, Fanyin Zheng, and Xin Jin. Capacity management in networks: A structural estimation approach for hospital inpatient wards.
>
> [6] Jimmy Qin, Carri W Chan, Jing Dong, Shunichi Homma, and Siqin Ye. Waiting online versus in-person: An empirical study on outpatient clinic visit incompletion. 2023.
>
> [7] Brookhart, M. A., & Schneeweiss, S. (2007). Preference-based instrumental variable methods for the estimation of treatment effects: assessing validity and interpreting results. The international journal of biostatistics, 3(1), 14.
>
> **W5: Incomplete Reference, W6: Language Error**
>
> We thank the reviewer for their feedback. We will carefully review the manuscript to ensure that all references are complete and correctly formatted in the revised version, and that language and grammar issues are addressed. We will also revise the text to improve clarity and readability throughout the paper.

---

> ### Author Response · Authors · 2025-11-25
>
> **Q1: Observed confounder**
>
> As noted in the footnote on page 3, we discuss the case of the observed confounders in Appendix D.
>
> **Q2: Omitted variable bias**
>
> As described in Section 2 (Paragraph 2) and illustrated in Figure 1, naive representation learning approaches can introduce omitted variable bias because the portion of the high-dimensional treatment X not captured in the learned representation D may contain elements correlated with both the instrument Z and the outcome Y. For example, using PCA as a representation method captures directions of largest variance, which are not necessarily the directions influenced by the instruments, potentially leaving instrument-relevant variation out of the representation.
>
> **Q3: Invertibility Assumption**
>
> This assumption requires that the observed high-dimensional treatment X is a mapping of some lower-dimensional, causally meaningful treatment (D, V).
>
> Realistic Example:
>
> In practice, such settings arise when many treatment features correspond to a small number of underlying decisions. For example, treatment features such as anesthesia type, operating room, or pain management protocol are often bundled into a small set of discrete surgical strategies—yielding a natural one-to-one mapping from the observed collection of actions to a lower-dimensional representation.
>
> Effect of minor violations:
>
> We hypothesize that minor violation from invertibility (e.g., slight entanglement or reconstruction error) primarily introduces noise in the encoded D rather than biasing the alignment between D and the instrument Z. Downstream IV regression is robust in practice because the first-stage regression selects only the instrument-aligned component of D, leading to nonnegative improvements. However, invertibility violation may affect interpretability. Specifically, we may not be able to perfectly decode an intervention back into the original human-readable treatment space.
>
> Empirical evidence (from our existing simulations):
>
> The Colored-MNIST Experiment 1 in Section 5 intentionally violates invertibility: the continuous color gradient in each image does not map to a unique latent color. Despite this violation, our method still improves downstream causal estimation, illustrating that Assumption 2.1 is a convenient sufficient condition rather than a strict practical requirement. Furthermore, our MNIST-2 ablation (Appendix E) further explores the invertibility criteria by allowing different dimensions of estimated V. A smaller space would prevent encoding of information necessary to decode back to the image, and we observed that intervened and reconstructed images become visually degraded as expected. This confirms that small violations of Assumption 2.1 do not harm downstream improvement, but extreme violations can reduce human-level interpretability of the decoded treatment.
>
> **Q4: Benchmark**
>
> In addition to PCA, we included comparisons against vanilla autoencoder (Vanilla AE), variational autoencoder (VAE), and identifiable VAE (iVAE), as well as variants of our method with different regularization weights. A key constraint in our setting is that the dimension of the learned D must not exceed the number of instruments. Many existing treatment representation learning methods cannot enforce the disentanglement between D and V, requiring the setting of a very low-dimensional latent representation similar to PCA or Vanilla AE (leading to poor reconstruction), or they rely on observed confounders, which are not available in our setting. Therefore, they are not directly applicable to our case.

---

### Official Review · Reviewer_zpPH · 2025-10-31

**Soundness:** 2
**Presentation:** 2
**Contribution:** 2
**Rating:** 4
**Confidence:** 2

**Summary:**

This paper addresses a fundamental limitation in instrumental variable (IV) regression when dealing with high-dimensional treatments: the violation of the exclusion restriction due to unsupervised dimensionality reduction. The authors propose Instrument-Guided Representation Learning (IGRL), a novel framework that integrates instruments directly into the representation learning process to ensure the latent treatment representation captures all instrument-driven variation. Through both linear (LIRR) and non-linear (IRAE) implementations, supported by theoretical guarantees and extensive experiments, the method demonstrates superior performance over conventional approaches in identifying outcome-improving interventions, effectively mitigating the omitted variable bias that plagues standard two-stage procedures.

**Strengths:**

The authors pinpoint how standard unsupervised dimensionality reduction of high dimensional treatments can violate the exclusion restriction by discarding instrument driven variation, a problem they term omitted treatment bias. The proposed solution of instrument guided representation learning represents a creative fusion of causal inference and representation learning, directly addressing this limitation in prior two stage approaches.

The authors develop a complete framework with specialized methods for both linear and nonlinear settings, supported by strong theoretical guarantees on the identifiability of valid intervention directions.  The writing is remarkably clear, with logical progression from linear to nonlinear cases.

**Weaknesses:**

The theoretical foundation of this paper relies on a set of strong structural assumptions that may be difficult to satisfy in real world applications. A key example is the core assumption of joint independence. This assumption requires the instrument, the confounder representation, and the orthogonal components to be fully independent. This is particularly challenging to guarantee with high dimensional and complex data. Furthermore, prerequisites such as the invertibility of the encoding and decoding functions and specific linear relationships among latent variables significantly limit the direct applicability of the theoretical model to practical scenarios. While these assumptions provide a necessary foundation for theoretical derivation, the paper does not sufficiently discuss the robustness of the estimates when these conditions are violated. This omission creates uncertainty for the method's practical deployment.

The paper exhibits notable shortcomings in reproducibility. This is particularly problematic for the non linear IRAE model. This model involves tuning regularization weights for up to six hyperparameters. The strategy for setting these critical parameters and their sensitivity to model performance are not elaborated. I encourage author provide anonymous github link.

**Questions:**

Please see my concerns in weakness.

---

> ### Author Response · Authors · 2025-11-25
>
> We thank the reviewer for the detailed feedback and helpful suggestions.
>
> **W1: Independence assumption**
>
> The joint independence assumption in our framework captures the two pair-wise independence relationships: $U  \perp Z$ and $D \perp V$. The former is the standard unconfoundedness assumption for instrumental variables. The latter is natural in our structural formulation, as it assumes that the endogenous (instrument-driven) and exogenous components of the treatment are independent. While the joint independence assumption is stronger than the standard assumptions above, they are required for identification results.
>
> We acknowledge that joint independence can be difficult to guarantee in high-dimensional or complex data. A weaker assumption is conditional joint independence, where the dependence can be explained by some observed covariates. This extension is discussed in Appendix D, providing a practical way to relax joint independence.
>
> **W2: Invertible encoding**
>
> This assumption requires that the observed high-dimensional treatment X is a mapping of some lower-dimensional, causally meaningful treatment (D, V).
>
> Realistic Example:
>
> In practice, such settings arise when many treatment features correspond to a small number of underlying decisions. For example, treatment features such as anesthesia type, operating room, or pain management protocol are often bundled into a small set of discrete surgical strategies—yielding a natural one-to-one mapping from the observed collection of actions to a lower-dimensional representation.
>
> Effect of minor violations:
>
> We hypothesize that minor violation from invertibility (e.g., slight entanglement or reconstruction error) primarily introduces noise in the encoded D rather than biasing the alignment between D and the instrument Z. Downstream IV regression is robust in practice because the first-stage regression selects only the instrument-aligned component of D, leading to nonnegative improvements. However, invertibility violation may affect interpretability. Specifically, we may not be able to perfectly decode an intervention back into the original human-readable treatment space.
>
> Empirical evidence (from our existing simulations):
>
> The Colored-MNIST Experiment 1 in Section 5 intentionally violates invertibility: the continuous color gradient in each image does not map to a unique latent color. Despite this violation, our method still improves downstream causal estimation, illustrating that Assumption 2.1 is a convenient sufficient condition rather than a strict practical requirement. Furthermore, our MNIST-2 ablation (Appendix E) further explores the invertibility criteria by allowing different dimensions of estimated V. A smaller space would prevent encoding of information necessary to decode back to the image, and we observed that intervened and reconstructed images become visually degraded as expected. This confirms that small violations of Assumption 2.1 do not harm downstream improvement, but extreme violations can reduce human-level interpretability of the decoded treatment.
>
>
> **W3: Linearity between instrument and latent decisions**
>
> We acknowledge that assuming linear relationships among latent variables may limit applicability in complex practical scenarios. The linearity assumption is necessary in our theory to prove that the learned latent representation is an invertible transformation of the true latent representation. This assumption is also used in prior work [1].
>
> Reference:
>
> [1] Saengkyongam, S., Rosenfeld, E., Ravikumar, P., Pfister, N., & Peters, J. (2023). Identifying representations for intervention extrapolation. arXiv preprint arXiv:2310.04295.

---

> ### Author Response · Authors · 2025-11-25
>
> **W4: Robustness when the assumptions are violated**
>
> While the assumptions provide a foundation for our theoretical derivations, our experiments show that the method remains effective even under minor violations:
>
> *Violation of invertible encoding*
>
> In all MNIST experiments (Section 5 and Appendix E), the invertibility assumption is slightly violated because the observed images contain continuous color gradients, which cannot be perfectly mapped onto the latent color representation. We approximate this by performing 2-mean clustering to extract the dominant color (e.g., black background). Despite this, the learned latent captures the relevant variation for downstream IV regression, and the method performs well.
>
> *Violation of full-rank condition*
>
> In MNIST experiment 1 (Section 5), the treatment D is three-dimensional (R, G, B) while the instrument Z is two-dimensional (chosen for visualization). The true matrix A therefore cannot have full row rank. Even so, the method finds relevant two-dimensional representations that capture all information from Z, and still achieves strong performance in practice.
>
> *Violation of linearity of latent with respect to instruments*
>
> To keep color values valid, we clip entries of D to [0,1]. Most values vary linearly with Z as intended, though a few are clipped, creating slight deviations. These minor violations have little impact on downstream results.
>
> *Violation of other assumptions*
>
> The remaining assumptions correspond to standard regularity conditions in nonparametric IV, and are not tested separately.
>
> These results illustrate that the method is robust to small deviations from the assumptions, ensuring practical applicability even when conditions are not perfectly met.
>
> **W5: Reproducibility**
>
> We have now attached a zip file of our code in the supplementary material. We plan to make our GitHub public once the review process ends.

---

### Official Review · Reviewer_XpqP · 2025-11-01

**Soundness:** 3
**Presentation:** 1
**Contribution:** 3
**Rating:** 4
**Confidence:** 3

**Summary:**

This paper considers an instrument variable setting with a potentially higher dimensional confounded treatment. The paper argues that naively adapted methods for such a setting, adopting dimensionality reduction on the treatment first and then performing IV regression, may introduce omitted variable bias. To address this challenge, the authors propose a framework to incorporate the instrument information during the dimensionality reduction/representation learning of the treatment to ensure improvement guarantees on soft interventions in this setting. The authors derive a theory and apply their method to both linear and non-linear setups, providing experimental results on (semi-)synthetic datasets.

**Strengths:**

- The IV setting with potentially high dimensional confounded treatments is an important and underexplored research direction in causal inference research.
- The motivation showing that omitted variable bias can also come from dimensionality reduction/representation learning of treatments (and not only of cofounders like most previous work showed) is an interesting and important finding
- Their method, ensuring that the IV information is maintained in the treatment representation, is a nice and intuitive idea and the theoretical analysis in linear and non-linear setting helps to motivate this choice.

**Weaknesses:**

The clarity of the paper could be improved and implications of different parts of the method could be mentioned.
- Part of the main motivation is that IV estimators “can only accommodate as many endogenous treatment variables as available instruments”. I think while in general it makes sense that effect estimation with high dimensional treatments is challenging, this statement should be explained more and shown in more detail.
- I think the related work is a bit limited to some specific works and could be extended to better show the contribution of their method with respect to existing works, in particular around representation learning for high dimensional confounders and OMV bias (e.g., https://www.pnas.org/doi/abs/10.1073/pnas.2427298122, https://arxiv.org/abs/2311.11321, https://arxiv.org/abs/2502.04274 ) , highdimensional treatments (e.g., https://arxiv.org/abs/2009.14061, https://arxiv.org/abs/2106.01939, https://arxiv.org/abs/2301.12292 ) , and representation learning for IVs (e.g., https://arxiv.org/abs/1612.09596 ).
- Assumptions 2.1 and 2.2, i.e. invertibility of encodings and existence and full rank of A, are specific to their novel setting and method. Thus, the implications of these assumptions should be discussed in more detail. How realistic are these in practice and how would minor violations of these assumptions affect the framework (theoretically/some intuition and maybe even provide some ablation study on simulated data)?
- The motivation of choosing soft interventions only and not deterministic ones is not fully clear to me. From a practical perspective, I think this would assume that during inference time we first can sample from the observational policy which might limit the practicality in real-world applications?

Presentation could be improved, especially in the experiments section
- The baselines should be introduced and described properly, also there is currently a mix between linear and non-linear setting (baselines are already referenced first and then described later).
- In the tables, it would be better to display their method in the last rows and not in the middle to better show their method.
- Table 3 over sample size could also be a plot to better compare the methods with sample size as x axis
- Intuition on why the different IRAE[x] variants perform differently and and some discussion on in which general settings the respective regularization terms are helpful and maybe some guideline when to use which variant could be helpful.
- real world applicability not clear, no motivation for real world application or any experiments to give some intuition provided (validation not possible but some insights could be shown in real world data)
- No discussion/conclusion section is provided. I think a short summary and mentioning of potential limitations would strengthen the paper.
- Presentation and fontsize of the plots needs to be improved clearly. Also, Figure 3 is hard to interpret and understand in general.

Overall, I think when my main concerns regarding unclarities are answered, I think this is a nice and interesting paper from the content side and I am open to increase my score. However, I think especially regarding the presentation (Figures, providing discussion, etc.), substantive changes are required for publication.

**Questions:**

- How realistic are Assumptions 2.1 and 2.2 in practice and how would minor violations of these assumptions affect the framework (theoretically/some intuition and maybe even provide some ablation study on simulated data)?
- Could the authors clarify why in ll. 1.66 they consider soft interventions and not deterministic ones and how they define them?
- Could the authors elaborate more on when it is useful/necessary to learn representations of V and when learning only D is sufficient?

---

> ### Author Response · Authors · 2025-11-25
>
> We thank the reviewer for the detailed feedback and helpful suggestions.
>
>
> **W1: Clarification on Underidentification of IV**
>
> In the classical IV framework, when the number of instruments is smaller than the dimension of the endogenous treatment, or when instruments are weak, the model becomes underidentified. In this case, multiple solutions for the structural parameters exist and the treatment effect cannot be uniquely determined.
>
> More concretely, in the linear IV setting, the parameters of the causal model can be thought of as the solution the linear system:
> $$\mathbb{E}[ZX^\top] \theta = \mathbb{E}[Zy]$$
> If Z has fewer dimensions than X, then the above system is underdetermined and has multiple solutions.
>
> This limitation is well-documented in the IV literature, and several tests for underidentification have been developed for linear models (e.g., [1, 2]). We will revise the introduction to clearly describe this identification issue and the related citations.
>
> Reference:
>
> [1] Murray, M. P. (2006). Avoiding invalid instruments and coping with weak instruments. Journal of Economic Perspectives, 20(4), 111-132.
>
> [2] Windmeijer, F. (2024). Testing underidentification in linear models, with applications to dynamic panel and asset pricing models. Journal of Econometrics, 240(2), 105104.
>
> **W2: Related Work**
>
> We thank the reviewer for the additional pointers to related work. We would like to mention that many of these directions of related work have been covered in the Appendix A, where we expand on our discussion of prior work. Due to space constraints we moved some discussions there. We will include in the main text some of these connections and expand the appendix with some of the references provided.
>
> *High-dimensional confounders and omitted-variable (OMV) bias*
>
> As discussed in the introduction, prior work on OMV bias, including Vafa et al. [3], assumes that all confounders are present in the high-dimensional treatment data and focuses on learning representations that retain this confounding information. For example, Vafa et al. [3] add a group-membership prediction loss to the wage-prediction objective to encourage the representation to preserve confounding features. In GraphITE [4] (reviewed in Appendix A), a separate model learns molecular-structure representations and employs HSIC penalties to disentangle treatment and confounding components before feeding them into the outcome model. Kaddour et al. [5] extend this approach using Robin Decomposition to separate confounding and treatment representations.
>
> Melnychuk et al. [6] address OMV bias using orthogonal (OR) learners; however, these methods still require that all confounders be observed so that the unconstrained nuisance estimators in OR-learners can correct bias. Similarly, Nilforoshan et al. [7], and Melnychuk et al [8] focus on the setting where all confounding is observed for CATE estimation.
>
> In contrast, our work studies the setting where confounders are unobserved but instruments are available, enabling identification by leveraging IV structure rather than assuming the treatment contains all confounding information.
>
> *Representation learning for IVs*
>
> We thank the reviewer for highlighting Deep IV [9]. While this line of work is directly related to IV-based inference, these methods rely on modeling the full conditional distribution p(X|Z, W) where X is the high dimensional treatment, W is the observed confounder, and Z is the instrument. This becomes intractable when X is high dimensional, whereas our method is specifically designed to address this challenge through treatment representation learning with the relatively tractable loss function (Eq. IRAE).
>
> Reference:
>
> [3] Vafa, K., Athey, S., & Blei, D. M. (2025). Estimating wage disparities using foundation models. Proceedings of the National Academy of Sciences, 122(22), e2427298122.
>
> [4] Harada, S., & Kashima, H. (2021, October). Graphite: Estimating individual effects of graph-structured treatments. In Proceedings of the 30th ACM international conference on information & knowledge management (pp. 659-668).
>
> [5] Kaddour, J., Zhu, Y., Liu, Q., Kusner, M. J., & Silva, R. (2021). Causal effect inference for structured treatments. Advances in Neural Information Processing Systems, 34, 24841-24854.
>
> [6] Melnychuk, V., Frauen, D., Schweisthal, J., & Feuerriegel, S. (2025). Orthogonal representation learning for estimating causal quantities. arXiv preprint arXiv:2502.04274.
>
> [7] Nilforoshan, H., et al. (2023). Zero-shot causal learning. Advances in Neural Information Processing Systems, 36, 6862-6901.
>
> [8] Melnychuk, V., Frauen, D., & Feuerriegel, S. (2023). Bounds on representation-induced confounding bias for treatment effect estimation. arXiv preprint arXiv:2311.11321.
>
> [9] Hartford, J., Lewis, G., Leyton-Brown, K., & Taddy, M. (2016). Counterfactual prediction with deep instrumental variables networks. arXiv preprint arXiv:1612.09596.

---

> ### Author Response · Authors · 2025-11-25
>
> **W3: Assumptions 2.1 Invertible Encoding and 2.2 Full-Rank Latents**
>
> Assumptions 2.1 – 2.2 ensure that the latent treatments (D, V) are well separated into an endogenous and exogenous component that enables downstream IV methods. We agree that dropping some of these restrictions is an important direction for future work (dropping the full rank assumption seems a more feasible next step, but dropping the invertibility would require a lot of substantial new ideas); though we believe that the identifiability result we provide in this paper is already highly non-trivial and the assumptions are realistic in many settings as illustrated in the example provided below. Moreover, some of our experiments provide evidence that the resulting algorithm that we propose is robust to violations of these assumptions, details below.
>
> *Assumption 2.1 (Invertible Encoding)*
>
> This assumption requires that the observed high-dimensional treatment X is a mapping of some lower-dimensional, causally meaningful treatment (D, V).
>
> Realistic Example:
>
> In practice, such settings arise when many treatment features correspond to a small number of underlying decisions. For example, treatment features such as anesthesia type, operating room, or pain management protocol are often bundled into a small set of discrete surgical strategies—yielding a natural one-to-one mapping from the observed collection of actions to a lower-dimensional representation.
>
> Effect of minor violations:
>
> We hypothesize that minor violation from invertibility (e.g., slight entanglement or reconstruction error) primarily introduces noise in the encoded D rather than biasing the alignment between D and the instrument Z. Downstream IV regression is robust in practice because the first-stage regression selects only the instrument-aligned component of D, leading to nonnegative improvements. However, invertibility violation may affect interpretability. Specifically, we may not be able to perfectly decode an intervention back into the original human-readable treatment space.
>
> Empirical evidence (from our existing simulations):
>
> The Colored-MNIST Experiment 1 in Section 5 intentionally violates invertibility: the continuous color gradient in each image does not map to a unique latent color. Despite this violation, our method still improves downstream causal estimation, illustrating that Assumption 2.1 is a convenient sufficient condition rather than a strict practical requirement. Furthermore, our MNIST-2 ablation (Appendix E) further explores the invertibility criteria by allowing different dimensions of estimated V. A smaller space would prevent encoding of information necessary to decode back to the image, and we observed that intervened and reconstructed images become visually degraded as expected. This confirms that small violations of Assumption 2.1 do not harm downstream improvement, but extreme violations can reduce human-level interpretability of the decoded treatment.
>
> *Assumption 2.2 (Full-Rank Latents)*
>
> Under the linear structural equation $D=AZ+U$, full row rank of A, or equivalently $\text{rank}(\text{Cov}(D,Z))=\text{dim}(D)$, ensures that every direction in the treatment space responds to some perturbation in the instrument. This is the natural generalization of the classical IV relevance assumption to multivariate or continuous treatments.
>
> Realistic example:
>
> This relevance assumption is standard in IV practice. Researchers typically select instruments known to affect treatment based on prior studies. For instance, in a patient treatment setting, hospital capacity constraints have been used as instruments [10]: if a hospital is busy, patients may receive temporary or alternative treatments before scheduling long-term or invasive treatment, creating variation in treatment that instruments can exploit.
>
> Effect of minor violations:
>
> We note that rank deficiencies in A are analogous to weak or irrelevant instruments in classical IV models.
>
> Empirical evidence (from our existing simulations):
>
> In Colored-MNIST Experiment 1 in Section 5, A is slightly rank-deficient (2 instruments for 3-dimensional color) for visualization purposes, but our model still improves downstream causal estimation as we can restrict the learned dimension to 2, the same as the dimension of instrument $Z$. Ablation studies with fully rank-sufficient A are provided in Appendix E, Colored-MNIST Experiment 2.
> We will revise the main text to include a brief discussion of the practical significance of these assumptions and explicitly clarify how they are violated in our experiments.
>
> Reference:
>
> [10] Jing Dong, Pengyi Shi, Fanyin Zheng, and Xin Jin. Off-service placement in inpatient ward network: Resource pooling versus service slowdown. Columbia Business School Research Paper. 2019.

---

> ### Author Response · Authors · 2025-11-25
>
> **W4: Soft Intervention**
>
> We used the term “soft intervention” to describe that the intervention refers to a deviation from the original assignment (so it depends on the observed treatment). To avoid further confusion, we will update our description to “incremental intervention”, or “infinitesimal nudges to continuous treatment” as seen in other literature [10, 11].
>
> Reference:
>
> [10] Athey, S., & Wager, S. (2021). Policy learning with observational data. Econometrica, 89(1), 133-161.
>
> [11] Rothenhäusler, D., & Yu, B. (2019). Incremental causal effects. arXiv preprint arXiv:1907.13258.
>
>
> **W5: Presentation**
>
> We thank the reviewer for the detailed feedback and helpful suggestions.
>
> *Experiments & baselines:*
>
> We have included full descriptions of all baselines in Appendix E. In the revised manuscript, we will rewrite the experimental sections to clearly introduce and discuss each baseline, resolve the mix between linear and non-linear settings, and adjust the order in tables to display our method in the last row for clarity.
>
> *Figures & tables:*
>
> We will improve figure readability and font sizes, convert Table 3 (sample size comparison) into a plot with sample size on the x-axis, and clarify Figure 3 to make it easier to interpret.
>
> *Real-world motivation:*
>
> While some examples were already included in the introduction, we will expand one example below (also seen in W3):
> In clinical settings, treatment information is often recorded for administrative and billing purposes. Researchers thus observe detailed billing codes—such as medication charges, room and board, equipment use, and specialist services—rather than the underlying clinical decisions. Treatment choices are heavily confounded by patient health status and other unobserved factors. For example, severe fractures are more likely to be treated surgically, whereas milder fractures—or cases in which patients prefer to avoid surgery—may be managed with casting or observation. Such unmeasured clinical variation introduces substantial confounding when estimating causal effects.
>
> Prior work has addressed unobserved confounding by leveraging hospital capacity constraints or characteristics as instruments. However, these instruments are often limited to a small, hand-selected set of operational metrics (e.g., [10, 12, 13]), which constrains the number of treatment variables that can be studied.
>
> Given detailed billing codes, it is natural to hypothesize that the underlying treatment decisions generate these observed patterns. For instance, a decision to perform surgery may produce a combination of billing items including preoperative diagnostics, operating-room time, anesthesia, and postoperative care. Instead of hand-selecting treatment variables, one can apply dimension-reduction techniques to the observed billing codes to approximate the underlying treatment decision. This reduces dimensionality, matching the number of available instruments and enabling identification of causal effects.
>
> Reference:
>
> [12] Jing Dong, Pengyi Shi, Fanyin Zheng, and Xin Jin. Capacity management in networks: A structural estimation approach for hospital inpatient wards.
>
> [13] Jimmy Qin, Carri W Chan, Jing Dong, Shunichi Homma, and Siqin Ye. Waiting online versus in-person: An empirical study on outpatient clinic visit incompletion. 2023.
>
> *IRAE variants and Discussion:*
>
> While a short discussion is included in Section 5, we will add a separate discussion section to highlight the performance differences between IRAE variants and discuss settings in which each regularization term is most helpful, providing practical guidelines for users.
>
> *Conclusion:*
>
> We will add a short conclusion section summarizing our contributions, highlighting potential limitations, and reflecting on practical applicability.

---

> ### Author Response · Authors · 2025-11-25
>
> **Q1: Assumptions 2.1-2.2**
>
> See W3.
>
> **Q2: Soft Intervention**
>
> See W4.
>
> **Q3: Learning V**
>
> There are several considerations for including an exogenous component V in the learned representation. These are briefly discussed in MNIST Experiment 1 (Section 5), and we will expand this discussion in the revised manuscript:
>
> *Invertible Encoding Constraint*
>
> Having the flexibility to include other latent dimensions makes the invertible encoding assumption more plausible.
>
> *Computational Consideration*
>
> If we did not include V in estimation, then the minimizer of our IRAE loss objective could choose to encode some of the V-related information onto the learned space, trading off against information from instrument-relevant D, in order to improve the reconstruction error. Including V explicitly allows the encoder to separate instrument-driven variation, avoiding mixing and improving downstream IV performance.
>
> *Interpretability*
>
> Retaining V in estimation improves interpretability by supporting higher-quality reconstructions and making it clear how latent interventions map back to observable treatments. As shown in MNIST-2 ablations (Appendix E), overly aggressive dimensionality reduction can produce unreadable reconstructions, whereas a separate V preserves human-interpretable structure.
>
> *Outcome prediction*
>
> While not tested in our simulations, V may carry predictive information useful for auxiliary tasks (e.g., modeling costs or secondary outcomes). Although V is unnecessary for IV identification, including it can benefit multi-task or predictive modeling applications.
>
> Overall, learning V is not required for downstream improvement but can substantially aid computation, interpretability, and auxiliary predictive tasks. We will incorporate these into the discussion section in the revised version.

---

### Meta-Review · Area_Chair_MZHn · 2026-01-05

**Summary:**

This paper aims to overcome the representation bias accounting for the directional information between IV and treatment when compressing the high-dimensional treatments. To this end, authors proposes a Instrument Regularized Auto-Encoder.

My main concerns align with Reviewer zpPH that too many restrictive assumptions are imposed in this paper, raising critical doubts that whethe the proposed method can achieve broader applications in real-world analysis, e.g., the multi-dimensional MR analysis in bio-technology. Aftering reading the rebuttal, authors cannot address this core issue, without experimental results on the robustness when the assumptions are violated.

Overall, my recommendation is reject.

**Reviewer Concerns:**

- Reviewer zpPH: too many structural assumptions are imposed in this paper

**Reviewer Scores:**

No reviewer will change the score.

---

### Decision · Program_Chairs · 2026-01-26

Reject